# Choose a Transformer: Fourier or Galerkin

**Shuhao Cao**
Department of Mathematics and Statistics
Washington University in St. Louis
s.cao@wustl.edu

## Abstract

In this paper, we apply the self-attention from the state-of-the-art Transformer in *Attention Is All You Need* [88] for the first time to a data-driven operator learning problem related to partial differential equations. An effort is put together to explain the heuristics of, and to improve the efficacy of the attention mechanism. By employing the operator approximation theory in Hilbert spaces, it is demonstrated for the first time that the softmax normalization in the scaled dot-product attention is sufficient but not necessary. Without softmax, the approximation capacity of a linearized Transformer variant can be proved to be comparable to a Petrov-Galerkin projection layer-wise, and the estimate is independent with respect to the sequence length. A new layer normalization scheme mimicking the Petrov-Galerkin projection is proposed to allow a scaling to propagate through attention layers, which helps the model achieve remarkable accuracy in operator learning tasks with unnormalized data. Finally, we present three operator learning experiments, including the viscid Burgers' equation, an interface Darcy flow, and an inverse interface coefficient identification problem. The newly proposed simple attention-based operator learner, Galerkin Transformer, shows significant improvements in both training cost and evaluation accuracy over its softmax-normalized counterparts.

## 1 Introduction

Partial differential equations (PDEs) arise from almost every multiphysics and biological systems, from the interaction of atoms to the merge of galaxies, from the formation of cells to the change of climate. Scientists and engineers have been working on approximating the governing PDEs of these physical systems for centuries. The emergence of the computer-aided simulation facilitates a cost-friendly way to study these challenging problems. Traditional methods, such as finite element/difference [20, 22], spectral methods [12], etc., leverage a discrete structure to reduce an infinite dimensional operator map to a finite dimensional approximation problem. Meanwhile, in the field practice of many scientific disciplines, substantial data for PDE-governed phenomena available on discrete grids enable modern black-box models like Physics-Informed Neural Network (PINN) [71, 62, 49] to exploit measurements on collocation points to approximate PDE solutions.

Nonetheless, for traditional methods or data-driven function learners such as PINN, given a PDE, the focus is to approximate a single instance, for example, solving for an approximated solution for one coefficient with a fixed boundary condition. A slight change to this coefficient invokes a potentially expensive re-training of any data-driven function learners. In contrast, an operator learner aims to learn a map between infinite-dimensional function spaces, which is much more difficult yet rewarding. A well-trained operator learner can evaluate many instances without re-training or collocation points, thus saving valuable resources, and poses itself as a more efficient approach in the long run. Data-driven resolution-invariant operator learning is a booming new research direction [60, 5, 56, 64, 90, 57, 61, 91, 37, 74]. The pioneering model, DeepONet [60], attributes architecturally to a universal approximation theorem for operators [18]. Fourier Neural Operator (FNO) [57] notably

shows an awing state-of-the-art performance outclassing classic models such as the one in [100] by orders of magnitudes in certain benchmarks.

Under a supervised setting, an operator learner is trained with the operator's input functions and their responses to the inputs as targets. Since both functions are sampled at discrete grid points, this is a special case of a `seq2seq` problem [81]. The current state-of-the-art `seq2seq` model is the Transformer first introduced in [88]. As the heart and soul of the Transformer, the scaled dot-product attention mechanism is capable of unearthing the hidden structure of an operator by capturing long-range interactions. Inspired by many insightful pioneering work in Transformers [50, 19, 75, 84, 96, 97, 95, 59, 76, 66], we have modified the attention mechanism minimally yet in a mathematically profound manner to better serve the purpose of operator learning.

Among our new Hilbert space-inspired adaptations of the scaled dot-product attention, the first and foremost change is: no softmax, or the approximation thereof. In the vanilla attention [88], the softmax succeeding the matrix multiplication convexifies the weights for combining different positions' latent representations, which is regarded as an indispensable ingredient in the positive kernel interpretation of the attention mechanism [84]. However, softmax acts globally in the sequence length dimension for each row of the attention matrix, and further adds to the quadratic complexity of the attention in the classic Transformer. Theory-wise, instead of viewing "row $\approx$ word" in the Natural Language Processing (NLP) tradition, the columns of the query/keys/values are seen as sampling of functions in Hilbert spaces on discretized grids. Thus, taking the softmax away allows us to verify a discrete Ladyzhenskaya–Babuška–Brezzi (LBB) condition, which further amounts to the proof that the newly proposed Galerkin-type attention can explicitly represent a Petrov-Galerkin projection, and this approximation capacity is independent of the sequence length (Theorem 4.3).

Numerically, the softmax-free models save valuable computational resources, outperforming the ones with the softmax in terms of training FLOP and memory consumption (Section 5). Yet in an ablation study, the training becomes unstable for softmax-free models (Table 8). To remedy this, a new Galerkin projection-type layer normalization scheme is proposed to act as a cheap diagonal alternative to the normalizations explicitly derived in the proof of the Petrov-Galerkin interpretation (equation (40)). Since a learnable scaling can now be propagated through the encoder layers, the attention-based operator learner with this new layer normalization scheme exhibits better comprehension of certain physical properties associated with the PDEs such as the energy decay. Combining with other approximation theory-inspired tricks including a diagonally dominant rescaled initialization for the projection matrices and a layer-wise enrichment of the positional encodings, the evaluation accuracies in various operator learning tasks are boosted by a significant amount.

**Main contributions.** The main contributions of this work are summarized as follows.

- **Attention without softmax.** We propose a new simple self-attention operator and its linear variant without the softmax normalization. Two new interpretations are offered, together with the approximation capacity of the linear variant proved comparable to a Petrov-Galerkin projection.

- **Operator learner for PDEs.** We combine the newly proposed attention operators with the current best state-of-the-art operator learner Fourier Neural Operator (FNO) [57] to significantly improve its evaluation accuracy in PDE solution operator learning benchmark problems. Moreover, the new model is capable of recovering coefficients based on noisy measurements that traditional methods or FNO cannot accomplish.

- **Experimental results.** We present three benchmark problems to show that operator learners using the newly proposed attentions are superior in computational/memory efficiency, as well as in accuracy versus those with the conventional softmax normalization. The PyTorch codes to reproduce our results are available as an open-source software. [1]

## 2   Related Works

**Operator learners related to PDEs.** In [4, 5], certain kernel forms of the solution operator of parametric PDEs are approximated using graph neural networks. The other concurrent notable approach is DeepONet [60, 61]. [56] further improves the kernel approach by exploiting the multilevel grid structure. [57] proposes a discretization-invariant operator learner to achieve a state-of-the-art

---

[1] https://github.com/scaomath/galerkin-transformer

performance in certain benchmark problems. [90, 91] proposed a DeepONet roughly equivalent to an additive attention, similar to the one in the Neural Turing Machine (NMT) in [7]. Model/dimension reduction combined with neural nets is another popular approach to learn the solution operator for parametric PDEs [10, 64, 55, 24]. Deep convolutional neural networks (DCNN) are widely applied to learn the solution maps with a fixed discretization size [1, 9, 40, 36, 35, 100, 86]. Recently, DCNN has been successfully applied in various inverse problems [35, 47] such as Electrical Impedance Tomography (EIT). To our best knowledge, there is no work on data-driven approaches to an inverse interface coefficient identification for a class of coefficients with random interface geometries.

**Attention mechanism and variants.** Aside from the ground-breaking scaled dot-product attention in [88], earlier [7] proposed an additive content-based attention, however, with a vanishing gradient problem due to multiple nonlinearity composition. [25] shows the first effort in removing the softmax normalization in [7] after the projection, however, it still uses a Sigmoid nonlinearity before the additive interpolation propagation stage, and performs worse than its softmax counterpart. The current prevailing approach to linearize the attention leverages the assumption of the existence of a feature map to approximate the softmax kernel [50, 19, 70]. Another type of linearization exploits the low-rank nature of the matrix product using various methods such as sampling or projection [73, 11, 79, 92], or fast multipole decomposition [65]. The conjecture in [75] inspires us to remove the softmax overall. [76] first proposed the inverse sequence length scaling normalization for a linear complexity attention without the softmax, however, the scaling normalization has not been extensively studied in examples and performs worse.

**Various studies on Transformers.** The kernel interpretation in [84] inspires us to reformulate the attention using the Galerkin projection. [95, Theorem 2] gives a theoretical foundation of removing the softmax normalization to formulate the Fourier-type attention. The Nyström approximation [97] essentially acknowledges the similarity between the attention matrix and an integral kernel. [96, 66, 59] inspires us to try different layer normalization and the rescaled diagonally dominant initialization schemes. The practices of enriching the latent representations with the positional encoding recurrently in our work trace back to [2, 26], and more recently, contribute to the success of AlphaFold 2 [48], as it is rewarding to exploit the universal approximation if the target has a dependence ansatz in the coordinate frame and/or transformation group but hard to be explicitly quantified. Other studies on adapting the attention mechanisms to conserve important physical properties are in [82, 31, 44].

## 3   Operator learning related to PDEs

Closely following the setup in [56, 57], we consider a data-driven model to approximate a densely-defined operator $T : \mathcal{H}_1 \to \mathcal{H}_2$ between two Hilbert spaces with an underlying bounded spacial domain $\Omega \subset \mathbb{R}^m$. The operator $T$ to be learned is usually related to certain physical problems, of which the formulation is to seek the solution to a PDE of the following two types.

Parametric PDE: given coefficient $a \in \mathcal{A}$, and source $f \in \mathcal{Y}$, find $u \in \mathcal{X}$ such that $L_a(u) = f$.

  (i) To approximate the nonlinear mapping from the varying parameter $a$ to the solution with a fixed right-hand side, $T : \mathcal{A} \to \mathcal{X}, \ a \mapsto u$.

 (ii) The inverse coefficient identification problem to recover the coefficient from a noisy measurement $\tilde{u}$ of the steady-state solution $u$, in this case, $T : \mathcal{X} \to \mathcal{A}, \ \tilde{u} \mapsto a$.

Nonlinear initial value problem: given $u_0 \in \mathcal{H}_0$, find $u \in C([0, T]; \mathcal{H})$ such that $\partial_t u + N(u) = 0$.

(iii) Direct inference from the initial condition to the solution. $T : \mathcal{H}_0 \to \mathcal{H}, \ u_0(\cdot) \mapsto u(t_1, \cdot)$ with $t_1 \gg \Delta t$ with $t_1$ much greater than the step-size in traditional explicit integrator schemes.

Using (i) as an example, based on the given $N$ observations $\{a^{(j)}, u^{(j)}\}_{j=1}^N$ and their approximations $\{a_h^{(j)}, u_h^{(j)}\}$ defined at a discrete grid of size $h \ll 1$, the goal of our operator learning problem is to build an approximation $T_\theta$ to $T$, such that $T_\theta(a_h)$ is a good approximation to $u = L_a^{-1} f =: T(a) \approx u_h$ independent of the mesh size $h$, where $a_h$ and $u_h$ are in finite dimensional spaces $\mathbb{A}_h, \mathbb{X}_h$ on this grid. We further assume that $a^{(j)} \sim \nu$ for a measure $\nu$ compactly supported on $\mathcal{A}$, and the sampled data form a reasonably sized subset of $\mathcal{A}$ representative of field applications. The loss $\mathcal{J}(\theta)$ is

$$\mathcal{J}(\theta) := \mathbb{E}_{a \sim \nu} \left[ \| \left( T_\theta(a) - u \|_{\mathcal{H}}^2 + \mathfrak{G}(a, u; \theta) \right] \right. \tag{1}$$

and in practice is approximated using the sampled observations on a discrete grid

$$\mathcal{J}(\theta) \approx \frac{1}{N} \sum_{j=1}^{N} \left\{ \left\| \left( T_\theta \left( a_h^{(j)} \right) - u_h^{(j)} \right) \right\|_{\mathcal{H}}^2 + \mathfrak{G} \left( a_h^{(j)}, u_h^{(j)}; \theta \right) \right\}. \qquad (2)$$

In example (i), $\|\cdot\|_{\mathcal{H}}$ is the standard $L^2$-norm, and $\mathfrak{G}(a, u; \theta)$ serves as a regularizer with strength $\gamma$ and is problem-dependent. In Darcy flow where $L_a := -\nabla \cdot (a\nabla(\cdot))$, it is $\gamma \|a\nabla(T_\theta(a) - u)\|_{L^2(\Omega)}^2$, since $u \in H^{1+\alpha}(\Omega)$ ($\alpha > 0$ depends on the regularity of $a$) and $a\nabla u \in \boldsymbol{H}(\mathrm{div}; \Omega)$ a priori. For the evaluation metric, we drop the $\mathfrak{G}(a, u; \theta)$ term, and monitor the minimization of (2) using $\|\cdot\|_{\mathcal{H}}$.

## 4 Attention-based operator learner

**Feature extractor.** We assume the functions in both inputs and targets are sampled on a uniform grid. In an operator learning problem on $\Omega \subset \mathbb{R}^1$, a simple feedforward neural network (FFN) is used as the feature extractor that is shared by every position (grid point).

**Interpolation-based CNN.** If $\Omega \subset \mathbb{R}^2$, inspired by the multilevel graph kernel network in [56], we use two 3-level interpolation-based CNNs (CiNN) as the feature extractor, but also as the downsampling and upsampling layer, respectively, in which we refer to restrictions/prolongations between the coarse/fine grids both as interpolations. For the full details of the network structure please refer to Appendix B.

**Recurrent enrichment of positional encoding.** The Cartesian coordinates of the grid, on which the attention operator's input latent representation reside, are concatenated as additional feature dimension(s) to the input, as well as to each latent representation in every attention head.

**Problem-dependent decoder.** The decoder is a problem-dependent admissible network that maps the learned representations from the encoder back to the target dimension. For smooth and regular solutions in $H^{1+\alpha}(\Omega)$, we opt for a 2-layer spectral convolution that is the core component in [57]. A simple pointwise feedforward neural network (FFN) is used for nonsmooth targets in $L^\infty(\Omega)$.

### 4.1 Simple self-attention encoder

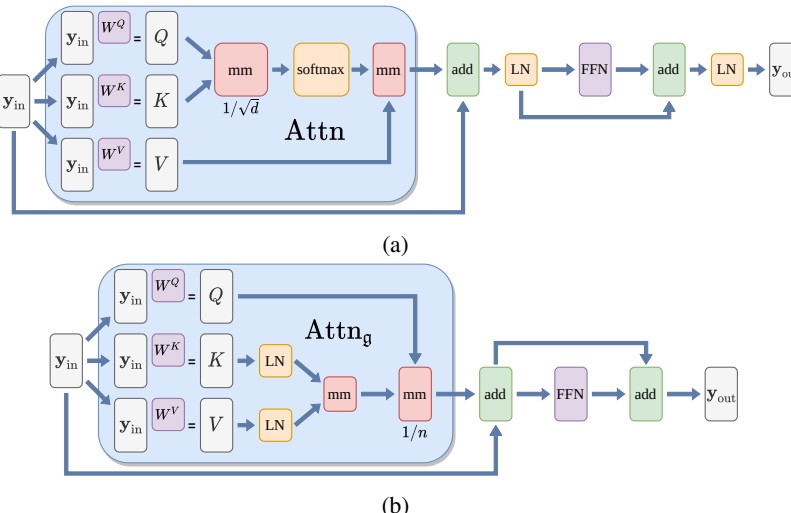

(a)

(b)

Figure 1: Comparison of the vanilla attention [88] with the Galerkin-type simple self-attention in a single head; (a) in the standard softmax attention, the softmax is applied row-wise after the matrix product `matmul`; (b) a mesh-weighted normalization allows an integration-based interpretation.

The encoder contains a stack of identical simple attention-based encoder layers. For simplicity, we consider a single attention head that maps $\mathbf{y} \in \mathbb{R}^{n \times d}$ to another element in $\mathbb{R}^{n \times d}$, and define the

trainable projection matrices, and the latent representations $Q/K/V$ as follows.

$$W^Q, W^K, W^V \in \mathbb{R}^{d \times d}, \quad \text{and} \quad Q := \mathbf{y}W^Q, \quad K := \mathbf{y}W^K, \quad V := \mathbf{y}W^V. \tag{3}$$

We propose the following simple attention that (i) uses a mesh (inverse sequence length)-weighted normalization without softmax, (ii) allows a scaling to propagate through the encoder layers.

$$\text{Attn}_{\text{sp}} : \mathbb{R}^{n \times d} \to \mathbb{R}^{n \times d}, \quad \widetilde{\mathbf{y}} \leftarrow \mathbf{y} + \text{Attn}_{\dagger}(\mathbf{y}), \quad \mathbf{y} \mapsto \widetilde{\mathbf{y}} + g(\widetilde{\mathbf{y}}), \tag{4}$$

where the head-wise normalizations are applied pre-dot-product: for $\dagger \in \{\mathfrak{f}, \mathfrak{g}\}$,

$$\text{(Fourier-type attention)} \qquad \mathbf{z} = \text{Attn}_{\mathfrak{f}}(\mathbf{y}) := (\widetilde{Q}\widetilde{K}^{\top})V/n, \tag{5}$$

$$\text{(Galerkin-type attention)} \qquad \mathbf{z} = \text{Attn}_{\mathfrak{g}}(\mathbf{y}) := Q(\widetilde{K}^{\top}\widetilde{V})/n, \tag{6}$$

and $\widetilde{\diamond}$ denotes a trainable non-batch-based normalization. As in the classic Transformer [88], and inspired by the Galerkin projection interpretation, we choose $\widetilde{\diamond}$ as the layer normalization $\text{Ln}(\diamond)$, and $g(\cdot)$ as the standard 2-layer FFN identically applied on every position (grid point). In simple attentions, the weight for each row of $V$, or column of $Q$ in the linear variant, is not all positive anymore. This can be viewed as a cheap alternative to the cosine similarity-based attention.

**Remark 4.1.** *If we apply the regular layer normalization rule that eliminates any scaling:*

$$\mathbf{y} \mapsto \text{Ln}\big(\mathbf{y} + \text{Attn}_{\dagger}(\mathbf{y}) + g\big(\text{Ln}(\mathbf{y} + \text{Attn}_{\dagger}(\mathbf{y}))\big)\big), \quad \text{where} \ \text{Attn}_{\dagger}(\mathbf{y}) := Q(K^{\top}V)/n, \tag{7}$$

*then this reduces to the efficient attention first proposed in [76].*

### 4.1.1 Structure-preserving feature map as a function of positional encodings

Consider an operator learning problem with an underlying domain $\Omega \subset \mathbb{R}^1$. $\{x_i\}_{i=1}^n$ denotes the set of grid points in the discretized $\Omega$ such that the weight $1/n = h$ is the mesh size. Let $\zeta_q(\cdot), \phi_k(\cdot), \psi_v(\cdot) : \Omega \to \mathbb{R}^{1 \times d}$ denote the feature maps of $Q, K, V$, i.e., the $i$-th row of $Q, K, V$ written as $\boldsymbol{q}_i = \zeta_q(x_i)$, $\boldsymbol{k}_i = \phi_k(x_i)$, $\boldsymbol{v}_i = \psi_v(x_i)$. They are, in the NLP convention, viewed as the feature (embedding) vector at the $i$-th position, respectively. The inter-position topological structure such as continuity/differentiability in the same feature dimension is learned thus not explicit. The following ansatz for $Q/K/V$ in the same attention head is fundamental to our new interpretations.

**Assumption 4.2.** *The columns of $Q/K/V$, respectively, contain the vector representations of the learned basis functions spanning certain subspaces of the latent representation Hilbert spaces.*

Using $V \in \mathbb{R}^{n \times d}$ with a full column rank as an example, its columns contain potentially a set of bases $\{v_j(\cdot)\}_{j=1}^d$ evaluated at the grid points (degrees of freedom, or DoFs). Similarly, the learned bases whose DoFs form the columns of $Q, K$ are denoted as $\{q_j(\cdot)\}_{j=1}^d$, $\{k_j(\cdot)\}_{j=1}^d$, as well as $\{z_j(\cdot)\}_{j=1}^d$ for the outputs in (5) and (6). To be specific, the $j$-th column of $V$, denoted by $\boldsymbol{v}^j$, then stands for a vector representation of the $j$-th basis function evaluated at each grid point, i.e., its $l$-th position stands for $(\boldsymbol{v}^j)_l = v_j(x_l)$. Consequently, the row $\boldsymbol{v}_i = (v_1(x_i), \ldots, v_d(x_i))$ can be alternatively viewed as the evaluation of a vector latent basis function at $x_i$.

### 4.1.2 Fourier-type attention of a quadratic complexity

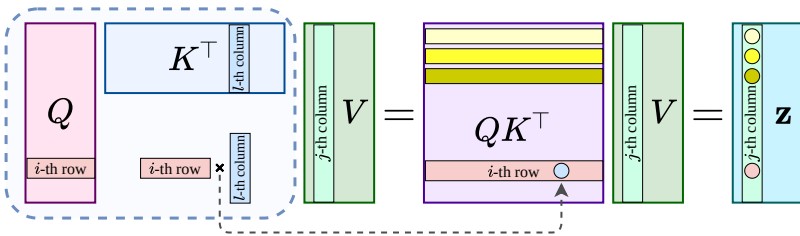

Figure 2: A dissection of Fourier-type attention's output. Both `matmuls` have complexity $O(n^2 d)$.

In the Fourier-type attention (5), $Q, K$ are assumed to be normalized for simplicity, the $j$-th column ($1 \leq j \leq d$) in the $i$-th row ($1 \leq i \leq n$) of $\mathbf{z}$ is computed by (see Figure 2):

$$
\begin{aligned}
(\boldsymbol{z}_i)_j &= h(QK^{\top})_{i\bullet} \, \boldsymbol{v}^j = h\big(\boldsymbol{q}_i \cdot \boldsymbol{k}_1, \ldots, \boldsymbol{q}_i \cdot \boldsymbol{k}_l, \ldots, \boldsymbol{q}_i \cdot \boldsymbol{k}_n\big)^{\top} \cdot \boldsymbol{v}^j \\
&= h \sum_{l=1}^{n} (\boldsymbol{q}_i \cdot \boldsymbol{k}_l)(\boldsymbol{v}^j)_l \approx \int_{\Omega} \big(\zeta_q(x_i) \cdot \phi_k(\xi)\big) v_j(\xi) \, \mathrm{d}\xi,
\end{aligned}
\tag{8}
$$

where the $h$-weight facilitates the numerical quadrature interpretation of the inner product. Concatenating columns $1 \leq j \leq d$ yields the $i$-row $\boldsymbol{z}_i$ of the output $\mathbf{z}$: $\boldsymbol{z}_i \approx \int_\Omega \left( \zeta_q(x_i) \cdot \phi_k(\xi) \right) \psi_v(\xi) \, \mathrm{d}\xi$. Therefore, without the softmax nonlinearity, the local dot-product attention output at $i$-th row computes approximately an integral transform with a non-symmetric learnable kernel function $\kappa(x, \xi) := \zeta_q(x)\phi_k(\xi)$ evaluated at $x_i$, whose approximation property has been studied in [95, Theorem 2], yet without the logits technicality due to the removal of the softmax normalization.

After the skip-connection, if we further exploit the learnable nature of the method and assume $W^V = \mathrm{diag}\{\delta_1, \cdots, \delta_d\}$ such that $\delta_j \neq 0$ for $1 \leq j \leq d$, under Assumption 4.2:

$$\delta_j^{-1} v_j(x) \approx z_j(x) - \int_\Omega \kappa(x, \xi) v_j(\xi) \, \mathrm{d}\xi, \quad \text{for } j = 1, \cdots, d, \text{ and } x \in \{x_i\}_{i=1}^n. \tag{9}$$

This is the forward propagation of the Fredholm equation of the second-kind for each $v_j(\cdot)$. When using an explicit orthogonal expansion such as Fourier to solve for $\{v_j(\cdot)\}_{j=1}^d$, or to seek for a better set of $\{v_j(\cdot)\}$ in our case, it is long known being equivalent to the Nyström's method with numerical integrations [8] (similar to the $h = 1/n$ weighted sum). Therefore, the successes of the random Fourier features in [19, 70] and the Nyströmformer's approximation [97] are not surprising.

Finally, we name this type of simple attention "Fourier" is due to the striking resemblance between the scaled dot-product attention and a Fourier-type kernel [30] integral transform, since eventually the target resides in a Hilbert space with an underlying spacial domain $\Omega$, while the latent representation space parallels a "frequency" domain on $\Omega^*$. This also bridges the structural similarity of the scaled dot-product attention with the Fourier Neural Operator [57] where the Fast Fourier Transform (FFT) can be viewed as a non-learnable change of basis.

### 4.1.3 Galerkin-type attention of a linear complexity

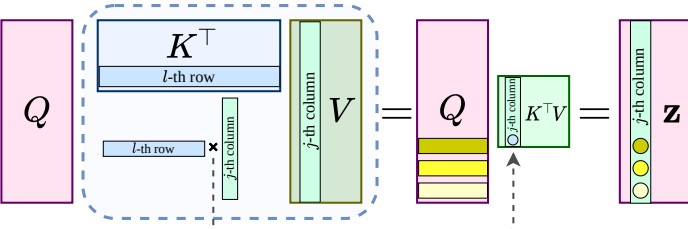

Figure 3: A dissection of Galerkin-type attention's output. Both `matmuls` have complexity $O(nd^2)$.

For the Galerkin-type simple attention in (6), $K, V$ are assumed to be normalized for simplicity, we first consider the $i$-th entry in the $j$-th column $\boldsymbol{z}^j$ of $\mathbf{z}$ (see Figure 3):

$$(\boldsymbol{z}^j)_i = h \, \boldsymbol{q}_i^\top \cdot (K^\top V)_{\bullet j}, \tag{10}$$

which is the inner product of the $i$-th row of $Q$ and the $j$-th column of $K^\top V$. Thus,

$$\boldsymbol{z}^j = h \left( \begin{array}{cccc} | & | & | & | \\ \boldsymbol{q}_1 & \boldsymbol{q}_2 & \cdots & \boldsymbol{q}_n \\ | & | & | & | \end{array} \right)^\top (K^\top V)_{\bullet j} = h \left( (K^\top V)_{\bullet j}^\top \left( \begin{array}{c} \underline{\quad} \ \boldsymbol{q}^1 \ \underline{\quad} \\ \vdots \\ \underline{\quad} \ \boldsymbol{q}^d \ \underline{\quad} \end{array} \right) \right)^\top \tag{11}$$

This reads as: $(K^\top V)_{\bullet j}$ contains the coefficients for the linear combination of the vector representations $\{\boldsymbol{q}^l\}_{l=1}^d$ of the bases stored in $Q$'s column space to form the output $\mathbf{z}$. Meanwhile, the $j$-th column $(K^\top V)_{\bullet j}$ of $K^\top V$ consists the inner product of $j$-th column of $V$ with every column of $K$.

$$\boldsymbol{z}^j = h \sum_{l=1}^d \boldsymbol{q}^l (K^\top V)_{lj}, \quad \text{where } (K^\top V)_{\bullet j} = \left( \boldsymbol{k}^1 \cdot \boldsymbol{v}^j, \boldsymbol{k}^2 \cdot \boldsymbol{v}^j, \cdots, \boldsymbol{k}^d \cdot \boldsymbol{v}^j \right)^\top. \tag{12}$$

As a result, using Assumption 4.2, and for simplicity the latent Hilbert spaces $\mathcal{Q}, \mathcal{K}, \mathcal{V}$ are assumed to be defined on the same spacial domain $\Omega$, i.e., $k_l(\cdot), v_j(\cdot)$ evaluated at every $x_i$ are simply their vector representations $\boldsymbol{k}^l$ ($1 \leq l \leq d$) and $\boldsymbol{v}^j$, we have the functions represented by the columns of the output $\mathbf{z}$ can be then compactly written as: rewriting $\langle v_j, k_l \rangle := (K^\top V)_{lj}$

$$z_j(x) := \sum_{l=1}^d \langle v_j, k_l \rangle \, q_l(x), \quad \text{for } j = 1, \cdots, d, \text{ and } x \in \{x_i\}_{i=1}^n, \tag{13}$$

where the bilinear form $\langle\cdot,\cdot\rangle : \mathcal{V} \times \mathcal{K} \to \mathbb{R}$. (13) can be also written in a componentwise form:

$$z_j(x_i) := (\boldsymbol{z}^j)_i = h \sum_{l=1}^{d} (\boldsymbol{k}^l \cdot \boldsymbol{v}^j)(\boldsymbol{q}^l)_i \approx \sum_{l=1}^{d} \left( \int_{\Omega} v_j(\xi) k_l(\xi) \, \mathrm{d}\xi \right) q_l(x_i). \tag{14}$$

Therefore, when $\{\diamond_j(\cdot)\}_{j=1}^{d}, \diamond \in \{q, k, v\}$ consist approximations to three sets of bases for potentially different subspaces, and if we set the trial spaces as the column spaces of $Q$ and the test space as that of $K$, respectively, the forward propagation of the Galerkin-type attention is a recast of a learnable Petrov–Galerkin-type projection (cf. Appendix D.1) for every basis represented by the columns of $V$. While the form of (14) suggests the orthonormality of the basis represented by $Q, K, V$, as well as being of full column ranks, the learnable nature of the method suggests otherwise (see Appendix D). At last, we have the following strikingly simple yet powerful approximation result.

**Theorem 4.3** (Céa-type lemma, simplified version). *Consider a Hilbert space $\mathcal{H}$ defined on a bounded domain $\Omega \subset \mathbb{R}^m$ discretized by $n$ grid points, and $f \in \mathcal{H}$. $\mathbf{y} \in \mathbb{R}^{n \times d}$ is the current latent representation for $n > d > m$ and full column rank. $\mathbb{Q}_h \subset \mathcal{Q} \subset \mathcal{H}$ and $\mathbb{V}_h \subset \mathcal{V} \subset \mathcal{H}$ are the latent approximation subspaces spanned by basis functions with the columns of $Q$ and $V$ in (3) as degrees of freedom, respectively, and $0 < \dim \mathbb{Q}_h = r \le \dim \mathbb{V}_h = d$. Let $\mathfrak{b}(\cdot,\cdot) : \mathcal{V} \times \mathcal{Q} \to \mathbb{R}$ be a continuous bilinear form, and if for any fixed $q \in \mathbb{Q}_h$ the functional norm of $\mathfrak{b}(\cdot, q)$ is bounded below by $c > 0$, then there exists a learnable map $g_\theta(\cdot)$ that is the composition of the Galerkin-type attention operator with an updated set of projection matrices $\{W^Q, W^K, W^V\}$, and a pointwise universal approximator, such that for $f_h \in \mathbb{Q}_h$ being the best approximation of $f$ in $\|\cdot\|_{\mathcal{H}}$ it holds:*

$$\|f - g_\theta(\mathbf{y})\|_{\mathcal{H}} \le c^{-1} \min_{q \in \mathbb{Q}_h} \max_{v \in \mathbb{V}_h} \frac{|\mathfrak{b}(v, f_h - q)|}{\|v\|_{\mathcal{H}}} + \|f - f_h\|_{\mathcal{H}}. \tag{15}$$

**Remarks on and interpretations of the best approximation result.** Theorem 4.3 states that the Galerkin-type attention has the architectural capacity to represent a quasi-optimal approximation in $\|\cdot\|_{\mathcal{H}}$ in the current subspace $\mathbb{Q}_h$. For the mathematically rigorous complete set of notations and the full details of the proof we refer the readers to Appendix D.3. Even though Theorem 4.3 is presented for a single instance of $f \in \mathcal{H}$ for simplicity, the proof shows that the attention operator is fully capable of simultaneously approximating a collection of functions (Appendix D.3.4).

Estimate (15) comes with great scalability with respect to the sequence length in that it all boils down to whether $c$ is independent of $n$ in the lower bound of $\|\mathfrak{b}(\cdot, q)\|_{\mathbb{V}_h'}$. The existence of an $n$-independent lower bound is commonly known as the discrete version of the Ladyzhenskaya–Babuška–Brezzi (LBB) condition [21, Chapter 6.12], also referred as the Banach-Nečas-Babuška (BNB) condition in Galerkin methods on Banach spaces [29, Theorem 2.6].

As the cornerstone of the approximation to many PDEs, the discrete LBB condition establishes the surjectivity of a map from $\mathbb{V}_h$ to $\mathbb{Q}_h$. In a simplified context (15) above of approximating functions using this linear attention variant ($Q$: values, query, $V$: keys), it roughly translates to: for an incoming "query" (function $f$ in a Hilbert space), to deliver its best approximator in "value" (trial function space), the "key" (test function space) has to be sufficiently rich such that there exists a key to unlock every possible value.

**Dynamic basis update.** Another perspective is to interpret the Galerkin-type dot-product attention (14) as a change of basis: essentially, the new set of basis is the column space of $Q$, and how to linearly combine the bases in $Q$ is based on the inner product (response) of the corresponding feature dimension's basis in $V$ against every basis in $K$. From this perspective ($Q$: values, $K$: keys, $V$: query), we have the following result of a layer-wise dynamical change of basis: through testing against the "keys", a latent representation is sought such that "query" (input trial space) and "values" (output trial space) can achieve the minimum possible difference under a functional norm; for details and the proof please refer to Appendix D.3.4.

**Theorem 4.4** (layer-wise dynamic basis update, simple version). *Under the same assumption as Theorem 4.3, it is further assumed that $\mathfrak{b}(\cdot, q)$ is bounded below on $\mathbb{K}_h \subset \mathcal{K} = \mathcal{V} \subset \mathcal{H}$ and $\mathfrak{a}(\cdot,\cdot) : \mathcal{V} \times \mathcal{K} \to \mathbb{R}$ is continuous. Then, there exists a set of projection matrices to update the value space $\{\tilde{q}_l(\cdot)\}_{l=1}^{d} \subset \mathbb{Q}_h = \mathrm{span}\{q_l(\cdot)\}_{l=1}^{d}$, for $z_j \in \mathbb{Q}_h$ ($j = 1, \cdots, d$) obtained through the basis update rule (14), it holds*

$$\big\| \mathfrak{a}(v_j, \cdot) - \mathfrak{b}(\cdot, z_j) \big\|_{\mathbb{K}_h'} \le \min_{q \in \mathbb{Q}_h} \max_{k \in \mathbb{K}_h} \frac{|\mathfrak{a}(v_j, k) - \mathfrak{b}(k, q)|}{\|k\|_{\mathcal{K}}}. \tag{16}$$

**The role of feed-forward networks and positional encodings in the dynamic basis update.** Due to the presence of the concatenated coordinates $\mathbf{x} := \|_{i=1}^n x_i \in \mathbb{R}^{n \times m}$ to the latent representation $\mathbf{y}$, the pointwise subnetwork $g_s(\cdot) : \mathbb{R}^{n \times m} \to \mathbb{R}^{n \times d}$ of the nonlinear universal approximator (FFN) in each attention block is one among many magics of the attention mechanism. In every attention layer, the basis functions in $\mathbb{Q}_h/\mathbb{K}_h/\mathbb{V}_h$ are being constantly enriched by $\mathrm{span}\{w_j \in \mathbb{X}_h : w_j(x_i) = (g_s(\mathbf{x}))_{ij}, 1 \le j \le d\} \subset \mathcal{H}$, thus being dynamically updated to try to capture how an operator of interest responses to the subset of inputs. Despite the fact that the FFNs, when being viewed as a class of functions, bear no linear structure within, the basis functions produced this way act as a building block to characterize a linear space for a learnable projection. This heuristic shows to be effective when the target is assumed to be a function of the (relative) positional encodings (coordinates, transformation groups, etc.), in that this is incorporated in many other attention-based learners with applications in physical sciences [82, 31, 44, 48].

## 5 Experiments

In this section we perform a numerical study the proposed Fourier Transformer (**FT**) with the Fourier-type encoder, and the Galerkin Transformer (**GT**) with the Galerkin-type encoder, in various PDE-related operator learning tasks. The models we compare our newly proposed models with are the operator learners with the simple attention replaced by the standard softmax normalized scaled dot-product attention (**ST**) [88], and a linear variant (**LT**) [76] in which two independent softmax normalizations are applied on $Q, K$ separately.[2] The data are obtained courtesy of the PDE benchmark under the MIT license.[3] For full details of the training/evaluation and model structures please refer to Appendix C.

Instead of the standard Xavier uniform initialization [34], inspired by the interpretations of Theorem 4.3 in Appendix D.3.4, we modify the initialization for the projection matrices slightly as follows

$$W_{\mathrm{init}}^{\diamond} \leftarrow \eta U + \delta I, \quad \text{for } \diamond \in \{Q, K, V\}, \tag{17}$$

where $U = (x_{ij})$ is a random matrix using the Xavier initialization with gain 1 such that $x_{ij} \sim \mathcal{U}([-\sqrt{3/d}, \sqrt{3/d}])$, and $\delta$ is a small positive number. In certain operator learning tasks, we found that this tiny modification boosts the evaluation performance of models by up to $50\%$ (see Appendix C.2) and improves the training stability acting as a cheap remedy to the lack of a softmax normalization. We note that similar tricks have been discovered concurrently in [23].

Unsurprisingly, when compared the memory usage and the speed of the networks (Table 1), the Fourier-type attention features a 40%–50% reduction in memory versus the attention with a softmax normalization. The Galerkin attention-based models have a similar memory profile with the standard linear attention, it offers up to a 120% speed boost over the linear attention in certain tests.

Table 1: The memory usage/FLOP/complexity comparison of the models. Batch size: 4; the CUDA mem (GB): the sum of the `self_cuda_memory_usage`; GFLOP: Giga FLOP for 1 backpropagation (BP); both are from the PyTorch `autograd` profiler for 1 BP averaging from 1000 BPs; the mem (GB) is recorded from `nvidia-smi` of the memory allocated for the active Python process during profiling; the speed (iteration per second) is measured during training; the exponential operation is assumed to have an explicit complexity of $c_e > 1$ [14].

|  | Example 1: $n = 8192$ | | | | Encoders only: $n = 8192, d = 128, l = 10$ | | | | Computational complexity of the dot-product per layer |
|---|---|---|---|---|---|---|---|---|---|
|  | Mem | CUDA Mem | Speed | GFLOP | Mem | CUDA Mem | Speed | GFLOP | |
| ST | 18.39 | 31.06 | 5.02 | 1393 | 18.53 | 31.34 | 4.12 | 1876 | $O(n^2 c_e d)$ |
| FT | 10.05 | 22.92 | 6.10 | 1138 | 10.80 | 22.32 | 5.46 | 1610 | $O(n^2 d)$ |
| LT | 2.55 | 2.31 | 12.70 | 606 | 2.73 | 2.66 | 10.98 | 773 | $O(n(d^2 + c_e d))$ |
| **GT** | **2.36** | **1.93** | **27.15** | **275** | **2.53** | **2.33** | **19.20** | **412** | $O(nd^2)$ |

The baseline models for each example are the best operator learner to-date, the state-of-the-art Fourier Neural Operator (FNO) in [57] but without the original built-in batch normalization. All attention-based models match the parameter quota of the baseline, and are trained using the loss in

---

[2] https://github.com/lucidrains/linear-attention-transformer
[3] https://github.com/zongyi-li/fourier_neural_operator

(2) with the same `1cycle` scheduler [78] for 100 epochs. For fairness, we have also included the results for the standard softmax normalized models (ST and LT) using the new layer normalization scheme in (5) and (6). We have retrained the baseline with the same `1cycle` scheduler using the code provided in [57], and listed the original baseline results using a step scheduler of 500 epochs of training from [57] Example 5.1 and Example 5.2, respectively.

## 5.1 Example 1: viscous Burgers' equation

In this example, we consider a benchmark problem of the viscous Burgers' equation with a periodic boundary condition on $\Omega := (0, 1)$ in [57]. The nonlinear operator to be learned is the discrete approximations to the solution operator $T : C_p^0(\Omega) \cap L^2(\Omega) \to C_p^0(\Omega) \cap H^1(\Omega)$, $u_0(\cdot) \mapsto u(\cdot, 1)$. The initial condition $u_0(\cdot)$'s are sampled following a Gaussian Random Field (GRF).

The result can be found in Table 2a. All attention-based operator learners achieve a resolution-invariant performance similar with FNO1d in [57]. The new Galerkin projection-type layer normalization scheme significantly outperforms the regular layer normalization rule in this example, in which both inputs and targets are unnormalized. For full details please refer to Appendix C.2.

## 5.2 Example 2: Darcy flow

In this example, we consider another well-known benchmark $-\nabla \cdot (a\nabla u) = f$ for $u \in H_0^1(\Omega)$ from [10, 57, 56, 64], and the operator to be learned is the approximations to $T : L^\infty(\Omega) \to H_0^1(\Omega), a \mapsto u$, in which $a$ is the coefficient with a random interface geometry, and $u$ is the weak solution. Here $L^\infty(\Omega)$ is a Banach space and cannot be compactly embedded in $L^2(\Omega)$ (a Hilbert space), we choose to avoid this technicality as the finite dimensional approximation space can be embedded in $L^2(\Omega)$ given that $\Omega$ is compact.

The result can be found in Table 2b. As the input/output are normalized, in contrast to Example 5.1, the Galerkin projection-type layer normalization scheme does not significantly outperform the regular layer normalization rule in this example. The attention-based operator learners achieve on average 30% to 50% better evaluation results than the baseline FNO2d (only on the fine grid) using the same trainer. For full details please refer to Appendix C.3.

Table 2: (a) Evaluation relative error ($\times 10^{-3}$) of Burgers' equation 5.1. (b) Evaluation relative error ($\times 10^{-2}$) of Darcy interface problem 5.2.

|  | (a) | | | | (b) | |
|---|---|---|---|---|---|---|
|  | $n = 512$ | $n = 2048$ | $n = 8192$ |  | $n_f, n_c = 141, 43$ | $n_f, n_c = 211, 61$ |
| FNO1d [57] | 15.8 | 14.6 | 13.9 | FNO2d [57] | 1.09 | 1.09 |
| FNO1d `1cycle` | 4.373 | 4.126 | 4.151 | FNO2d `1cycle` | 1.419 | 1.424 |
| FT regular Ln | 1.400 | 1.477 | 1.172 | FT regular Ln | **0.838** | **0.847** |
| GT regular Ln | 2.181 | 1.512 | 2.747 | GT regular Ln | 0.894 | 0.856 |
| ST regular Ln | 1.927 | 2.307 | 1.981 | ST regular Ln | 1.075 | 1.131 |
| LT regular Ln | 1.813 | 1.770 | 1.617 | LT regular Ln | 1.024 | 1.130 |
| FT Ln on $Q, K$ | **1.135** | **1.123** | **1.071** | FT Ln on $Q, K$ | 0.873 | 0.921 |
| GT Ln on $K, V$ | **1.203** | **1.150** | **1.025** | GT Ln on $K, V$ | **0.839** | **0.844** |
| ST Ln on $Q, K$ | 1.271 | 1.266 | 1.330 | ST Ln on $Q, K$ | 0.946 | 0.959 |
| LT Ln on $K, V$ | 1.139 | 1.149 | 1.221 | LT Ln on $K, V$ | 0.875 | 0.970 |

## 5.3 Example 3: inverse coefficient identification for Darcy flow

In this example, we consider an inverse coefficient identification problem based on the same data used in Example 5.2. The input (solution) and the target (coefficient) are reversed from Example 5.2, and the noises are added to the input. The inverse problems in practice are a class of important tasks in many scientific disciplines such as geological sciences and medical imaging but much more difficult due to poor stability [51]. We aim to learn an approximation to an ill-posed operator $T : H_0^1(\Omega) \to L^\infty(\Omega), u + \epsilon N_\nu(u) \mapsto a$, where $N_\nu(u)$ stands for noises related to the sampling distribution and the data. $\epsilon = 0.01$ means 1% of noise added in both training and evaluation, etc.

The result can be found in Table 3. It is not surprising that FNO2d, an excellent smoother which filters higher modes in the frequency domain, struggles in this example to recover targets consisting of high-frequency traits (irregular interfaces) from low-frequency prevailing data (smooth solution due to ellipticity). We note that, the current state-of-the-art methods [16] for inverse interface coefficient identification need to carry numerous iterations to recover a single instance of a simple coefficient with a regular interface, provided that a satisfactory denoising has done beforehand. The attention-based operator learner has capacity to unearth structurally how this inverse operator's responses on a subset, with various benefits articulated in [56, 57, 5, 64, 10].

Table 3: Evaluation relative error ($\times 10^{-2}$) of the inverse problem 5.3.

| | $n_f, n_c = 141, 36$ | | | $n_f, n_c = 211, 71$ | | |
|---|---|---|---|---|---|---|
| | $\epsilon = 0$ | $\epsilon = 0.01$ | $\epsilon = 0.1$ | $\epsilon = 0$ | $\epsilon = 0.01$ | $\epsilon = 0.1$ |
| FNO2d (only $n_f$) | 13.71 | 13.78 | 15.12 | 13.93 | 13.96 | 15.04 |
| FNO2d (only $n_c$) | 14.17 | 14.31 | 17.30 | 13.60 | 13.69 | 16.04 |
| FT regular Ln | **1.799** | **2.467** | 6.814 | 1.563 | 2.704 | 8.110 |
| GT regular Ln | 2.026 | 2.536 | **6.659** | 1.732 | 2.775 | 8.024 |
| ST regular Ln | 2.434 | 3.106 | 7.431 | 2.069 | 3.365 | 8.918 |
| LT regular Ln | 2.254 | 3.194 | 9.056 | 2.063 | 3.544 | 9.874 |
| FT Ln on $Q, K$ | 1.921 | 2.717 | 6.725 | **1.523** | **2.691** | 8.286 |
| GT Ln on $K, V$ | 1.944 | 2.552 | 6.689 | 1.651 | 2.729 | **7.903** |
| ST Ln on $Q, K$ | 2.160 | 2.807 | 6.995 | 1.889 | 3.123 | 8.788 |
| LT Ln on $K, V$ | 2.360 | 3.196 | 8.656 | 2.136 | 3.539 | 9.622 |

## 6 Conclusion

We propose a general operator learner based on a simple attention mechanism. The network is versatile and is able to approximate both the PDE solution operator and the inverse coefficient identification operator. The evaluation accuracy on the benchmark problems surpasses the current best state-of-the-art operator learner Fourier Neural Operator (FNO) in [57]. However, we acknowledge the limitation of this work: (i) similar to other operator learners, the subspace, on which we aim to learn the operator's responses, may be infinite dimensional, but the operator must exhibit certain low-dimensional attributes (e.g., smoothing property of the higher frequencies in GRF); (ii) it is not efficient for the attention operator to be applied at the full resolution for a 2D problem, and this limits the approximation to a nonsmooth subset such as functions in $L^\infty$; (iii) due to the order of the matrix product, the proposed linear variant of the scaled dot-product attention is non-causal thus can only apply to encoder-only applications.

## 7 Broader Impact

Our work introduces the state-of-the-art self-attention mechanism the first time to PDE-related operator learning problems. The new interpretations of attentions invite numerical analysts to work on a more complete and delicate approximation theory of the attention mechanism. We have proved the Galerkin-type attention's approximation capacity in an ideal Hilbertian setting. Numerically, the new attention-based operator learner has capacity to approximate the difficult inverse coefficient identification problem with an extremely noisy measurements, which was not attainable using traditional iterative methods for nonlinear mappings. Thus, our method may pose a huge positive impact in geoscience, medical imaging, etc. Moreover, traditionally the embeddings in Transformer-based NLP models map the words to a high dimensional space, but the topological structure in the same feature dimension between different positions are learned thereby not efficient. Our proof provides a theoretical guide for the search of feature maps that preserve, or even create, structures such as differentiability or physical invariance. Thus, it may contribute to the removal of the softmax nonlinearity to speed up significantly the arduous training or pre-training of larger encoder-only models such as BERT [27], etc. However, we do acknowledge that our research may negatively impact on the effort of building a cleaner future for our planet, as inverse problems are widely studied in reservoir detection, and we have demonstrated that the attention-based operator learner could potentially help to discover new fossil fuel reservoirs due to its capacity to infer the coefficients from noisy measurements.

## Acknowledgments and Disclosure of Funding

The hardware to perform this work is kindly donated by Andromeda Saving Fund. The first author was supported in part by the National Science Foundation under grants DMS-1913080 and DMS-2136075. No additional revenues are related to this work. We would like to thank the anonymous reviewers and the area chair for the suggestions on improving this article. We would like to thank Dr. Long Chen (Univ of California Irvine) for the inspiration of and encouragement on the initial conceiving of this paper, as well as numerous constructive advices on revising this paper, not mentioning his persistent dedication of making publicly available tutorials [17] on writing beautiful vectorized code. [4] We would like to thank Dr. Ari Stern (Washington Univ in St. Louis) for the help on the relocation during the COVID-19 pandemic. We would like to thank Dr. Likai Chen (Washington Univ in St. Louis) for the invitation to the Stats and Data Sci seminar at WashU that resulted the reboot of this study. [5] We would like to thank Dr. Ruchi Guo (Univ of California Irvine) and Dr. Yuanzhe Xi (Emory Univ) for the invaluable feedbacks on the choice of the numerical experiments. We would like to thank the Kaggle community, including but not limited to Jean-François Puget (CPMP@Kaggle) for sharing a simple Graph Transformer in TensorFlow,[6] Murakami Akira (mrkmakr@Kaggle) for sharing a Graph Transformer with a CNN feature extractor in Tensorflow, [7] and Cher Keng Heng (hengck23@Kaggle) for sharing a Graph Transformer in PyTorch.[8] We would like to thank daslab@Stanford, OpenVaccine, and Eterna for hosting the COVID-19 mRNA Vaccine competition and Deng Lab (Univ of Georgia) for collaborating in this competition. We would like to thank CHAMPS (Chemistry and Mathematics in Phase Space) for hosting the $J$-coupling quantum chemistry competition and Corey Levinson (Eligo Energy, LLC) for collaborating in this competition. We would like to thank Zongyi Li (Caltech) for sharing some early dev code in the updated PyTorch `fft` interface and the comments on the viscosity of the Burgers' equation. We would like to thank Ziteng Pang (Univ of Michigan) and Tianyang Lin (Fudan Univ) to update us with various references on Transformers. We would like to thank Joel Schlosser (Facebook) to incorporate our change to the PyTorch `transformer` module to simplify our testing pipeline. We would be grateful to the PyTorch community for selflessly code sharing, including Phil Wang(lucidrains@github) and Harvard NLP group [52]. We would like to thank the `chebfun` [28] for integrating powerful tools into a simple interface to solve PDEs. We would like to thank Dr. Yannic Kilcher (ykilcher@twitter) and Dr. Hung-yi Lee (National Taiwan Univ) for frequently covering the newest research on Transformers in video formats. We would also like to thank the Python community [87, 68] for sharing and developing the tools that enabled this work, including PyTorch [69], NumPy [39], SciPy [89], Plotly [45] Seaborn [93], Matplotlib [43], and the Python team for Visual Studio Code. We would like to thank `draw.io` [46] for providing an easy and powerful interface for producing vector format diagrams. For details please refer to the documents of every function that is not built from the ground up in our open-source software library.[9]

---

[4] https://github.com/lyc102/ifem
[5] Transformer: A Dissection from an Amateur Applied Mathematician
[6] https://www.kaggle.com/cpmpml/graph-transfomer
[7] https://www.kaggle.com/mrkmakr/covid-ae-pretrain-gnn-attn-cnn
[8] https://www.kaggle.com/c/stanford-covid-vaccine/discussion/183518
[9] https://github.com/scaomath/galerkin-transformer

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
