# Appendices of Choose a Transformer: Fourier or Galerkin

## A  Table of notations

Table 4: Notations used in an approximate chronological order and their meaning in this work.

| Notation | Meaning |
|---|---|
| $\mathcal{H}, \mathcal{X}, \mathcal{Y}$ | Hilbert spaces defined on a domain $\Omega$, $f \in \mathcal{H} : \Omega \to \mathbb{R}$ |
| $\mathcal{Q}, \mathcal{K}, \mathcal{V}$ | Latent representation Hilbert spaces, e.g., $v \in \mathcal{V} : \Omega^* \to \mathbb{R}$ |
| $(\mathcal{H}, \langle \cdot, \cdot \rangle)$ | $\mathcal{H}$ and its inner-product structure, $\langle u, v \rangle := \int_\Omega u(x)v(x)dx$ for simplicity |
| $\mathcal{H}'$ | the space of the bounded linear functionals defined on a Hilbert space |
| $\|v\|_{\mathcal{H}}$ | The norm defined by the inner product $\|v\|_{\mathcal{H}} := \langle v, v \rangle^{1/2}$ |
| $\|f_u\|_{\mathcal{H}'}$ | The natural induced norm of $f_u(\cdot) := \langle u, \cdot \rangle$, $\|f_u\|_{\mathcal{H}'} := \sup_{v \in \mathcal{H}} |f_u(v)|/\|v\|_{\mathcal{H}}$ |
| $\|\boldsymbol{v}\|$ | The $\ell^2$-norm defined by the inner product $\|\boldsymbol{v}\| := (\boldsymbol{v} \cdot \boldsymbol{v})^{1/2}$ for $\boldsymbol{v} \in \mathbb{R}^d$ |
| $\mathfrak{b}(\cdot, \cdot)$ | A bilinear form, having two inputs from potentially different subspaces |
| $L^p(\Omega)$ | The space of functions with integrable $p$-th moments in $\Omega$ |
| $L^\infty(\Omega)$ | The space of functions with a bounded essential supremum in $\Omega$ |
| $H^1(\Omega)$ | Sobolev space $W^{1,2}(\Omega) := \{\phi \in L^2(\Omega) : D\phi \in L^2(\Omega)\}$ |
| $H_0^1(\Omega)$ | $\{v \in H^1(\Omega) : \Upsilon(u) = 0 \text{ on } \partial\Omega\}$, where $\Upsilon(\cdot)$ is the trace operator |
| $\boldsymbol{H}(\mathrm{div}; \Omega)$ | Hilbert space with a graph norm $\{\phi \in \boldsymbol{L}^2(\Omega) : \mathrm{div}\,\phi \in L^2(\Omega)\}$ |
| $C_p^0(\Omega) \simeq C^0(\mathbb{S}^1)$ | The space of continuous functions with a periodic boundary condition |
| $\|u\|_{L^2(\Omega)}$ | The $L^2$-norm of $u$, $\|u\|_{L^2(\Omega)}^2 := \int_\Omega |u|^2 \,\mathrm{d}x$ |
| $|u|_{H^1(\Omega)}$ | The $H^1$-seminorm of $u$, $|u|_{H^1(\Omega)}^2 := \int_\Omega |Du|^2 \,\mathrm{d}x$ |
| $x \in \Omega \subset \mathbb{R}^m$ | A point in the spacial domain $\Omega$ of interest |
| $m$ | The dimension of the underlying spacial domain in $\mathbb{R}^m$ |
| $d$ | The dimension of the latent representation approximation subspace |
| $L_a(u) = f$ | The operator form (strong form) of a PDE with coefficient $a$ |
| $u(\cdot)$ | The solution to the weak form $\langle Lu, v \rangle = \langle f, v \rangle$ for any test function $v \in \mathcal{H}$ |
| $a(\cdot)$ | The coefficients in a PDE operator |
| $\partial_t u + N(u) = 0$ | A time-dependent stiff PDE, where $N(\cdot)$ is nonlinear differential operator |
| $h$ | The mesh size of a uniform grid |
| $n \approx 1/h^m$ | The discretization size (sequence length) of data, $O(n^m)$ for an $\mathbb{R}^m$ problem |
| $n_f, n_c$ | The fine grid size, the coarse grid size |
| $\mathbb{X}_h, \mathbb{Y}_h, \mathbb{A}_h$ | The discrete function space with degrees of freedom on grid points of mesh size $h$ |
| $\mathbb{Q}_h, \mathbb{V}_h$ | Certain subspaces spanned by functions in $\mathbb{X}_h, \mathbb{Y}_h$ |
| $u_h, a_h$ | The approximation to $u, a$ whose degrees of freedom defined at the grid points |
| $T$ | The operator to be learned related to a partial differential equation |
| $T_h$ | The approximation to $T$ applied on functions on a discrete grid with mesh size $h$ |
| $\mathbf{y}, \mathbf{z}$ | the input of and the output from the attention operator, in $\mathbb{R}^{n \times d}$ |
| $\boldsymbol{q}_i, \boldsymbol{k}_i, \boldsymbol{v}_i$ | the $i$-th row of, or the $i$-th position's feature vector in a latent representation |
| $\boldsymbol{z}^i, \boldsymbol{q}^i, \boldsymbol{k}^i, \boldsymbol{v}^i$ | the $i$-th column of, or the $i$-th basis's discrete DoFs in a latent representation |
| $A_{i\bullet} / A_{\bullet j}$ | the $i$-th row/$j$-th column of a matrix $A$ |
| $\{y_j(\cdot)\}_{j=1}^d$ | A set of latent basis whose DoFs form the column space of $Y \in \mathbb{R}^{n \times d}$ |
| $\{\chi_{y_j}(\cdot)\}_{j=1}^d$ | The set of degrees of freedom associated with the set of bases $\{y_j(\cdot)\}_{j=1}^d$ |
| $(\boldsymbol{v})_i$ | the $i$-th entry/row of a vector $\boldsymbol{v}$ |
| $I_h$ | The nodal interpolation operator such that $(I_h v)(x_i) = v(x_i)$ |
| $\Pi_h$ | The interpolation or projection operator that maps function to a grid with mesh size $h$ |
| $\mathcal{H} \hookrightarrow C^0(\Omega)$ | $\mathcal{H}$ is continuously embedded in the space of continuous functions |

## B  Network structures

The network in Figure 4 is used in Example 5.1. The model used in the forward Darcy problem 5.2 is in Figure 5. A detailed comparison can be found in Table 5.

Table 5: The detailed comparison of networks; SC: spectral convolution layer; a `torch.cfloat` type parameter entry counts as two parameters.

| | Encoder | | | | Decoder | | | # params |
|---|---|---|---|---|---|---|---|---|
| | layers | `dmodel` | `nhead` | # SC | `dmodel` | modes | activation | |
| FNO 1D | 0 | N/A | N/A | 4 | 64 | 16 | ReLU | 550k |
| FT/GT in 5.1 | 4 | 96 | 1 | 2 | 48 | 16 | SiLU | 523k–530k |
| FNO 2D | 0 | N/A | N/A | 4 | $32 \times 32$ | 12 | ReLU | 2.37m |
| FT/GT in 5.2 | 6 | 128 | 4 | 2 | $32 \times 32$ | 12 | SiLU | 2.22m |
| FT/GT in 5.3 | 6 | 192 | 4 | 0 | N/A | N/A | SiLU | 2.38m |

When the target is smooth, the spectral convolution layer from [57] is used in our network as a smoother (decoder), and the original ReLU activation is replaced by the Sigmoid Linear Unit (SiLU) [72]. We have removed the batch normalization (BN) from the original spectral convolution layers as well. Whenever the input or the output of certain layers of the network is approximating a non-smooth function a priori, the activation are changed from SiLU to ReLU.

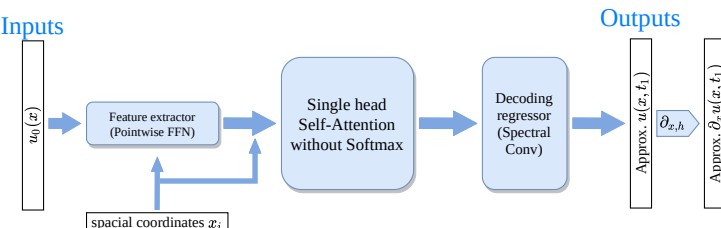

Figure 4: A simple attention-based operator learner in Example 5.1.

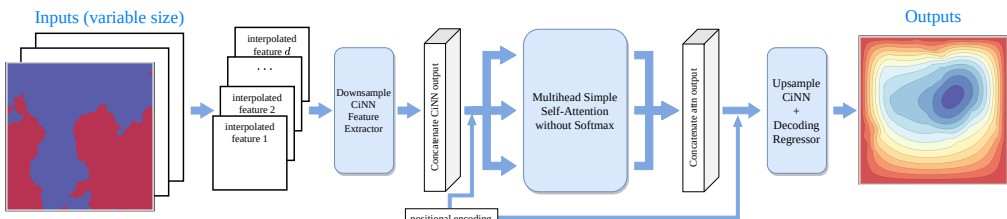

Figure 5: An attention-based operator learner on $\Omega \subset \mathbb{R}^2$.

**Downsampling CNN.** The downsampling interpolation-based CNN (CiNN) is to reduce the size of the input to a computationally feasible extent for attention-based encoder layers, in addition to a channel expansion to match the hidden feature dimension (channels). The full resolution input function sampled at a fine grid of size $n_f \times n_f$ is downsampled by CiNN to a collection of latent functions on an $n_c \times n_c$ coarse grid. Then, the coarse grid representations are concatenated with the positional encoding (Euclidean coordinates on the coarse grid) to be sent to the attention-based encoder layers.

The structures of the downsampling CiNN can be found in Figure 6. In CiNN, instead of pooling, the downsampling are performed through a bilinear interpolation with nonmatching coarse/fine grids. The convolution block adopts a simplified variant from the basic block in [41]. The convolution layer is applied only once before the skip-connection, and the batch normalization is removed from the block.

In the downsampling CiNN, the first convolution layer maps the input data to a tensor of which the number of channels matches the number of hidden dimension of the attention layers. Then, the full resolution representations in all channels are interpolated from the $n_f \times n_f$ grid to an $n_m \times n_m$ grid of an intermediate size between $n_f$ and $n_c$, and $n_m \approx \sqrt{n_f n_c}$. Next, another three convolution layers are applied consecutively together with their outputs stacked in the channel dimension. Finally, this

stacked tensor is downsampled again by another bilinear interpolation as the output the downsampling CiNN.

The coarse grid positional encoding of size $n_c \times n_c \times 2$ is concatenated to the output latent representations before flattening and the scaled dot-product attention. As a result, the input/output dimensions of an attention-based encoder layer are both $n_c^2 \times d$. Nevertheless, inside an encoder layer, the propagation runs for a tensor of size $n_c^2 \times (d + m \cdot (\texttt{nhead}))$.

**Upsampling CNN.** The output from the attention-based encoder layers is first reshaped from an $n_c^2 \times d$ matrix to an $n_c \times n_c \times d$ tensor, then upsampled by another CiNN to the full resolution. The upsampling CiNN has a simpler structure (Figure 7) than the downsampling CiNN. We have two interpolations that map tensors of $n_c \times n_c \times d$ to $n_m \times n_m \times d$, and $n_m \times n_m \times d$ to $n_f \times n_f \times d$, respectively. A simple convolution layer with matching number of channels is between them. The positional encoding of the fine grid is then concatenated with the output from the upsample layer, and sent to the decoder layers.

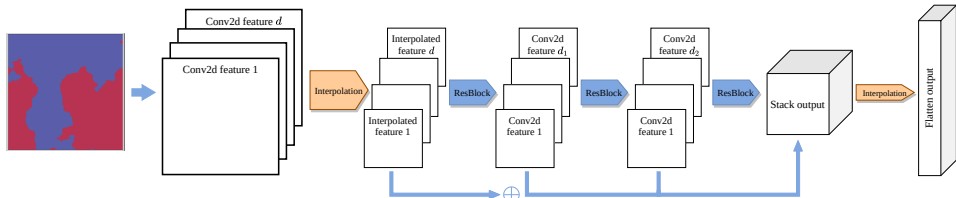

Figure 6: A 2D bilinear interpolation-based CNN (CiNN) for downsampling.

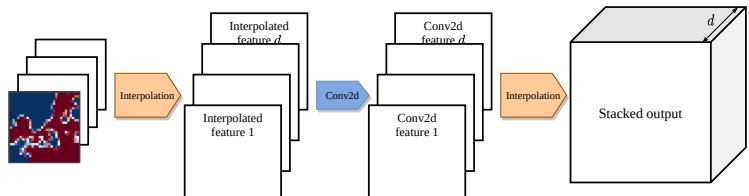

Figure 7: A 2D bilinear interpolation-based CNN (CiNN) for upsampling.

## C  Supplemental details of the experiments

### C.1  Training and evaluation setup

In training our models, we opt for a standard `1cycle` [78] learning rate strategy with a warm-up phase for an environmental responsible and a seed-invariant training. We run a mini-batch ADAM iterations for a total number of 100 epochs (12800 iterations with batch size 8). The learning rate starts and ends with $10^{-4} \cdot lr_{\max}$, and reaches the maximum of $lr_{\max}$ at the end of the 30-th epoch. The $lr_{\max} = 10^{-3}$ for all models except being $5 \times 10^{-4}$ for ST and FT in 2D problems.

The batch size is set to 8 in 1D in $n = 512, 2048$, and 4 in $n = 8192$ as well as in 2D examples. The training cost of our models are reported in Table 6. It is not surprising that attention-based operator learners can be more efficiently trained than traditional MLP-based operator learners. Our model can be trained using merely a fraction of time versus MLP-based operator learners (cf. Burgers' equation training time in [91, Appendix C]), thus proven to be a more environment-friendly data-driven model.

For the three examples prepared, there are 1024 samples in the training set, and 100 in the testing set. Even though the initial conditions or the coefficients for different samples, in Example 5.1 and Example 5.2 respectively, follow the same distribution constructed based on GRF, there are no repetitions between the functions in the training set and those in the testing set.

During training, there is no regularization applied for the weights of the models. A simple gradient clip of 1 is applied. When the target function is known a priori being smooth with an $H^{1+\alpha}$ regularity ($\alpha > 0$), we employ an $H^1$-seminorm regularization between the 2nd order approximation (the central

Table 6: Environmental impact measured in computational cost of training (in hours).

| | Example 1 | | | Example 2 | | Example 3 | |
|---|---|---|---|---|---|---|---|
| | $n = 512$ | $n = 2048$ | $n = 8192$ | $n_f = 141$ | $n_f = 211$ | $n_f = 141$ | $n_f = 211$ |
| FT | 0.063 | 0.138 | 1.217 | 0.615 | 1.553 | 0.452 | 2.638 |
| GT | 0.064 | 0.079 | 0.245 | 0.367 | 0.610 | 0.248 | 0.857 |

Table 7: The dropout comparison during training.

| | attention | FFN | downsample | upsample | decoder |
|---|---|---|---|---|---|
| FT both Lns in 5.1 | 0.0 | 0.05 | N/A | N/A | 0.0 |
| GT new Ln in 5.1 | 0.0 | 0.0 | N/A | N/A | 0.0 |
| GT reg Ln in 5.1 | 0.1 | 0.1 | N/A | N/A | 0.0 |
| FT in 5.2 | 0.1 | 0.1 | 0.05 | 0.0 | 0.0 |
| GT in 5.2 | 0.1 | 0.05 | 0.05 | 0.0 | 0.0 |
| FT/GT in 5.3 | 0.05 | 0.05 | 0.05 | N/A | 0.05 |

difference in 1D, and the 5-point stencil in 2D) to derivatives of the targets and those of the outputs from the model (see Section 3). We choose $\gamma = 0.1h$ in Example 5.1 and $\gamma = 0.5h$ in Example 5.2. The dropouts during training are obtained through a simple grid search in $\{0.0, 0.05, 0.1\}$ and can be found in Table 7.

Throughout all the numerical experiments, the result is obtained from setting 1127802 as the random number generator seed, and the PyTorch cuDNN backend to deterministic. Error bands are reported using 10 different seeds (see Figure 8). All benchmarks are run on a single RTX 3090. Due to the nature of our problem and the data pairs, there is no randomness between the input and the output for a single instance in Example 5.1 and Example 5.2, the only stochastic components are the sampling for optimizations and the initializations of the model, and the impact due to the choice of seeds for our models is empirically minimal.

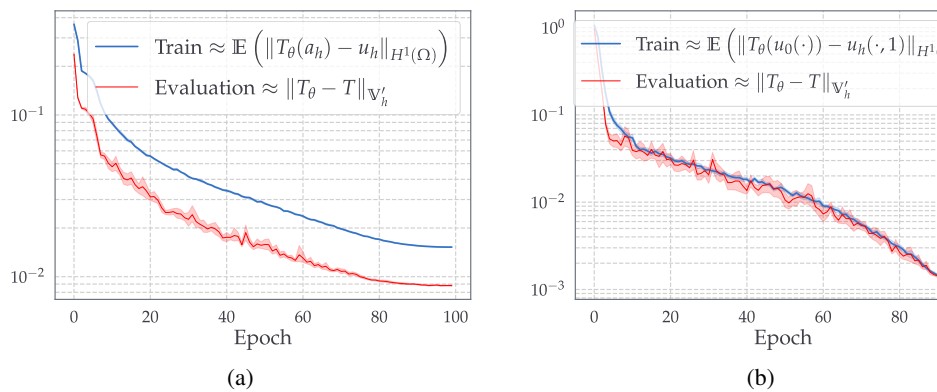

(a)                    (b)

Figure 8: Typical training and evaluation convergences of the Galerkin Transformer. (a) example 5.2: 10 different seeds for the model initialization; (b) example 5.1: 10 different seeds in both model initializations and the train loader; the discrete norm of a nonlinear operator $\|N\|_{\mathbb{V}'_h} :=$ $\sup_{v \in \mathbb{V}_h} \|N(v)\|_{\mathcal{H}} / \|v\|_{\mathcal{H}}$ is defined similarly to that of a linear operator. The $H^1$-seminorm part in the $H^1$-norm shown in figures is weighted by $\gamma$ from Section 3.

## C.2   Experimental details for Example 5.1

**Data preparation.**   The following problem is considered in Example 5.1:

$$\begin{cases} \partial_t u + u \partial_x u = \nu \partial_{xx} u & \text{for } (x,t) \in (0,1) \times (0,1], \\ u(x,0) = u_0(x) \text{ for } x \in (0,1), \end{cases} \tag{18}$$

and it is assumed that $u_0 \in C_p^0(\Omega) \cap L^2(\Omega)$. The operator to be learned is:

$$T : C_p^0(\Omega) \cap L^2(\Omega) \to C_p^0(\Omega) \cap H^1(\Omega), \quad u_0(\cdot) \mapsto u(\cdot, 1).$$

Following [57], the initial data are prepared using a Gaussian Random Field (GRF) simulation $\sim \mathcal{N}(0, 25^2(-\Delta + 25I)^{-2})$, and the system is solved using the `chebfun` package [28] with a spectral method using a very fine time step $\delta t \ll 1$ for the viscosity $\nu = (2\pi)^{-1}0.1$ on a grid with $n = 8192$. Solutions and initial conditions in other two resolutions ($n = 512, \ 2048$) are downsampled from this finest grid. Therefore, the discrete approximation the model learns is: for $h \in \{2^{-9}, 2^{-11}, 2^{-13}\}$

$$T_h : \mathbb{X}_h \to \mathbb{X}_h, \quad I_h u_0(\cdot) \mapsto \Pi_h u_{h_*}(\cdot, 1),$$

where $\mathbb{X}_h \simeq \mathbb{R}^n$ denotes a function space of which the degrees of freedom are defined on the grid with mesh size $h$, such as the space of piecewise linear functions. $I_h(\cdot)$ denotes the nodal value interpolation, and $\Pi_h(\cdot)$ denotes the downsampling operator from the space on the finest grid with mesh size $h_* = 2^{-13}$ to $\mathbb{X}_h$.

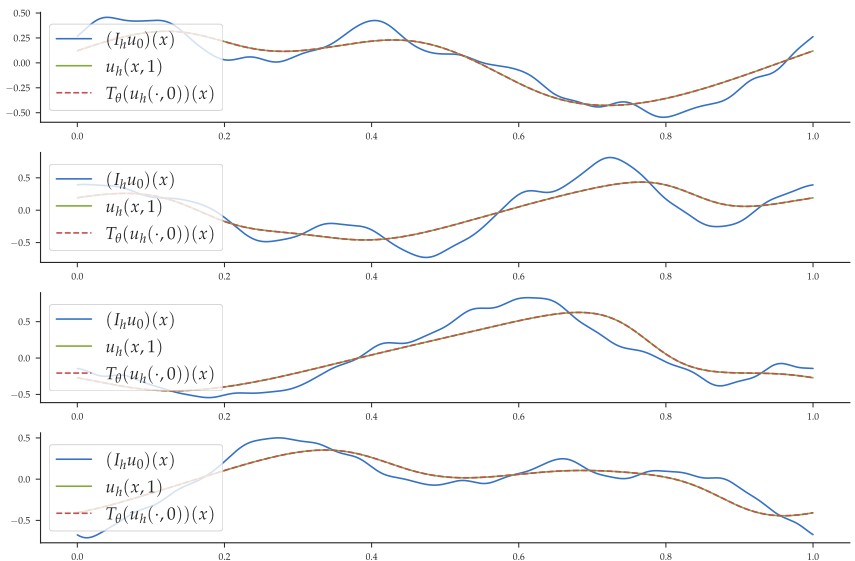

Figure 9: Evaluation results for 4 randomly chosen samples in the test set; the average relative error $= 1.079 \times 10^{-3}$.

**Effect of the diagonal initialization.** When using the regular layer normalization rule (7) to train the GT models (which is equivalent to the efficient attention proposed in [76]), it fails to converge under the `1cycle` learning rate scheduling with the $0.05$ dropout in FFN and $0.0$ dropout for the attention weights. We observe that the evaluation metric peaks after the warmup phase ($\approx 2 \times 10^{-2}$ around epoch 30). The GT using the new rule (6) is convergent almost unconditionally under the same initialization, which reaches the common $\approx 1 \times 10^{-3}$ range if the diagonal initialization is employed (e.g., see Figure 8b). The result reported in Table 2a for GT with the regular layer normalization is obtained through imposing a $0.1$ dropout for the attention weights. A more detailed comparison can be found in Table 8. The training becomes divergent for a certain model if the best epoch shown in the table is not around 100. We conjecture that the success of the diagonal initialization is due to the input (initial conditions, blue curves in Figure 9) being highly correlated spacially with the target (solutions at $t = 1$, green curves in Figure 9), despite the highly nonlinear mapping. Thus, in the encoder layers, what the attention operator learned is likely to be a perturbation of the identity operator in the latent Hilbert space, if a suitable basis can be found for $\mathbb{V}_h$ (or $\mathbb{Q}_h$ in the linear variant).

**Effect of the Galerkin projection-type layer normalization scheme.** Multiplying $u$ on both sides of (18), the energy law of the Burgers' equation can be obtained through an integration by parts on $\Omega := (0, 1)$ with the periodic boundary condition:

$$\langle \partial_t u, u \rangle + \langle \partial_x(u^2), u \rangle / 2 = \nu \langle \partial_{xx} u, u \rangle \implies \mathrm{d}\left(\|u\|_{L^2(\Omega)}^2\right) / \mathrm{d}t = -\nu \|\partial_x u\|_{L^2(\Omega)}^2. \qquad (19)$$

Table 8: Ablation study: evaluation relative error ($\times 10^{-3}$) of the Galerkin Transformer with the regular layer normalization (7) and the new one (6), using various types of initialization; the baseline model is the GT with the default Xavier uniform initialization. $GT_R$: layer normalization (7); $GT_A$: layer normalization (6); $\delta$: the weight added to the diagonal; $\eta$: the gain of Xavier uniform initialization; $\zeta$: dropout for the attention weights.

| | $n = 512$ | | $n = 2048$ | | $n = 8192\ (b = 4)$ | |
|---|---|---|---|---|---|---|
| | Rel. err | Best ep. | Rel. err | Best ep. | Rel. err | Best ep. |
| $GT_R$ (ET in [76]) | 200.4 | 86 | 208.4 | 39 | 217.5 | 28 |
| $GT_R, \zeta = 0.1$ | 206.4 | 46 | 205.9 | 59 | 207.0 | 75 |
| $GT_R, \eta = 10^{-2}$ | 1.406 | 99 | 21.38 | 14 | 17.75 | 16 |
| $GT_R, \eta = 10^{-2}, \zeta = 0.1$ | 16.85 | 20 | 11.19 | 21 | 2.571 | 98 |
| $GT_R, \eta = \delta = 10^{-2}$ | 15.14 | 35 | 1.512 | 97 | 15.98 | 34 |
| $GT_R, \eta = \delta = 10^{-2}, \zeta = 0.1$ | 2.181 | 100 | 13.96 | 19 | 2.331 | 92 |
| $GT_A$ | 10.06 | 96 | 10.12 | 99 | 9.129 | 100 |
| $GT_A, \eta = 10^{-2}$ | 1.927 | 95 | 2.453 | 100 | 1.689 | 99 |
| $GT_A, \eta = \delta = 10^{-2}$ | **1.203** | 99 | **1.150** | 100 | **1.025** | 100 |

Consequently, once the initial condition is given, integrating (19) from $t = 0$ to a fixed future time yields how much the energy of a single instance of $u$ has decayed, and this is a deterministic quantity. This indicates that the scale-preserving property of the Galerkin projection-type layer normalization would potentially learn this decaying property resulted by the operator, thus outperforms the regular layer normalization scheme that normalizes each position's feature vector. We also note that using an instance normalization [85] may appear to be more sensible as it normalizes $\|\boldsymbol{v}^j\|$ to 1 ($1 \leq j \leq d$) after the $1/n$ weight, however, we find that opting for the instance normalization deteriorates the training's stability and the dropout needs to be further dialed up. For more heuristics of the Galerkin-type layer normalization scheme please refer to Section D.4.

### C.3 Experimental details for Example 5.2

**Data preparation.** Example 5.2 considers another well-known benchmark problem used in [57, 64, 56]. The Darcy flow in porous media $\Omega := (0,1)^2$, in which the diffusion coefficient $a \in L^\infty(\Omega)$ : $x \mapsto \mathbb{R}^+$ represents the permeability of the media and has a sharp contrast within the domain.

$$\begin{cases} -\nabla \cdot (a\nabla u) = f \text{ in } \Omega, \\ \qquad\qquad u = 0 \text{ on } \partial\Omega. \end{cases} \tag{20}$$

For each sample in the training and validation data, $a(x)$ is generated according to $a \sim \nu := \psi_\sharp \mathcal{N}(0, (-\Delta + 9I)^{-2})$, where within the covariance $-\Delta$ is defined on $(0,1)^2$ and has homogeneous Neumann boundary conditions. The mapping $\psi : \mathbb{R} \to \mathbb{R}$ is constructed by

$$\psi(\rho) = 12\mathbb{1}_{(0,\infty)}(\rho) + 3\mathbb{1}_{(-\infty,0)}(\rho).$$

Thus, the resulting coefficient $a$ follows a pushforward probability measure $\nu$, and takes values of 12 and 3 almost surely within $\Omega$. The geometry of the interface exhibits a random pattern in $\Omega$ (see Figure 10c). The forcing $f$ is fixed as $f \equiv 1$.

The operator to be learned is between the diffusion coefficient and the unique weak solution:

$$T : L^\infty(\Omega) \to H_0^1(\Omega), \quad a \mapsto u.$$

The finite dimensional approximation $u_h$'s are obtained using a 5-point stencil second-order finite difference scheme on a $421 \times 421$ grid. Therefore, the discrete operator $T_h$ to be learned is:

$$T_h : \mathbb{A}_h \mapsto \mathbb{V}_h, \quad a_h \mapsto \Pi_h u_{h^*}, \tag{21}$$

where $\mathbb{A}_h$ and $\mathbb{V}_h$ are function spaces of which the degrees of freedom are defined on the fine grid points with mesh size $h$, $h^* = 1/421$ is the finest grid size, and $a_h := I_h a$ and $\Pi_h(\cdot)$ are defined accordingly in a similar fashion with Example 5.1 explained in Appendix C.2. Following the practice of [57], a non-trainable Gaussian normalizer is built in the network to transform the input and the target to be $\sim \mathcal{N}(0,1)$ pointwisely on each grid point.

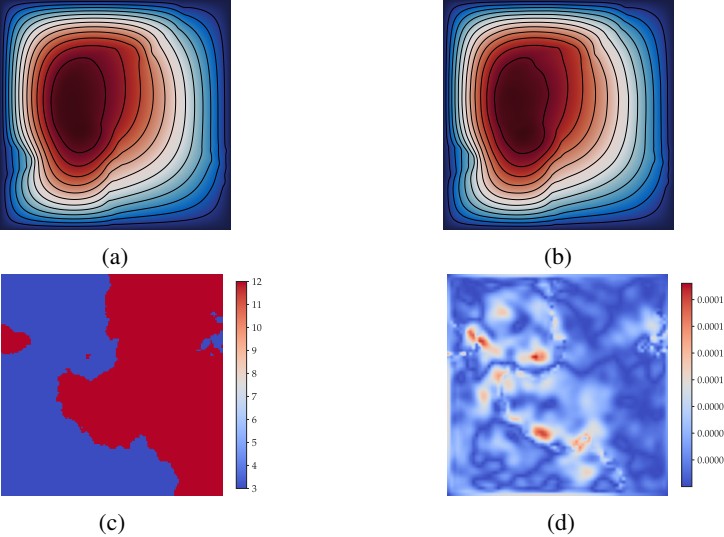

Figure 10: Interface Darcy flow in example 5.2: a randomly chosen sample from the test dataset. (a) the target being the finite difference approximation to the solution on a very fine grid; (b) the inference approximation by model evaluation (relative $L^2$-error $6.454 \times 10^{-3}$); (c) the input $a(x)$; (d) the $L^\infty$-error distribution for the inference solution.

By choosing $\mathbb{V}_h$ as the standard bilinear Lagrange finite element on a uniform Cartesian grid on $\Omega$, a standard summation by parts argument for $-\Delta_h$ and the discrete Poincaré inequality guarantees the well-posedness of problem using the Lax-Milgram lemma, i.e., given $a_h \in \mathbb{A}_h \simeq \mathbb{R}^{n_f \times n_f}$, the linear system of the $-a_h \Delta_h(\cdot)$ discretization has a unique solution $u_h$. Even though the inversion of the stiffness matrix in resulting linear system is a linear problem, the mapping in (21) is highly nonlinear between two spaces isomorphic to $\mathbb{R}^{n_f \times n_f}$.

**Limitations.** We acknowledge that our method, despite surpassing the current best operator learner's benchmark evaluation accuracy, still does reach the accuracy of traditional discretization-based methods that aim to best the approximation for a single instance. For example, in Figure 10d, the error is more prominent at the location where the coefficient has sharp contrast. How to incorporate the adaptive methods (allocating more degrees of freedom based on the a posteriori local error) to data-driven operator learners will be a future study.

## C.4 Experimental details for Example 5.3

**Inverse problems.** Playing a central role in many practical applications, the inverse problems are a class of important tasks in many scientific disciplines [51]. The problem summarizes to using the measurements to infer the material/physical coefficients. In almost all cases, the inverse problem is much harder than solving the forward problem (e.g., solving for $u$ in problem $L_a(u) = f$), as the mapping from the solution (measurements) back to the coefficient is much less stable than the forward operator due to a much bigger Lipschitz constant. As a result, the inverse operator amplifies noises in measurements by a significant amount. For example, by [3, Theorem 5.1], the error estimate of coefficient reconstruction indicates that in order that coefficient can be recovered, the measurements have to reach an accuracy with an error margin under $O(h)$ where $h$ denotes the mesh size. Meanwhile, standard iterative techniques that construct $a_\epsilon$ to approximate $a$ relies on the regularity of the coefficient $a$ itself [3, Section 3]. This regularity assumption is largely violated in our problem setting, as the $a$ has sharp material interfaces (see Figure 10c), has no extra regularity, and is only in $L^\infty$.

Having the measurements on the discrete grid, the ideal goal is to learn the following inverse map $T_h$,

$$T_h : \mathbb{X}_h \mapsto \mathbb{A}_h, \quad \Pi_h u_{h^*} \mapsto a_h, \tag{22}$$

However, in practice the measurements (solution) could have noise. Therefore, we aim to learn the following discrete operator in this example, i.e., to reconstruct the coefficient on the coarse grid based on a noisy measurement on the fine grid. We note that this operator is not well-posed anymore due to noise.

$$T_h : \mathbb{X}_{h_f} \mapsto \mathbb{A}_{h_c}, \quad u_{h_f} + \epsilon \nu_{h_f} \mapsto \Pi_{h_c} a_h, \tag{23}$$

where $h_c, h_f$ denotes the mesh size of the coarse grid $n_c \times n_c$, and the fine grid $n_f \times n_f$, respectively. $\Pi_{h_c}$ denotes a map that restricts a function $\mathbb{X}_h$ defined on $n_f \times n_f$ to $n_c \times n_c$. $\epsilon$ is the strength of the noise, and $\nu_{h_f}(x_i) \sim \mathcal{N}(0, c_i)$ where $c_i$ is the variance of all training samples' value at $x_i$. If $\epsilon = 0.1$, we have 10% of noise in the solution measurements for the training and testing data (see Figure 11).

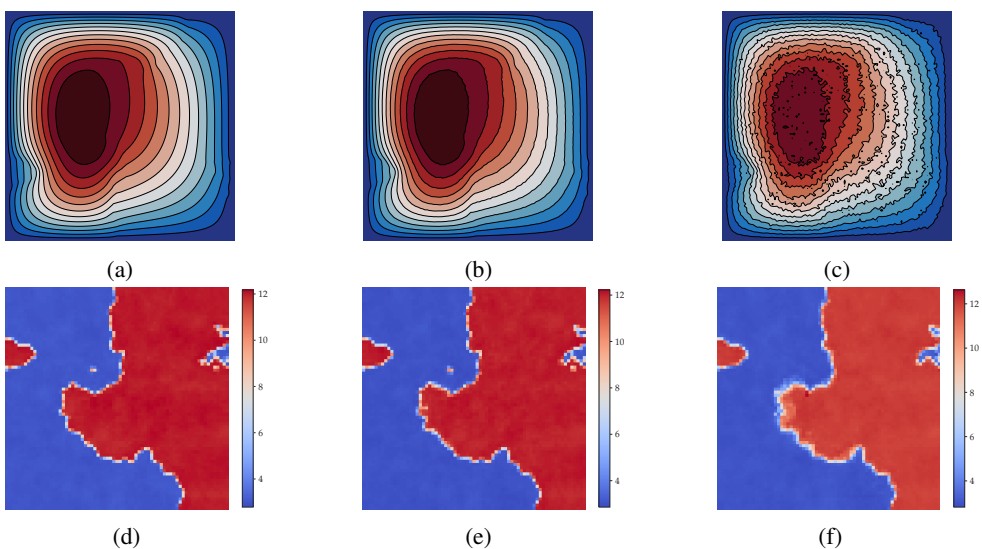

(a)        (b)        (c)

(d)        (e)        (f)

Figure 11: Galerkin transformer evaluation for the inverse interface coefficient identification problem using the same sample with Figure 10, model trained and evaluated under the same amount of noises: (a)–(c) the input $u_h(x)$ with noise level 0, 1%, and 10% on $211 \times 211$ grid; (d)–(f) the recovered coefficient through evaluation with noise level 0, 1%, and 10% on $71 \times 71$ grid with relative error being 0.0160, 0.0292, and 0.0885, respectively.

**Why not fine grid reconstruction?** The reason we can only reconstruct the coarse grid coefficient is as follows. Since we use an upsampling interpolation from the coarse to fine grids, a limitation of the 2D operator learner structure in Figure 5 is that it can approximate well if the target is smooth, and consists most combination of basis functions of lower frequencies. The low-frequency part, which can be roughly interpreted as the general trends, of the solution can be well-resolved by the coarse grid, then the operator learner benefits from the smoothing property of the operator a priori, as well as the approximation property of the interpolation operator. If the high frequency part of the target is prevailing due to low regularity (such as $L^\infty$), the model can only resolve the frequency up to of grid size $n_c \times n_c$ as the upsampling interpolation loses the approximation order (prolongation error estimate from coarse to fine grids, see e.g., [38, Chapter 6.3]).

**More limitations.** Moreover, we do acknowledge another limitation of the proposed operator learner: it suffers from a common foe in many applications of deep learning, the instability with respect to the noise during evaluation. If the model is trained with clean data, in evaluation it becomes oversensitive to noises, especially in Example 5.3 due to the large Lipschitz constant in the original problem itself, which is further amplified by the black-box model. Therefore, we recommend adding certain amount of noises for inverse coefficient identification problems. See Figure 12.

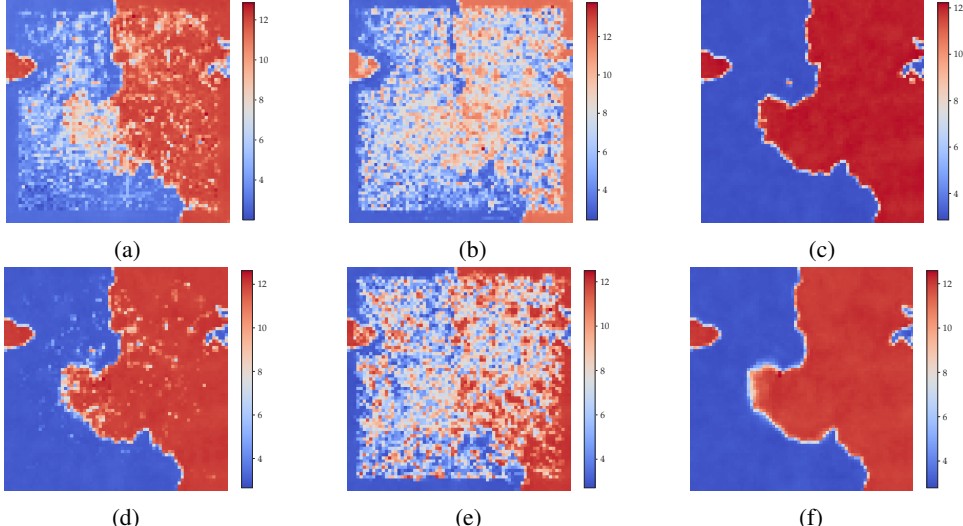

Figure 12: Effect of noise in the inverse interface coefficient identification using the same sample with Figure 10 and 11, $\varepsilon$: relative error in $L^2$-norm. (a)–(b) model trained with no noise, 1% and 1.5% noises in evaluation, $\varepsilon = 0.194$ and $\varepsilon = 0.416$; (c) model trained with 1% noise, no noise in evaluation, $\varepsilon = 0.0235$; (d) model trained with 1% noise, 2% in evaluation, $\varepsilon = 0.0754$. (e) model trained with 1% noise, 5% in evaluation, $\varepsilon = 0.403$. (f) model trained with 10% noise, 5% in evaluation, $\varepsilon = 0.0691$.

## D Proof of The Approximation Capacity of A Linear Attention

In this section, we prove Theorem 4.3, which shows that the linear attention variant we proposed in (6), Galerkin-type attention operator (nonlinear), is capable of replicating explicitly a Petrov-Galerkin projection (linear) in the current latent representation subspace under a Hilbertian setup. To better elaborate our Galerkin projection-inspired modifications to the attention operator, in Section D.1, the technical background that bridges (14) to a learnable Petrov-Galerkin projection is presented. Then, some historical contexts are provided in Section D.2 for an overview of Theorem 4.3, connecting the sequence-length invariant training of the attention operator to how important a theorem like Theorem 4.3 is for an operator approximation problem in traditional applied mathematics. In Section D.3, the proof of Theorem 4.3 is shown with a full array of mathematically rigorous setting and assumptions. Thereafter, in Section D.4 some possible generalizations are discussed, together with the role of removing the softmax in obtaining a sequence-length uniform bound, as well as the heuristic behind the choice of the Galerkin projection-type normalization in the scaled dot-product attention. Last but not least, technical lemmata that are needed to show Theorem 4.3 are proved in Section D.5.

### D.1 Background on Galerkin methods

The huge success of many Galerkin-type methods in approximating solutions to operator equations such as PDEs [20] attributes partly to the following two fundamental properties of Hilbert spaces (see e.g., [21, Chapter 4]): for $(\mathcal{H}, \langle \cdot, \cdot \rangle)$

- Let $\mathcal{Y}$ be a convex and complete subset of $\mathcal{H}$, and $\mathcal{Y}$ is potentially infinite-dimensional. For any $f \in \mathcal{H}$, the projection $\Pi f \in \mathcal{Y}$ is uniquely determined by

$$\|f - \Pi f\|_{\mathcal{H}} = \inf_{y \in \mathcal{Y}} \|f - y\|_{\mathcal{H}}, \tag{24}$$

  i.e., the projection recovers the unique element in $\mathcal{Y}$ that is "closest" to $f$.

- If $\mathcal{H}$ is infinitely-dimensional and separable, then there exists a set of orthogonal basis functions $\{q_l(\cdot)\}_{l=1}^{\infty}$ such that any $f \in \mathcal{H}$ can have its Fourier series expansion:

$$f(\cdot) = \sum_{l=1}^{\infty} a_l q_l(\cdot) := \sum_{l=1}^{\infty} \frac{\langle f, q_l \rangle}{\langle q_l, q_l \rangle} q_l(\cdot), \tag{25}$$

i.e., $\mathcal{H}$ can be identified by $\ell^2$ (the space that the Fourier coefficients $\{a_l\}_{l=1}^\infty$ are in). Thus, given any fixed tolerance under the norm induced by the inner product, any $f \in \mathcal{H}$ can be approximated using a finite number of basis $\{q_l(\cdot)\}_{l=1}^d$ (identified by a finite number of coefficients) thanks to the square summability.

Together, they rationalize the practice of using a finite dimensional vector (space) to approximate any element in $\mathcal{H}$, or a function in the solution subspace of an operator equation that is compact in $\mathcal{H}$. Consider the finite dimensional approximation space (trial space) $\mathbb{Q}_h := \mathrm{span}\{\tilde{q}_l(\cdot)\}_{l=1}^d$ (that is convex and closed), where $\tilde{q}_l(\cdot) := q_l(\cdot)/\|q_l\|_{\mathcal{H}}$. The projection $\Pi f$ onto $\mathbb{Q}_h$ is the best approximator in $\mathbb{Q}_h$ to $f \in \mathcal{H}$:

$$\|f - \Pi f\|_{\mathcal{H}} = \inf_{q \in \mathbb{Q}_h} \|f - q\|_{\mathcal{H}}. \tag{26}$$

Exploiting the definition of $\|\cdot\|_{\mathcal{H}}$, any perturbation $q \in \mathbb{Q}_h$ (test space) to the unique (local) minimizer $p := \Pi f$ shall increase the difference in $\|\cdot\|_{\mathcal{H}}$, thus

$$0 = \lim_{\tau \to 0} \frac{d}{d\tau} \|f - (p + \tau q)\|_{\mathcal{H}}^2 = \lim_{\tau \to 0} \frac{d}{d\tau} \langle f - (p + \tau q), f - (p + \tau q)\rangle,$$

Choosing $q$ as $\tilde{q}_l$ ($l = 1, \ldots, d$), one shall obtain the following if assuming that $\mathcal{H} = L^2(\Omega)$ in a simple case

$$\Pi f(x) = \sum_{l=1}^d \langle f, \tilde{q}_l\rangle \tilde{q}_l(x) = \sum_{l=1}^d \left(\int_\Omega f(\xi)\tilde{q}_l(\xi)\, \mathrm{d}\xi\right) \tilde{q}_l(x), \quad \text{for } x \in \Omega. \tag{27}$$

When the set $\{f_j(\cdot)\}_{j=1}^d$ is projected onto $\mathbb{Q}_h$ element by element, (27) carries a resoundingly similar form to that of (14). In light of proving the approximation capacity of (14), the differences are:

(a) The test and trial function spaces are the same in the ideal case above (27), while in a Galerkin-type attention operator (14), they are different and become learnable. This difference brings the Petrov-Galerkin projection into the picture. In Section D.3, we shall see that the minimization is done for a more general functional (dual) norm $\|\cdot\|_{\mathbb{V}'_h}$ (min-max problem (33)), instead of $\|\cdot\|_{\mathcal{H}}$.

(b) $\{\tilde{q}_l(\cdot)\}_{l=1}^d$ needs to be orthonormal to yield a compact formula as (27). In the forward propagation of the attention operations, there is no such guarantee unless certain orthogonalization/normalization is performed. We shall see in the proof in Section D.3, the Galerkin projection-type layer normalization acts as a cheap learnable alternative to the normalization shown in the explicit formula Petrov-Galerkin projection (inverse of the Gram matrices in (40)).

### D.2 Overview of Theorem 4.3

**Historical context.** Theorem 4.3 resembles the famous Céa's lemma (e.g., see [13, Theorem 2.8.1], [20, Theorem 2.4.1]). It is one of the most fundamental theorems in approximating an operator equation such as a PDE under the Hilbertian framework. Define the operator norm to be the induced norm from the original Hilbertian norm, the Céa's "lemma" reads: if the norm of the operator associated with the bilinear form is bounded below (either by the Lax-Milgram lemma or the Ladyzhenskaya–Babuška–Brezzi inf-sup condition) in an approximation subspace of the original Hilbert space, which implies its invertibility, then this mere invertibility implies the quasi-optimality (optimal up to a constant) of the Galerkin-type projection to a given function. This says: in the current approximation space, measured by the distance of the Hilbertian norm, the Galerkin-type or the Petrov-Galerkin-type projection (see e.g., [80]) is equivalent to the closest possible approximator to any given target function (see (26)).

As might be expected, whether this approximation space has enough approximation power, such that this closest approximator is actually close to the target, is another story. This closeness, either in terms of distance or structure, is usually referred as a part of "consistency" in the context of approximating a PDE's solution operator. Any method with a sufficient approximation power together with the invertibility has "convergence", and this is also known as the Lax equivalence principle [53].

**Heuristic comparison: convergence of traditional numerical methods versus a data-driven operator learner.** For a traditional numerical method that approximates an operator equation to be successful, in that scientists and engineers can trust the computer-aided simulations, the aforementioned convergence is indispensable. One key difference of a traditional numerical method to an attention-based operator learner is how the "convergence" is treated.

- A traditional numerical method: once the discretization is fixed (fixed degrees of freedom such as grids, radial bases, Fourier modes, etc.), the approximation power of this finite dimensional approximation space is fixed. The method seeks the best approximator to a single instance in this fixed space. The convergence refers to the process of error decreasing as one continuously enlarges the approximation subspaces (e.g., grid refinement, more Fourier modes).

- An attention-based operator learner: the approximation space is not fixed, and is constantly replenished with new bases during optimization (e.g., see the remarks in 4.1.3), thus able to approximate an operator's responses in a subset/subspace in a much more dynamic manner. A possible exposition of the "convergence" is more problem-dependent as one obtains a better set of basis to characterize the operator of interest progressively through the stochastic optimization.

**Positional encoding and a dynamic feature/basis generation.** Even though [88] opens a new era in NLP by introducing the state-of-the-art Transformer, in an operator learning problem, we find that its explanation unsatisfactory on the absolute necessity to include the positional encodings in the attention mechanism. From the proof of Theorem 4.3 we see that, if we ought to learn the identity operator from $\mathbb{Q}_h$ to $\mathbb{Q}_h$ (with a few other caveats), i.e., the target $f$ itself is in the approximation subspace $\mathbb{Q}_h$, then the attention mechanism has certainly the capacity to learn this operator exactly without any positional encodings. We want to emphasize that in addition to the remarks in 4.1.3, in our interpretation, the utmost importance of the positional encoding is to make the approximation subspaces dynamic. Otherwise, the Galerkin-type attention (or a linear attention) is simply a linear combination of the current approximation subspace (or a convex combination in its softmax-normalized siblings). Consequently, the approximation power of the subspaces cannot enjoy this dynamic update mechanism through optimizations. In this sense, we can view the attention mechanism a universal "dynamic feature/basis generator" as well. For an empirical evidence of this layer-wise dynamic basis update in our experiments of the Darcy interface flow please refer to Figure 13.

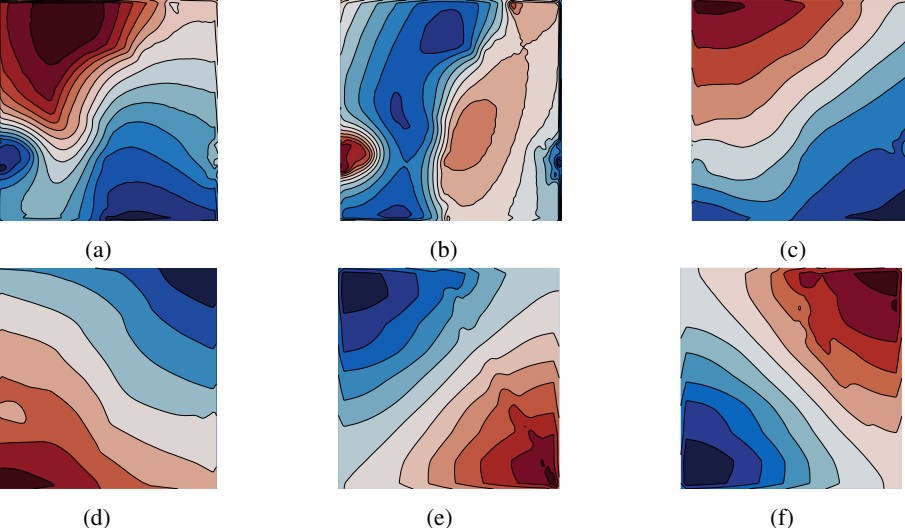

(a)  (b)  (c)

(d)  (e)  (f)

Figure 13: Extracted latent representation sequences reshaped to $n_c \times n_c$ from the encoder layers in the Galerkin Transformer using the same sample with Figure 10 and 11 in evaluation; (a)–(b): two basis functions from the first encoder layer; (c)–(d): two other basis functions from the fourth encoder layer, and we note that visually all basis functions in the fourth layers are smoother than those in the first; (e)–(f): $-\nabla \cdot (a\nabla(\cdot))$'s first two eigenfunction approximations using the bilinear finite element on the $n_c \times n_c$ grid, i.e., the first two Fourier bases associated with this self-adjoint operator.

## D.3 Proof of Theorem 4.3

**Notations.** Before presenting the proof, to avoid confusion of the notions for various finite dimensional function spaces, the settings for different spaces are paraphrased from the condensed versions in Table 4. The caveat is that for the same function in the latent approximation space, it has two vector representations: (i) nodal values at the grid points which can be used to form the columns of $Q, K, V$, this vector is in $\mathbb{R}^n$; (ii) the vector representation using degrees of freedom (coefficient functional) in an expansion using certain set of basis, this vector is in $\mathbb{R}^d$ or $\mathbb{R}^r$ ($r \leq d < n$).

We also note that in the context of using the Galerkin-type attention (or other linear attentions) to approximate functions under a Hilbertian framework, in that the output of the attention-based map can represent a Petrov-Galerkin projection, $Q$ stands for values, $K$ for query, and $V$ for keys. In the proof of Theorem 4.3, we shall refer the discrete approximation space generated by $Q$ as a "value space", and that of $V$ as a "key space". Meanwhile, $\Omega$ and $\Omega^*$ can be seen as spacial/temporal domain and frequency domain, respectively.

**Assumption D.1** (assumptions and settings for Theorem 4.3). *The following notations and assumptions are used throughout the proof of the quasi-optimal approximation result in Theorem 4.3:*

$(D_1)$ $(\mathcal{H}, \langle \cdot, \cdot \rangle_{\mathcal{H}})$ *is a Hilbert space. For $f \in \mathcal{H}$, $f : \Omega \to \mathbb{R}$. $\mathcal{H} \hookrightarrow C^0(\Omega)$. $\Omega \subset \mathbb{R}^m$ is a bounded domain, discretized by $\{x_i\}_{i=1}^n$ with a mesh size $h$.*

$(D_2)$ $\mathbb{Y}_h \subset \mathcal{H}$ *is an approximation space associated with $\{x_i\}_{i=1}^n$, such that for any $y \in \mathbb{Y}_h$, $y(\cdot) = \sum_{i=1}^n y(x_i)\phi_{x_i}(\cdot)$ where $\{\phi_{x_i}(\cdot)\}_{i=1}^n$ form a set of nodal basis for $\mathbb{Y}_h$ in the sense that $\phi_{x_i}(x_j) = \delta_{ij}$, and the support of every nodal basis $\phi_{x_i}(\cdot)$ is of $O(h^m)$.*

$(D_3)$ $(\mathcal{V}, \langle \cdot, \cdot \rangle_{\mathcal{V}})$ *is a latent Hilbert space. For $v \in \mathcal{V}$, $v : \Omega^* \to \mathbb{R}$. $\mathcal{V} \hookrightarrow C^0(\Omega^*)$. $\Omega^* \simeq \Omega$ and is discretized by $\{\xi_i\}_{i=1}^n$ with a mesh size $h$.*

$(D_4)$ $\mathbb{W}_h \subset \mathcal{V}$ *is an approximation space associated with $\{\xi_i\}_{i=1}^n$, i.e., for any $w \in \mathbb{W}_h$, $w(\cdot) = \sum_{i=1}^n w(\xi_i)\psi_{\xi_i}(\cdot)$ where $\{\psi_{\xi_i}(\cdot)\}_{i=1}^n$ form a set of nodal basis for $\mathbb{W}_h$ in the sense that $\psi_{\xi_i}(\xi_j) = \delta_{ij}$, and the support of every nodal basis $\psi_{\xi_i}(\cdot)$ is of $O(h^m)$.*

$(D_5)$ $\mathbf{y} \in \mathbb{R}^{n \times d}$ *is the current input latent representation. $W^Q, W^K, W^V$ denote the current projection matrices. $n > d > m$ and $\mathrm{rank}\, \mathbf{y} = d$.*

$(D_6)$ $\mathbb{Q}_h \subset \mathbb{Y}_h \subset \mathcal{Q}$ *is the current value space from $Q$. $\mathcal{Q}$ is a subspace of $\mathcal{H}$ with the same topology. $\mathbb{Q}_h$ is the spanned by basis functions whose degrees of freedom associated with $\{x_i\}_{i=1}^n$ form the columns of $Q := \mathbf{y}W^Q \in \mathbb{R}^{n \times d}$, i.e., $\mathbb{Q}_h = \mathrm{span}\{q_j(\cdot) \in \mathbb{Y}_h : q_j(x_i) = Q_{ij}, 1 \leq i \leq n, 1 \leq j \leq d\}$.*

$(D_7)$ $\mathbb{V}_h \subset \mathbb{W}_h \subset \mathcal{V}$ *is the current key space from $V$. $\mathbb{V}_h$ is the spanned by basis functions whose degrees of freedom associated with $\{\xi_i\}_{i=1}^n$ form the columns of $V := \mathbf{y}W^V \in \mathbb{R}^{n \times d}$, i.e., $\mathbb{V}_h = \mathrm{span}\{v_j(\cdot) \in \mathbb{W}_h : v_j(\xi_i) = V_{ij}, 1 \leq i \leq n, 1 \leq j \leq d\}$.*

$(D_8)$ $\mathbb{V}_h'$ *is the dual space of $\mathbb{V}_h$ consisting of all bounded linear functionals defined on $\mathbb{V}_h$; $\|g(\cdot)\|_{\mathbb{V}_h'} := \sup_{v \in \mathbb{V}_h} |g(v)|/\|v\|_{\mathcal{V}}$ for $g \in \mathbb{V}_h'$.*

$(D_9)$ $\dim \mathbb{Q}_h = r \leq \dim \mathbb{V}_h = d$, *i.e., the key space is bigger than the value space.*

$(D_{10})$ *For $w \in \mathbb{V}_h$, $w(\cdot) = \sum_{j=1}^d \mu_{v_j}(w)v_j(\cdot)$ is the expansion in $\{v_j(\cdot)\}_{j=1}^d$, where $\mu_{v_j}(\cdot) \in \mathbb{V}_h'$ is the coefficient functional; in this case, $w$ can be equivalently identified by its vector representation $\boldsymbol{\mu}(w) := (\mu_{v_1}(w), \cdots, \mu_{v_d}(w))^\top \in \mathbb{R}^d$.*

$(D_{11})$ *For $p \in \mathbb{Q}_h$, $p(\cdot) = \sum_{j=1}^r \lambda_{q_j}(p)q_j(\cdot)$ is the expansion in $\{q_j(\cdot)\}_{j=1}^r$, where $\lambda_{q_j}(\cdot) \in \mathbb{Q}_h'$ is the coefficient functional; in this case, $p$ can be equivalently identified by its vector representation $\boldsymbol{\lambda}(p) := (\lambda_{q_1}(p), \cdots, \lambda_{q_r}(q))^\top \in \mathbb{R}^r$.*

$(D_{12})$ $\mathfrak{b}(\cdot, \cdot) : \mathcal{V} \times \mathcal{Q} \to \mathbb{R}$ *is a continuous bilinear form, i.e., $|\mathfrak{b}(v, q)| \leq c_0 \|v\|_{\mathcal{V}} \|q\|_{\mathcal{H}}$ for any $v \in \mathcal{V}$, $q \in \mathcal{Q}$. For $(w, y) \in \mathbb{W}_h \times \mathbb{Y}_h \subset \mathcal{V} \times \mathcal{Q}$, $\mathfrak{b}(w, y) := h^m \sum_{i=1}^n w(\xi_i)y(x_i)$.*

$(D_{13})$ $g_\theta(\cdot) : \mathbb{R}^{n \times d} \to \mathbb{Q}_h, \mathbf{y} \mapsto z$ *is a learnable map that is the composition of the Galerkin-type attention operator (6) with an updated set of $\{\widetilde{W}^Q, \widetilde{W}^K, \widetilde{W}^V\}$ and a pointwise universal approximator; $\theta$ denotes all the trainable parameters within.*

**Theorem 4.3** (Céa-type lemma, general version). *For any $f \in \mathcal{H}$, under Assumption D.1, for $f_h \in \mathbb{Q}_h$ being the best approximator of $f$ in $\|\cdot\|_{\mathcal{H}}$, we have:*

$$\min_{\theta} \|f - g_{\theta}(\mathbf{y})\|_{\mathcal{H}} \le c^{-1} \min_{q \in \mathbb{Q}_h} \max_{v \in \mathbb{V}_h} \frac{|\mathfrak{b}(v, f_h - q)|}{\|v\|_{\mathcal{V}}} + \|f - f_h\|_{\mathcal{H}}. \tag{28}$$

*Proof.* By triangle inequality, inserting the best approximation $f_h \in \mathbb{Q}_h$

$$\|f - g_{\theta}(\mathbf{y})\|_{\mathcal{H}} \le \|f_h - g_{\theta}(\mathbf{y})\|_{\mathcal{H}} + \|f - f_h\|_{\mathcal{H}}. \tag{29}$$

$f_h := \operatorname{argmin}_{q \in \mathbb{Q}_h} \|f - f_h\|_{\mathcal{H}}$ describes the approximation capacity of the current value space $\mathbb{Q}_h$ and has nothing to do with $\theta$.

In the first part of the proof, we focus on bridging $\|f_h - g_{\theta}(\mathbf{y})\|_{\mathcal{H}}$ in (29) to the linear problem associated with seeking the Petrov-Galerkin projection of $f_h$. By Lemma D.6, the continuous linear functional defined by $\mathfrak{b}(\cdot, q) : \mathcal{V} \to \mathbb{R}$, $v \mapsto \mathfrak{b}(v, q)$ is bounded below on the current key space, i.e., there exists $c > 0$ independent of $q$ or the discretization (mesh size $h$) such that,

$$\|\mathfrak{b}(\cdot, q)\|_{\mathbb{V}'_h} \ge c\|q\|_{\mathcal{H}}, \quad \text{for any fixed } q \in \mathbb{Q}_h. \tag{30}$$

By the definition of $\|\cdot\|_{\mathbb{V}'_h}$ in $(D_8)$, we have

$$\|f_h - g_{\theta}(\mathbf{y})\|_{\mathcal{H}} \le c^{-1} \sup_{v \in \mathbb{V}_h} \frac{|\mathfrak{b}(v, f_h - g_{\theta}(\mathbf{y}))|}{\|v\|_{\mathcal{V}}} = c^{-1} \max_{v \in \mathbb{V}_h} \frac{|\mathfrak{b}(v, f_h - g_{\theta}(\mathbf{y}))|}{\|v\|_{\mathcal{V}}}.$$

In the rest of the proof, the goal is to establish

$$\min_{\theta} \max_{v \in \mathbb{V}_h} \frac{|\mathfrak{b}(v, f_h - g_{\theta}(\mathbf{y}))|}{\|v\|_{\mathcal{V}}} \le \min_{q \in \mathbb{Q}_h} \max_{v \in \mathbb{V}_h} \frac{|\mathfrak{b}(v, f_h - q)|}{\|v\|_{\mathcal{V}}}, \tag{31}$$

i.e., the best approximation based on the attention operator, if exists, is on par with that offered by a Petrov-Galerkin projection.

By the continuity of $\mathfrak{b}(\cdot, \cdot)$ in $(D_{12})$, given any fixed $q \in \mathbb{Q}_h$, $v \mapsto \mathfrak{b}(v, q)$ defines a bounded linear functional for any $v \in \mathbb{V}_h$. Moreover, since we use $\ell^2$-inner product to approximate $\langle \cdot, \cdot \rangle$ on $\mathbb{V}_h$, define

$$\langle u, v \rangle_h := h^m \sum_{i=1}^{n} u(\xi_i) v(\xi_i) \approx \int_{\Omega^*} u(\xi) v(\xi) \, d\xi =: \langle u, v \rangle_{\mathcal{V}}, \quad \text{for } u, v \in \mathbb{V}_h. \tag{32}$$

Applying the Riesz representation theorem (e.g., see [21, Theorem 4.6]), together with Lemma D.3, implies that there exists a value-to-key linear map $\Phi$ for $f_h \in \mathbb{Q}_h$

$$\Phi : \mathbb{Q}_h \to \mathbb{V}_h, \text{ such that } \mathfrak{b}(v, f_h) = \langle \Phi f_h, v \rangle_h \text{ for any } v \in \mathbb{V}_h.$$

Thus, we have for the right-hand side of (31)

$$\min_{q \in \mathbb{Q}_h} \max_{v \in \mathbb{V}_h} \frac{|\mathfrak{b}(v, f_h - q)|}{\|v\|_{\mathcal{V}}} = \min_{q \in \mathbb{Q}_h} \max_{v \in \mathbb{V}_h} \frac{|\langle \Phi f_h, v \rangle_h - \mathfrak{b}(v, q)|}{\|v\|_{\mathcal{V}}}. \tag{33}$$

Using $(D_6), (D_7), (D_{10}), (D_{11})$, and Lemma D.5, the problem on the right-hand side of (33) is equivalent to the block matrix form (47) as follows.

$$\begin{pmatrix} M & B^{\top} \\ B & 0 \end{pmatrix} \begin{pmatrix} \boldsymbol{\mu} \\ \boldsymbol{\lambda} \end{pmatrix} = \begin{pmatrix} \boldsymbol{\zeta} \\ 0 \end{pmatrix}. \tag{34}$$

In the system above, $(\boldsymbol{\mu}, \boldsymbol{\lambda}) \in \mathbb{R}^d \times \mathbb{R}^r$ are the vector representations of the critical point $(w, p) \in \mathbb{V}_h \times \mathbb{Q}_h$, if exists, under basis sets $\{v_j(\cdot)\}_{j=1}^d$ and $\{q_j(\cdot)\}_{j=1}^r$, respectively, and

$$\boldsymbol{\zeta} \in \mathbb{R}^d, \quad \text{where } (\boldsymbol{\zeta})_j = \langle \Phi f_h, v_j \rangle_h \approx \int_{\Omega^*} (\Phi f_h)(\xi) v_j(\xi) \, d\xi. \tag{35}$$

In the next part of the proof, we shall seek the solution to (34), which is the Petrov-Galerkin projection to the function of interest $f_h$. Since $\{v_j(\cdot)\}_{j=1}^d$ form a set of basis of $\mathbb{V}_h$, $M$ is invertible, and we can eliminate $\boldsymbol{\mu}$ by solving the first equation above and plugging it in the second to get

$$BM^{-1}B^{\top}\boldsymbol{\lambda} = BM^{-1}\boldsymbol{\zeta}. \tag{36}$$

Knowing that $M^{-1}$ is symmetric positive definite, to show

$$\boldsymbol{\lambda} = (BM^{-1}B^\top)^{-1}BM^{-1}\boldsymbol{\zeta}, \tag{37}$$

it suffices to show that $B$ is surjective (full row rank) if we ought to use Lemma D.4. A simple argument by contradiction is as follows: suppose $B$ is not full row rank, then there exists a linear combination of the rows of $B$ being the 0 vector, i.e., there exists a set of nontrivial coefficients $\tilde{\boldsymbol{\lambda}} := (\tilde{\lambda}_1, \ldots, \tilde{\lambda}_r)^\top$ such that

$$\mathfrak{b}(v, \tilde{p}) = 0, \text{ for any } v \in \mathbb{V}_h \text{ where } 0 \not\equiv \tilde{p}(\cdot) := \sum_{l=1}^r \tilde{\lambda}_l q_l(\cdot).$$

This is contradictory to the lower bound of $\mathfrak{b}(\cdot, \tilde{p})$ in (30):

$$0 < c\|\tilde{p}\|_{\mathcal{H}} \leq \|\mathfrak{b}(\cdot, \tilde{p})\|_{\mathbb{V}_h'} = \max_{v \in \mathbb{V}_h} \frac{|\mathfrak{b}(v, \tilde{p})|}{\|v\|_{\mathcal{V}}} = 0.$$

Thus, (37) holds and the critical point $p$ (or its vector representation) exists and is a local minimizer due to $M$ being positive definite.

The last part of the proof is to show that this $p \in \mathbb{Q}_h$ is representable by the learnable map $g_\theta(\mathbf{y})$. To this end, we multiply a permutation matrix $U \in \mathbb{R}^{d \times d}$ to $Q$, such that $QU$'s first $r$ columns $Q_0 \in \mathbb{R}^{n \times r}$ form the nodal value vector $(q_j(x_1), \cdots, q_j(x_n))^\top$ for bases $\{q_j(\cdot)\}_{j=1}^r$ of $\mathbb{Q}_h$; see $(D_2)$. Then, multiplying the permuted basis matrix $QU$ with $(\boldsymbol{\lambda} \ 0)^\top \in \mathbb{R}^d$ yields the best approximator $p$'s vector representation at the grid points $\boldsymbol{p} := (p(x_1), \cdots, p(x_n))^\top$

$$\boldsymbol{p} = (Q_0 \ \ Q_1) \begin{pmatrix} \boldsymbol{\lambda} \\ 0 \end{pmatrix} = QU \begin{pmatrix} \boldsymbol{\lambda} \\ 0 \end{pmatrix}. \tag{38}$$

Moreover, since $B_{ij} = \mathfrak{b}(v_j, q_i)$ with $\{q_i(\cdot)\}_{i=1}^r$ and $\{v_j(\cdot)\}_{j=1}^d$ being the sets of basis for $\mathbb{Q}_h$ and $\mathbb{V}_h$, it is straightforward to verify that $B = h^m Q_0^\top V$; see $(D_{12})$. Here without loss of generality, we simply assume that the nodal value vector representation $(v_j(\xi_1), \cdots, v_j(\xi_n))^\top =: \boldsymbol{v}^j$ forms the $j$-th column of $V$ in a sequential order. Therefore, using (37), $(\boldsymbol{\lambda} \ 0)^\top \in \mathbb{R}^d$ can be written as the following block form:

$$\begin{pmatrix} \boldsymbol{\lambda} \\ 0 \end{pmatrix} = \begin{pmatrix} (BM^{-1}B^\top)^{-1} & \\ & 0 \end{pmatrix} \begin{pmatrix} B \\ * \end{pmatrix} M^{-1}\boldsymbol{\zeta} = h^m \begin{pmatrix} (BM^{-1}B^\top)^{-1} & \\ & 0 \end{pmatrix} \begin{pmatrix} Q_0^\top \\ Q_1^\top \end{pmatrix} V M^{-1}\boldsymbol{\zeta}. \tag{39}$$

Consequently, the ideal updated $\{\widetilde{Q}, \widetilde{K}, \widetilde{V}\}$ that has capacity to obtain the current best approximator $p$, or its vector presentation $\boldsymbol{p}$ in (38), are

$$\begin{aligned} \widetilde{Q} &:= \mathbf{y}\widetilde{W}^Q \leftarrow \mathbf{y}W^Q U, \\ \widetilde{K} &:= \mathbf{y}\widetilde{W}^K \leftarrow \mathbf{y}W^Q U\Lambda, \\ \widetilde{V} &:= \mathbf{y}\widetilde{W}^V \leftarrow \mathbf{y}W^V M^{-1}, \\ \text{and } \ \boldsymbol{p} &= h^m \widetilde{Q}(\widetilde{K}^T \widetilde{V})\boldsymbol{\zeta}, \end{aligned} \tag{40}$$

where $\Lambda := \mathrm{blkdiag}\left\{(BM^{-1}B^\top)^{-1}, 0\right\}$ and is symmetric. This essentially implies that $g_\theta(\cdot)$ has capacity to learn this best approximator $p$ using the updated set of $\{\widetilde{Q}, \widetilde{K}, \widetilde{V}\}$ from (40):

$$g_\theta(\cdot) : \mathbb{R}^{n \times d} \to \mathbb{Q}_h, \mathbf{y} \mapsto p, \text{ such that } p(x_i) = \left(h^m \widetilde{Q}(\widetilde{K}^T \widetilde{V})\boldsymbol{\zeta}\right)_i \text{ for } i = 1, \cdots, n.$$

Passing the inequality from minimizing in a bigger set yields the desired inequality (31), which, in turn, shows the approximation capacity indicated in the theorem. $\qquad\square$

**Dynamical basis update for a set of functions.** Despite the fact that Theorem 4.3 is for a single instance of $f \in \mathcal{H}$, it is not difficult to see that the form (39) easily extends to a latent representation in $\mathbb{R}^{n \times d}$ whose columns are $\{f_{j,h}(\cdot)\}_{j=1}^d$ sampled at the grid points $\{x_i\}_{i=1}^n$. Simply replacing $\boldsymbol{\zeta}$ in (35) by a matrix $W^Z := (\boldsymbol{\zeta}_1, \cdots, \boldsymbol{\zeta}_d) \in \mathbb{R}^{d \times d}$, of which the $i$-th entry in the $j$-th column is $(\boldsymbol{\zeta}_j)_i = \langle \Phi f_{j,h}, v_i \rangle_h$, we can verify that a pointwise FFN $g_\theta(\cdot)$ is fully capable of representing $W^Z$.

Using the argument above, we can further elaborate the dynamical basis update nature of the scaled dot-product attention (14) (see also the remarks in 4.1.3 and Appendix D.2.3):

- Test the columns of $V$ (query) against the columns of $K$ (keys), and use the responses to seek a "better" set of basis using the columns of $Q$ (values).

To this end, we modify Assumption D.1 slightly by appending (with a superscript $+$) or redefining (with a superscript $*$) certain entries as follows.

**Assumption D.2** (assumptions and settings for Theorem 4.4). *Assumption D.1 are assumed to hold with the following amendments to their respective numbered entries for the proof of Theorem 4.4:*

$(D_6^+)$ $\mathbb{K}_h \subset \mathbb{Y}_h \subset \mathcal{K}$ *is the current key space from $K$ and is defined on $\Omega^*$. Define $\langle s, k \rangle_h := h^m \sum_{i=1}^n s(\xi_i) k(\xi_i)$ as an inner product for $s, k \in \mathbb{K}_h$.*

$(D_7^*)$ $\mathbb{V}_h \subset \mathbb{W}_h \subset \mathcal{V} \subseteq \mathcal{Q}$ *is the current query space from $V$ defined on $\Omega$.*

$(D_{12}^*)$ $\mathfrak{b}(\cdot, \cdot) : \mathcal{K} \times \mathcal{Q} \to \mathbb{R}$ *is continuous. $\mathfrak{b}(k, q) := h^m \sum_{i=1}^n k(\xi_i) q(x_i)$ for $(k, q) \in \mathbb{Y}_h \times \mathbb{W}_h$.*

$(D_{12}^+)$ $\mathfrak{a}(\cdot, \cdot) : \mathcal{V} \times \mathcal{K} \to \mathbb{R}$ *is continuous. $\mathfrak{a}(w, k) := h^m \sum_{i=1}^n w(x_i) k(\xi_i)$ for $(w, k) \in \mathbb{W}_h \times \mathbb{Y}_h$.*

After the introduction of $\mathfrak{a}(\cdot, \cdot)$, a new continuous functional can be defined as follows:

$$\ell_{(w,q)}(\cdot) : \mathcal{K} \to \mathbb{R}, \quad k \mapsto \mathfrak{a}(w, k) - \mathfrak{b}(k, q). \tag{41}$$

**Theorem 4.4** (layer-wise basis update of the scaled dot-product attention). *Under Assumption D.2, and the columns of $V$ and $Q$ are formed by the DoFs of query and value functions, respectively, i.e., $V_{ij} = v_j(x_i)$ and $Q_{ij} = q_j(x_i)$, $1 \le i \le n$, $1 \le j \le d$. Then, there exists an updated set $\{\tilde{q}_l(\cdot)\}_{l=1}^d \subset \mathbb{Q}_h = \mathrm{span}\{q_l(\cdot)\}_{l=1}^d$ of value functions, such that $z_j(\cdot) \in \mathbb{Q}_h$ satisfying the basis update rule in (14)*

$$z_j(\cdot) := \sum_{l=1}^d \mathfrak{a}(v_j, k_l) \, \tilde{q}_l(\cdot), \;\; for \; j = 1, \cdots, d,$$

*is the minimizer to the following problem for every $j = 1, \cdots, d$*

$$\left\| \ell_{(v_j, z_j)}(\cdot) \right\|_{\mathbb{K}_h'} = \min_{q \in \mathbb{Q}_h} \left\| \mathfrak{a}(v_j, \cdot) - \mathfrak{b}(\cdot, q) \right\|_{\mathbb{K}_h'} = \min_{q \in \mathbb{Q}_h} \max_{k \in \mathbb{K}_h} \frac{|\mathfrak{a}(v_j, k) - \mathfrak{b}(k, q)|}{\|k\|_{\mathcal{K}}}. \tag{42}$$

*Proof.* The proof repeats most parts from that of Theorem 4.3 in Appendix D.3. By Lemma D.5, $\mathfrak{b}(\cdot, q) : k \mapsto \mathfrak{b}(k, q)$ is bounded below on the current key space $\mathbb{K}_h$, i.e., $\|\mathfrak{b}(\cdot, q)\|_{\mathbb{K}_h'} \ge c\|q\|_{\mathcal{H}}$ for any $q \in \mathbb{Q}_h$ fixed. Noting that the variational formulation corresponding to the min-max problem in (42) have the same form as the one in (45)–(46) in Lemma D.5 with the right-hand side switched from $\langle \cdot, \cdot \rangle_h$ to $\mathfrak{a}(\cdot, \cdot)$, therefore, we conclude that the following operator equation associated with the right-hand side of (42) has a unique solution $z_j \in \mathbb{Q}_h$:

$$\begin{cases} \langle s, k \rangle_h + \mathfrak{b}(k, z_j) = \mathfrak{a}(v_j, k), & \forall k \in \mathbb{K}_h, \\ \mathfrak{b}(s, q) = 0, & \forall q \in \mathbb{Q}_h. \end{cases} \tag{43}$$

It is straightforward to verify that $h^m (K^\top V)_{ij} = \mathfrak{a}(v_j, k_i)$ thus the matrix associated with $\mathfrak{a}(\cdot, \cdot)$ for all $\{v_j(\cdot)\}_{j=1}^d$ is $h^m(K^\top V)$. Consequently, following the last part of the proof of Theorem 4.3 and writing in the form of scaled dot-product, we let $\mathbf{z} \in \mathbb{R}^{n \times d}$ be the latent representation with columns being $\{z_j(\cdot)\}_{j=1}^d$ at the grid points, i.e., $z_j(x_i) = \mathbf{z}_{ij}$, $1 \le i \le n$, $1 \le j \le d$, and carry over identical definitions for all matrices involved with the ones used in (40),

$$\mathbf{z} = h^m QU\widetilde{W}^Q(K^\top V), \;\; \text{where } \widetilde{W}^Q := (QU\Lambda)^\top (VM^{-1}) \in \mathbb{R}^{d \times d}.$$

The corollary is proved by setting $\{\tilde{q}_l(\cdot)\}_{l=1}^d$ as functions with DoFs being columns of $QU\widetilde{W}^Q$. $\square$

**Heuristics on learning the latent representations.** In the dynamical basis update interpretation above, the scaled dot-product attention is capable to bring the latent representation of query as close as possible to that of values in a functional distance, which is $\|\mathfrak{a}(v_j, \cdot) - \mathfrak{b}(\cdot, q)\|_{\mathbb{K}_h'}$ in (42).

Heuristically speaking in each encoder layer, an ideal operator learner could learn a latent subspace on which the input (query) and the target (values) are "close", and this closeness is measured by how they respond to a dynamically changing set of basis (keys).

## D.4 Interpretations, remarks, and possible generalizations

**Inspirations from finite element methods.** The first part of the proof for Theorem 4.3 follows closely to the finite element methods of the velocity-pressure formulation of a stationary 2D Stokesian flow [13, Lemma 12.2.12], where the key is to establish the solvability of an inf–sup problem (min–max in our finite dimensional setting). The $n$-independent lower bound of $\mathfrak{b}(\cdot, q)$ plays a key role in the convergence theory of traditional finite element methods (see the remarks in Appendix D.2.1–D.2.2). In our interpretation of the scaled dot-product attention, this independence is essentially consequent on the diagonalizability of normalizations of the two matrices in this product (cf. the discussion in Appendix D.4.4; see also the proof in Lemma D.6). The singular values of the attention matrix should be independent of the sequence length if the lower bound of $\mathfrak{b}(\cdot, q)$ is to be verified with a constant independent of $n$. The presence of softmax renders this verification impossible (see Remark D.7).

In traditional finite element methods [13, Chapter 12], the dimension of the approximation subspaces are usually tied to the geometries (discretization, mesh), for example, $\dim \mathbb{V}_h = 2n = 2(\#\text{grid points})$, and $\dim \mathbb{Q}_h = \#\text{triangles} \simeq O(n)$. Inspired by [13, Lemma 12.2.12], the introduction of the bilinear form $\mathfrak{b}(\cdot, \cdot) : \mathcal{V} \times \mathcal{Q} \to \mathbb{R}$ here in Theorem 4.3 is to cater the possibility that the column spaces of $V$ and $Q$ may represent drastically different functions.

In our proof, we have shown that, using small subspaces (dimension $d$ and $r$) of these complete finite element approximation spaces (dimension $n \gg d \geq r$) suffices to yield the best approximator if this key-to-value map exists. This fits perfectly into our setting to use merely the column space of $Q$ to build the degrees of freedom (DoF) for the "value" functions: the DoF functional is simply a function integrating against the function which is discretized at the grid points, i.e., a dot-product in the sequence-length dimension represents an integral. Recently, we also become aware that the CV and NLP communities begin to exploit this topological structure in the feature (channel) dimension by treating the vector in the same feature dimension as a function to exert operations upon, instead of mainly relying on position-wise operations, see e.g., the work in [83, 54] essentially acknowledges Assumption 4.2 implicitly.

It is also worth noting that, the subscript $h$ in the approximation subspaces is a choice of the exposition of the proof to facilitate the notion that "dot product $\approx$ integral". In computations, unlike finite element methods where the basis functions are locally supported, the learned basis functions are globally supported, dynamically updated, and not bound to a fixed resolution. As such, numerically the approximation subspaces are essentially mesh-free (e.g., see (14), (27), and Figure 13), how this nature attributes to the capability of zero-shot super-resolution of kernelizable architectures (e.g., [57]) will be an interesting future study.

**Difference with universal approximation theorems.** In the era of using a nonlinear approximator such as a deep neural network to approximate functions/operators, following the discussion in Section D.2, the "consistency" of an approximation class embodies itself as the Universal Approximation Theorem (UAT, e.g., see [42, 18], and [58, 67, 77, 99] for modern expositions). A UAT roughly translates to "a $d$-layer ($d \geq 2$) neural network with certain nonlinear activation can approximate any compactly supported continuous function", with the approximation error depending on the width and the depth of a network.

In our work, Theorem 4.3 is in its spirit a "stability" result of the attention-based learner. Under this fixed general dot-product attention architecture, Theorem 4.3 tries to bridge the approximation capacity of a nonlinear operator (hard to quantify) to a linear one (easy to find). To further give a more refined bound in terms of the architectural hyperparameters (number of layers, number of learned basis, etc.), one shall invoke a UAT on top of the result in Theorem 4.3. Theorem 4.3 states that the attention mechanism, in terms of architecture, allows practitioners to exploit simple non-learnable linear approximation results as well, for example the number of Fourier bases that one chooses to build the "value" space.

**PDE-inspired attention architectures.** The lower bound of $\mathfrak{b}(\cdot, q)$ gives a theoretical guideline on how to design a new dot-product attention between compatible subspaces. For example, we can use the inter-position differences in certain direction (flux) as query to test against the key, which may provides more information on how to choose the best value. This aforementioned "differential attention" corresponds to $\langle q, \operatorname{div} \boldsymbol{v} \rangle$ in the velocity-pressure formulation of the Stokesian flow.

In the proof, we assume nothing such as $\mathbb{Q}_h \subset \mathbb{V}_h$ but only bridge them through a value-to-key map. The lower bound of this value-to-key map using the simple scaled dot-product is verified in Lemma D.6. In general, we have two guidelines: (1) the scaling shall be chosen such that the lower bound (50) is independent of the sequence length; (2) the key space $\mathbb{V}_h$ (functions to test the responses against) is bigger than the value space $\mathbb{Q}_h$, which contains functions to approximate the target or to form a better set of latent basis. In practice, for example, $Q, V \in \mathbb{R}^{n \times r}, \mathbb{R}^{n \times d}$ with $d \geq r$ in [19]. We remark that the proof is done for $\dim \mathbb{V}_h = d$, but in general it applies to $\dim \mathbb{V}_h \leq d$ as the final block form matrices (39) can be easily adjusted.

From the proof, it is also straightforward to see that the columns of $Q, K, V$ merely act as the degrees of freedom representations associated with $x_i$ or $\xi_i$, thus not necessarily the pointwise value at $x_i$ or $\xi_i$. The essential requirement is that, there exists two sets of locally supported nodal basis functions in $\mathbb{Q}_h$ and $\mathbb{V}_h$ associated with $x_i$ or $\xi_i$ to build the globally supported learned bases. These nodal basis functions are assumed to be supported in an $O(h^m)$-neighborhood of $x_i$ or $\xi_i$, which implies the sparse connectivity of the grids in the discretization. Here the degrees of freedom being the pointwise values is assumed merely for simplicity, thus imposing $\mathcal{H} \hookrightarrow C^0(\Omega)$ and $\mathcal{V} \hookrightarrow C^0(\Omega^*)$ to ensure the existence of the pointwise values is just a technicality and not essential. As a result, this broadens the possibility of a more structure-preserving feature map. For example, the bilinear form $\mathfrak{b}(\cdot, \cdot)$ can incorporate information from the higher derivatives, DoFs for splines, etc.; another example is that a single entry in a column of $Q, K, V$ may stand for an edge DoF vector representation in a graph, and the edge–edge interaction is extensively studied for the simulation of Maxwell's equations [63, 15, 6] on manifolds [94].

**Galerkin-type layer normalization scheme.** In the proposed Galerkin-type attention (6), the layer normalization is pre-dot-product but post-projection; cf. the pre-LN scheme in [96] is pre-dot-product and pre-projection. From the proof of Theorem 4.3, it is natural to impose such a normalization from a mathematical perspective. As in (40), the updated $\widetilde{K}$ and $\widetilde{V}$ have $(BM^{-1}B^\top)^{-1}$ and $M^{-1}$ as their normalizations, respectively. While $M$ is the Gram matrix for $V$, the inverse of $M$'s Schur complement can be understood as the inverse of the Riesz map from $Q$ to $V$. Since a given layer normalization module is shared by every position of a latent representation upon which it is exerted, the weight acting as the variance (in $\mathbb{R}^d$) in the layer normalization acts a cheap learnable diagonal matrix approximation to $M^{-1} \in \mathbb{R}^{d \times d}$. This offers a reasonable approximation if one assumes the self-similarity of the bases $\|v_j\|_\mathcal{V}^2$ outweighs the inter-basis inner product $\langle v_i, v_j \rangle$ for $i \neq j$.

## D.5 Auxiliary results

**Lemma D.3.** *Consider $\Omega^* \subset \mathbb{R}^m$ a bounded domain, we assume $(\mathcal{V}, \langle \cdot, \cdot \rangle_\mathcal{V}) \hookrightarrow C^0(\Omega^*)$, where $\langle u, v \rangle_\mathcal{V} := \int_{\Omega^*} u(\xi) v(\xi) \, d\xi$. $\Omega$ is discretized by $\{\xi_i\}_{i=1}^n$. $\mathbb{Y}_h \subset \mathcal{V}$ is an approximation space defined on $\{\xi_i\}_{i=1}^n$ and $\mathbb{Y}_h \simeq \mathbb{R}^n$. $\mathbb{Y}_h = \mathrm{span}\{\phi_{\xi_1}, \cdots, \phi_{\xi_n}\}$ such that the degree of freedom for the i-th nodal basis is defined as $\chi_{\xi_i}(v) := v(\xi_i)$. $\mathbb{V}_h = \mathrm{span}\{v_1 \cdots, v_d\}$ for $d < n$ with $d$ linearly independent $v_j(\cdot) \in \mathbb{Y}_h$. Then, $\langle \cdot, \cdot \rangle_h : \mathcal{V} \times \mathcal{V} \to \mathbb{R}$ defines a continuous inner product in $\mathbb{V}_h$, where*

$$\langle u, v \rangle_h := h^m \sum_{i=1}^n u(\xi_i) v(\xi_i), \quad \text{for } u, v \in \mathbb{V}_h. \tag{44}$$

*Proof.* First we show that $\langle v, v \rangle_h = 0$ implies that $v \equiv 0$. By (44), obviously this $v$ satisfies $v(\xi_i) = 0$ for $i = 1, \cdots, n$. Now expanding $v$ in $\{\phi_{\xi_i}\}_{i=1}^n$ we have

$$v(\cdot) = \sum_{i=1}^n \chi_{\xi_i}(v) \phi_{\xi_i}(\cdot) = \sum_{i=1}^n v(\xi_i) \phi_{\xi_i}(\cdot) \equiv 0.$$

Thus, $\| \cdot \|_{\mathbb{V}_h}^2 := \langle \cdot, \cdot \rangle_h$ defines a norm, and it is equivalent to $\| \cdot \|_{L^2(\Omega^*)}$ restricted on $\mathbb{V}_h$ due to being finite dimensional. The desired result follows from applying the Cauchy-Schwarz inequality.

$\square$

**Lemma D.4.** *If $r \leq d$, $B \in \mathbb{R}^{r \times d}$ has full row rank, $G \in \mathbb{R}^{d \times d}$ is symmetric positive definite, then $BGB^\top \in \mathbb{R}^{r \times r}$ is invertible.*

*Proof.* With $G$ being symmetric positive definite, upon applying a Cholesky factorization $G = CC^\top$, we have $BGB^\top = (BC)(BC)^\top$. Now it is straightforward to see that $\mathrm{rank}(BC) = r$ as $\mathrm{rank}(BC) \leq \min\{r, d\}$ and $\mathrm{rank}(BC) \geq r$ by the Sylvester's inequality. Applying the same argument on the product of $BC$ with $(BC)^\top$ yields the desired result of $\mathrm{rank}(BGB^\top) = r$. $\qquad\square$

**Lemma D.5.** *Under the same setting with Theorem 4.3 (Assumption D.1), given a $u \in \mathbb{V}_h$, looking for a saddle point of*

$$\min_{q \in \mathbb{Q}_h} \max_{v \in \mathbb{V}_h} \frac{|\langle u, v \rangle_h - \mathfrak{b}(v, q)|}{\|v\|_\mathcal{V}} \tag{45}$$

*is equivalent to solving the following operator equation system: find $p \in \mathbb{Q}_h, w \in \mathbb{V}_h$*

$$\begin{cases} \langle w, v \rangle_h + \mathfrak{b}(v, p) = \langle u, v \rangle_h, & \forall v \in \mathbb{V}_h, \\ \mathfrak{b}(w, q) = 0, & \forall q \in \mathbb{Q}_h. \end{cases} \tag{46}$$

*It is further equivalent to solve the following linear system if $\{v_j(\cdot)\}_{j=1}^d$ and $\{q_j(\cdot)\}_{j=1}^r$ form sets of basis for $\mathbb{V}_h$ and $\mathbb{Q}_h$, respectively:*

$$\begin{pmatrix} M & B^\top \\ B & 0 \end{pmatrix} \begin{pmatrix} \boldsymbol{\mu} \\ \boldsymbol{\lambda} \end{pmatrix} = \begin{pmatrix} \boldsymbol{\zeta} \\ 0 \end{pmatrix}, \tag{47}$$

*where $M \in \mathbb{R}^{d \times d}$ with $M_{ij} = \langle v_j, v_i \rangle_h$. $B \in \mathbb{R}^{r \times d}$ with $B_{ij} = \mathfrak{b}(v_j, q_i)$. $\boldsymbol{\zeta} \in \mathbb{R}^d$ with $(\boldsymbol{\zeta})_j = \langle u, v_j \rangle_h$. For $w(\cdot) \in \mathbb{V}_h$, its vector representation is $\mathbb{R}^d \ni \boldsymbol{\mu} := \boldsymbol{\mu}(w) = (\mu_{v_1}(w), \dots, \mu_{v_d}(w))^\top$ for $w(\cdot) = \sum_{j=1}^d \mu_{v_j}(w) v_j(\cdot)$. Similar notion applies to $\boldsymbol{\lambda} := \boldsymbol{\lambda}(p) = (\lambda_{q_1}(p), \cdots, \lambda_{q_r}(p))^\top \in \mathbb{R}^r$ being the vector representation for $p(\cdot) = \sum_{j=1}^r \lambda_{q_j}(p) q_j(\cdot)$ where $p(\cdot) \in \mathbb{Q}_h$.*

*Proof.* This lemma is a finite dimensional rephrasing of the commonly-known variational formulation of a saddle point problem in infinite dimensional Hilbert spaces [33, Chapter I § 4.1], for completeness we include the proof here for the convenience of readers.

Define $\eta(\cdot) : \mathbb{V}_h \to \mathbb{R}$ such that $\eta(v) := \langle u, v \rangle_h - \mathfrak{b}(v, q)$, using Lemma D.3 and the assumptions $(D_{11})$ $(D_{12})$ on $\mathfrak{b}(\cdot, \cdot)$ in Assumption D.1, then clearly $\eta \in \mathbb{V}_h'$ for $(\mathbb{V}_h, \langle \cdot, \cdot \rangle_h)$. By Riesz representation theorem, there exists an isomorphism $R : \mathbb{V}_h \to \mathbb{V}_h'$ such that $w := R^{-1}(\eta) \in \mathbb{V}_h$ and

$$\eta(v) = \langle w, v \rangle_h = \langle u, v \rangle_h - \mathfrak{b}(v, q). \tag{48}$$

Then, (45) is equivalent to find the minimizer for the following problem, define $\|\cdot\|_{\mathbb{V}_h}^2 := \langle \cdot, \cdot \rangle_h$:

$$\min_{q \in \mathbb{Q}_h} \|\eta(\cdot)\|_{\mathbb{V}_h'}^2 = \min_{q \in \mathbb{Q}_h} \|w\|_{\mathbb{V}_h}^2 = \min_{q \in \mathbb{Q}_h} \|R^{-1}(\langle u, \cdot \rangle_h - \mathfrak{b}(\cdot, q))\|_{\mathbb{V}_h}^2 =: \min_{q \in \mathbb{Q}_h} J(q).$$

Taking the Gateaux derivative $\lim_{\tau \to 0} \mathrm{d}J(p + \tau q)/\mathrm{d}\tau$ in order to find the critical point(s) $p \in \mathbb{Q}_h$, we have for any perturbation $q \in \mathbb{Q}_h$ such that $p + \tau q \in \mathbb{Q}_h$

$$0 = \lim_{\tau \to 0} \frac{d}{d\tau} \left\langle R^{-1}(\langle u, \cdot \rangle_h - \mathfrak{b}(\cdot, p + \tau q)), R^{-1}(\langle u, \cdot \rangle_h - \mathfrak{b}(\cdot, p + \tau q)) \right\rangle_h,$$

and applying $R^{-1}$ on $\mathfrak{b}(\cdot, p) \in \mathbb{V}_h'$, it reads for any $q \in \mathbb{Q}_h$

$$\left\langle R^{-1}(\langle u, \cdot \rangle_h - \mathfrak{b}(\cdot, q)), R^{-1}(\mathfrak{b}(\cdot, q)) \right\rangle_h = \left\langle w, R^{-1}(\mathfrak{b}(\cdot, q)) \right\rangle_h = \mathfrak{b}(w, q) = 0. \tag{49}$$

Thus, (48) and (49) form the desired system (46). Lastly, applying an expansion of $w$ and $p$ in $\{v_i(\cdot)\}_{i=1}^d$ and $\{q_i(\cdot)\}_{i=1}^r$ with degrees of freedom $\mu_{v_j}(\cdot)$ and $\lambda_{q_j}(\cdot)$ respectively, and choosing the test functions to be each $v = v_j$ for $j = 1, \cdots, d$ and $q = q_j$ for $j = 1, \cdots, r$, yield the system (47) with $i$ and $j$ representing the row index and column index, respectively. $\qquad\square$

**Lemma D.6** (verification of the lower bound of $\mathfrak{b}(\cdot, p)$)**.** *Under the setting of Assumption D.1, for any given $p \in \mathbb{Q}_h$, we have*

$$c\|p\|_\mathcal{H} \leq \max_{w \in \mathbb{V}_h} \frac{|\mathfrak{b}(w, p)|}{\|w\|_\mathcal{V}}, \tag{50}$$

*where the constant $c = c_V^{-1} c_Q^{-1} \min_{j=1,\dots,r} |\sigma_j|$. $\{\sigma_j\}_{j=1}^r$ are the singular values of the matrix $B \in \mathbb{R}^{r \times d}$ with $B_{ij} = \mathfrak{b}(v_j, q_i)$ under the sets of basis $\{v_j(\cdot)\}_{j=1}^d$ and $\{q_j(\cdot)\}_{j=1}^r$. $c_V, c_Q$ are the norm equivalence constants between functions $w(\cdot) \in \mathbb{V}_h, p(\cdot) \in \mathbb{Q}_h$ and their vector representation*

*using the DoF functionals $\boldsymbol{\mu}(w)$ and $\boldsymbol{\lambda}(p)$ defined in Lemma D.5 under the sets of basis $\{v_j(\cdot)\}_{j=1}^d$ and $\{q_j(\cdot)\}_{j=1}^r$: for any $w \in \mathbb{V}_h$, and any $p \in \mathbb{Q}_h$, it is assumed that the following norm equivalence holds*

$$\|w\|_{\mathcal{V}} \leq c_V \|\boldsymbol{\mu}(w)\| \quad \text{and} \quad \|p\|_{\mathcal{H}} \leq c_Q \|\boldsymbol{\lambda}(p)\|. \tag{51}$$

*Proof.* The proof is straightforward by exploiting the singular value decomposition (SVD) to the matrix representation of the bilinear form $\mathfrak{b}(\cdot, \cdot)$. When $r \leq d$, $\mathrm{rank}(B) = r$. Consider an SVD of $B$: for $D = (\mathrm{diag}\{\sigma_1, \cdots, \sigma_r\}, 0)$ with $\sigma_j \neq 0$, $U_Q, U_V$ being orthonormal,

$$B = U_Q D U_V^\top, \quad \text{where } U_Q \in \mathbb{R}^{r \times r}, D \in \mathbb{R}^{r \times d}, \text{ and } U_V \in \mathbb{R}^{d \times d}.$$

Hence, $\mathfrak{b}(w, p)$ can be equivalent written as the following for $\boldsymbol{\mu} := \boldsymbol{\mu}(w)$ and $\boldsymbol{\lambda} := \boldsymbol{\lambda}(q)$

$$\mathfrak{b}(w, p) = \mathfrak{b}\left(\sum_{j=1}^d \mu_{v_j}(w) v_j(\cdot), \sum_{j=1}^r \lambda_{q_j}(p) q_j(\cdot)\right) = \boldsymbol{\lambda}^\top B \boldsymbol{\mu} = (U_Q^\top \boldsymbol{\lambda})^\top D (U_V^\top \boldsymbol{\mu}).$$

By the norm equivalence (51) and above,

$$\max_{w \in \mathbb{V}_h} \frac{|\mathfrak{b}(w, p)|}{\|w\|_{\mathcal{V}}} \geq c_V^{-1} \max_{\boldsymbol{\mu} \in \mathbb{R}^d} \frac{\left|(U_Q^\top \boldsymbol{\lambda})^\top D (U_V^\top \boldsymbol{\mu})\right|}{\|\boldsymbol{\mu}\|} = c_V^{-1} \max_{\boldsymbol{\mu} \in \mathbb{R}^d} \frac{\left|(U_Q^\top \boldsymbol{\lambda})^\top D (U_V^\top \boldsymbol{\mu})\right|}{\|U_V^\top \boldsymbol{\mu}\|}.$$

Since $U_V^\top$ is surjective, we choose a specific $U_V^\top \boldsymbol{\mu} \in \mathbb{R}^d$ to pass the lower bound: let the first $r$ entries of $U_V^\top \boldsymbol{\mu}$ be $U_Q^\top \boldsymbol{\lambda}$, we have

$$\begin{aligned}
c_V^{-1} \max_{\boldsymbol{\mu} \in \mathbb{R}^d} \frac{\left|(U_Q^\top \boldsymbol{\lambda})^\top D (U_V^\top \boldsymbol{\mu})\right|}{\|U_V^\top \boldsymbol{\mu}\|} &\geq c_V^{-1} \frac{\left|(U_Q^\top \boldsymbol{\lambda})^\top \mathrm{diag}\{\sigma_1, \cdots, \sigma_r\}(U_Q^\top \boldsymbol{\lambda})\right|}{\|U_Q^\top \boldsymbol{\lambda}\|} \\
&\geq c_V^{-1} \min_{j=1,\cdots,r} |\sigma_j| \|U_Q^\top \boldsymbol{\lambda}\| = c_V^{-1} \min_{j=1,\cdots,r} |\sigma_j| \|\boldsymbol{\lambda}\| \geq c_V^{-1} c_Q^{-1} \min_{j=1,\cdots,r} |\sigma_j| \|p\|_{\mathcal{H}}.
\end{aligned} \tag{52}$$

$\square$

**Remark D.7** (Constants $c_V, c_Q$ and the potential impact of softmax thereon)**.** *The norm equivalence constants bridging the integral-based $\|\cdot\|_{\mathcal{H}}$ and the $\ell^2$-norm $\|\cdot\|$ depend on the topology of the approximation spaces.*

*If the basis functions are globally supported, such as the Fourier-type basis from the eigenfunctions of the self-adjoint operator, or the learned bases shown in Figure 13, orthonormal, and defined on the same discretization on the spacial domain, then it is easy to see that $c_V$ and $c_Q$ are approximately 1 due to the Parseval identity, minus the caveat of approximating an integral on a discrete grid.*

*Even if the basis functions $\{v_j(\cdot)\}$ and $\{q_j(\cdot)\}$ for $\mathbb{V}_h$ and $\mathbb{Q}_h$ are locally supported, such as the nodal basis in $(D_2)$ and $(D_4)$, the $h^m$-weight in (44) will make $c_V$ and $c_Q$ be of $O(h^{m/2})$, or the inverse square root of the sequence length $1/\sqrt{n} = O(h^{m/2})$, see e.g., [98, Section 11]. Nevertheless, the final bound (50) will be sequence length-independent because now the minimum singular value of $B$ will scale as $O(h^m)$ the same with a mass matrix in the finite element method (see e.g., [29, Section 4.4.2]). Consequently, these two constants depend on the number of explicit connections (not learned) that a single position has in the discretization. In our examples, the Euclidean coordinate positional encodings yield a sparse connection (tri-diagonal in 1D, 5-point stencil in 2D) thus independent of $n$.*

*If a softmax normalization $\mathcal{S}(\cdot)$, such as the one in [76], is applied on the key matrix in the sequence length dimension, the norm equivalence constant $c_V$ is not $n$-independent anymore. To illustrate this, without loss of generality, let $\mathcal{V} \subset \mathcal{H} = L^2(\Omega)$, $\Omega = \Omega^* = (0, 1)$. By the definition of softmax, it is straightforward to verify that the test functions essentially change to $\widetilde{w} := \mathcal{S}(w) \approx \left(n \int_\Omega e^w dx\right)^{-1} e^w$. Following the argument of Lemma D.6, the lower bound can be proved for $\widetilde{w}$, i.e., $\max_{w \in \mathbb{V}_h} |\mathfrak{b}(w, p)|/\|\widetilde{w}\|_{\mathcal{H}} \geq c \|p\|_{\mathcal{H}}$. However, by an exponential Sobolev inequality [32, Theorem 7.21] if we further assume that $w$ has a weak derivative with a bounded $L^1(\Omega)$-norm, $\|w\|_{\mathcal{H}} \leq c_\Omega n \|\widetilde{w}\|_{\mathcal{H}}$, thus passing the inequality to try to obtain an estimate like (50) yields an inevitable constant related to $n$.*