# OpenReview forum: "Choose a Transformer: Fourier or Galerkin"
_NeurIPS.cc/2021/Conference — NeurIPS 2021 Poster_

### Official Review · Reviewer_wufR · 2021-07-12

**Rating:** 6
**Confidence:** 3

**Summary:**

This paper proposes two Transformer variants for data-driven operator learning problems of parametric partial differential equations: the Fourier-type and the Galerkin-type. This paper includes both theoretical evidence and empirical evidence to show that these two Transformer variants are both accurate and efficient for solving various parametric PDEs.

**Main Review:**

This paper first generalizes the Transformer architecture and then shows that vanilla Transformer, Fourier-type Transformer, Galerkin-type Transformer are special cases of this generalized Transformer architecture. This generalization is natural but creative. The author uses informative diagrams and equations to illustrate different kinds of Transformers talked about in this paper clearly.

The evaluation is on three well-known PDE benchmarks. This paper compares the error of all different combinations of Layer Normalization, so the experiment section is comprehensive and solid.

Although the variants seem to be inspired by the mathematical representations of some integrations, the proposed variants have only a few differences from the standard Transformer in the source code. I suggest that the author can list all these differences explicitly to make them clear. To my understanding, the differences only lie in three places:

1. Use mesh-weighted normalization without softmax
2. Where to add layer normalizations are different
3. (For Galerkin-type Transformer only) the ordering of matrix multiplication is different

The author does not explain why the speed/memory of ST, FT, LT, and GT are different in Table 1. To my understanding, LT and GT are faster than ST and FT because they are linear, but why is FT faster than ST, GT faster than LT? It would be better if there are more detailed explanations (maybe manually calculated FLOPs/memory).

To summarize, the motivation of this paper is interesting, but the real difference of the proposed methods with standard Transformer is limited and is not shown clearly. Shown by solid experimental results, the proposed variants are both accurate and efficient in solving PDEs, but the efficiency difference requires more explanations.

**Time Spent Reviewing:**

4

---

> ### Author Response · Authors · 2021-08-07
> **Full response to Reviewer wufR Part 2**
>
> (continued from part 1)
>
> ### Difference 5: tweak the Xavier initialization
> The newly proposed attention
> $$
> \mathbf{z} =  \mathbf{y} + \operatorname{Attn}\_{\dagger} (\mathbf{y}), \quad
> \mathbf{y} \mapsto \mathbf{z} +  \sigma(\mathbf{z})
> $$
> resembles something like a LeapFrog integrator scheme
> $$
> \mathbf{y}\_{k+1/2}\gets \mathbf{y}\_k + \Delta t \operatorname{Attn}\_{\mathfrak{g}} (\mathbf{y}\_k ),
> \quad
> \mathbf{y}\_{k+1} \gets \mathbf{y}\_{k+1/2} + \Delta t \sigma(\mathbf{y}\_{k+1/2}; \mathbf{x}).
> $$
> Thus it makes sense to further dial down the initialization of the weight matrices $W^Q, W^K, W^V$ from the default Xavier initialization (equation (18) in the full paper) to achieve that step size $\Delta t$ effect. For the difference this makes, please refer to Table 9 in the Appendix of the full paper: without the tweak, simply removing the softmax and using the default Xavier initialization, the model's evaluation accuracy is 200 times worse than the current Galerkin Transformer's (with tweak number 6).
>
> ----
>
> ### Difference 6: adding a small positive diagonal bias to the projection matrix
> Inspired by the proof of Theorem 3.3 (equation (34) and (35) to be specific), we have added a small positive deterministic diagonal to the initialization of $W^Q, W^K, W^V$ to make these matrices diagonally dominant. Its impact on training stability can be found in Table 9.
>
> To get the heuristics, let us use the following example to illustrate: consider the domain $\Omega=(-1,1)$, discretized uniformly by $-1=x_1<x_2<\cdots<x_n=1$.
>
> In the context of any linear variant of attentions, $Q$ stands for values, $K$ for query, and $V$ for keys. Let us assume that $Q$'s column approximation space is made by the first two Chebyshev polynomials $\\{1, x\\}$ for $x\in \Omega$, and the approximation spaces of $K$ and $V$'s are $\\{a, bx\\}$ and $\\{c, dx\\}$ for $a,b,c,d\in \mathbb{R}$ as learnable parameters.
>
> The actual columns of $Q,K,V$ are made by these functions' evaluation at $\{x_i\}$'s. Consider a function $f\in \mathcal{H} \subset L^2(\Omega)$, the Galerkin projection in $L^2$-norm of $f$ onto $\mathbb{Q}_h := \operatorname{span}\\{1, x\\}$ is:
>
> $$
> \Big(\text{Proj} f \Big)(x) := p\_0 + p\_1 x, \quad
> \text{ where } \quad p\_0 = \frac{1}{2} \int_{-1}^1 f(\xi)\mathrm{d}\xi, \\; \text{ and }\\; p\_1 = \frac{3}{2}\int_{-1}^1 \xi f(\xi)\mathrm{d}\xi
> \tag{1}
> $$
>
> If we interpret the columns of $K$ and $V$ as functions sampled at grids, the dot-product of $K^TV/n$ becomes a Riemann sum approximation to an integral and it is easy to verify:
>
> $$
> K^TV/n \approx
> \begin{pmatrix}
> \int^{1}\_{-1}ac\, \mathrm{d}x & 0 \\\\ 0 & \int^{1}\_{-1}bd x^2 \mathrm{d}x
> \end{pmatrix},
> $$
>
> In this case, upon a further simple check we can see that $K^TV/n$ has capacity to replicate the coefficients for a projection in (1) by just multiplying with a vector.
>
> However, what if $K$ and $V$'s columns are changed to the evaluations of $\\{a, bx\\}$ and $\\{cx, dx\\}$ at the grid points? then
>
> $$
> K^TV/n \approx \begin{pmatrix}
> 0 & 0 \\\\ \int^{1}\_{-1}bc x^2 \mathrm{d}x & \int^{1}\_{-1}bd x^2 \mathrm{d}x
> \end{pmatrix},
> $$
>
> Suddenly, the capacity to replicate the coefficients in the Galerkin projection (1) is gone! Because after multiplying this matrix to $Q$: the subspace represented by the columns of $Q(K^TV)$ has no constant function in it!
>
> In the first case, the optimal approximation capacity is achieved for the first two Chebyshev polynomials:
>
> $$
> W^Q = \begin{pmatrix} 1&0 \\\\ 0 & 1 \end{pmatrix},
> \quad W^K = \begin{pmatrix} a&0 \\\\ 0 & b \end{pmatrix}
> \quad \text{ and}\quad  \color{blue}{W^V = \begin{pmatrix} c&0 \\\\ 0 & d \end{pmatrix}},
> $$
>
> In the second case where the capacity to deliver the best approximation (Galerkin projection) is lost:
>
> $$
> W^Q = \begin{pmatrix} 1&0 \\\\ 0 & 1 \end{pmatrix},
> \quad W^K = \begin{pmatrix} a&0 \\\\ 0 & b \end{pmatrix}
> \quad \text{ and}\quad  \color{red}{W^V = \begin{pmatrix} 0&0 \\\\ c & d \end{pmatrix}},
> $$
>
> The difference? $W^V$ is not full rank. Therefore, intuitively, we have made the initialization of $W^Q, W^K, W^V$ diagonally dominant (equation (18) in the full paper), to let the optimization take over if a good set of basis being found (warm-up phase in the `1cycle`).
>
> This concludes the major and minor changes to the attention operator we have made, the line numbers in the beginning are referring to the ones in the full version of the paper in the supplementary material.
>
>
> -----
>
> ### Other rebuttals
>
> >  The proposed variants have only a few differences from the standard Transformer in the source code. To summarize, the motivation of this paper is interesting, but the real difference of the proposed methods with standard Transformer is limited and is not shown clearly.
>
> - As the reviewer suggested, the differences are few, and the changes are simple. The changes are so simple that our existing code can be enjoyed by the practitioners in community immediately, since their code base needs only some small effortless changes.
>
>     We have mentioned historically researchers had been seeking ways to remove softmax (or exponential-based nonlinearity) from `seq2seq` models such as NMT (Ref. [17] in the paper), but none of the attempts had succeeded in terms of surpassing the softmax ones.
>
>     In our humble opinion, small changes resulting huge gain shows the exciting magic of mathematics. It also tells the story of how much further potential the attention mechanism has.
>
> ----
>
> > It would be better if there are more detailed explanations (maybe manually calculated FLOPs/memory)
>
> - FLOPS/memory is not a measure of the efficiency of an algorithm, it measures how efficient a processor/memory architecture is: how many FLOPS can a processor pull off under the constraint of memory.
>
>     We do appreciate the reviewer brought this up, and now we use simply the FLOP (without that "per second") measures how efficient an algorithm finishes a task. To incorporate hardware utilization bias and possible fluctuations, the speed column in Table 1 is replaced by "number of floating point operations per backpropagation iteration" (or FLOPBI, in GFLOP, the lower the faster per iteration) recorded by the CUDA profiler as follows. This is the third column in Table 1 (encoder only performance). The updated source code is here: [please download `utils_ft.py` and `encoder_memory_profile.py` to replace the original ones in the supplemental material](https://www.dropbox.com/sh/3cip8ozjjjjb9rl/AAAdxa58GomMxMnYZp-h5qxVa?dl=0).
>
>     |        | FT 	| ST | GT	| LT |
>     |------- |---	|--- |---   |--- |
>     | FLOPBI |  1609.57$\pm$ 2.44 | 1876.36 $\pm$ 2.01 | 411.78 $\pm$ 1.83   | 772.66 $\pm$ 1.53
>
>     For the actual computational complexity count for the scaled dot-product with or without softmax, please still refer to the Table in our full response Part 1.

---

> > ### Comment · Reviewer_wufR · 2021-08-18
> > **Official Comment**
> >
> > Thank you for explaining all these differences between the proposed method and the standard Transformer, and for pointing out the complexity of the proposed method. The author's feedback successfully addressed most of the concerns in my review, so the score has been adjusted from 4 to 6. If this paper is accepted, I suggest that the author could include these explanations in the camera-ready version or the appendix.

---

> > > ### Author Response · Authors · 2021-08-18
> > > **Reply to Reviewer wufR's score update**
> > >
> > > We greatly appreciated that the reviewer has given us another chance.
> > >
> > > Inspired by by Reviewer wufR's comments (along with other other reviewer's comments), we have already incorporated most changes in the draft, including
> > > - In Contributions, we have highlighted the differences between the newly proposed simple attention without softmax with the one in the classical Transformer, and mentioned the gain in terms of computational efficiency.
> > > - In Appendix, we have added a table comparing the computational complexity of the forward propagation of the attention operator with and without softmax, and a remark about the memory requirement of softmax in the backpropagation.
> > > - In Numerical experiments section, we have replaced the "time per backprop iter" by "floating point operations per backprop iter".
> > >
> > > We plan to add a remark in Appendix D of the reason why we have introduced difference 6 mentioned above in a fashion with more mathematical rigor:
> > > - Explain why equations (34) and (35) in the current draft imply that this numerical trick introduced in equation (18) would greatly improve the accuracy for the attention mechanism. A generalized invertibility of the $W^{Q}, W^{K}, W^{V}$ (in terms of Schur complement argument in the proof) would heuristically preserve the dimension of the current approximation space (the column-rank of the $Q, K, V$), provided that a good set of basis has been trained.

---

> ### Author Response · Authors · 2021-08-07
> **Full Response to Reviewer wufR Part 1**
>
> (expanded from the short response above)
>
> ----
>
> ### Difference 1: No softmax, or the approximation thereof, at all.
> Different with all previous contributions to the attention-like architectures, the conventional "row-wise" or "position-wise" interpretation is changed to a "column-wise" interpretation. Each column of the latent representation is treated as Hilbert space functions sampled at discrete grids.
> As a result, the matrix product can be viewed as quadrature approximations of the integral of functions in Hilbert space. This further facilitates that the linear complexity scaled dot-product attention can enjoy a rigorous mathematical interpretation.$Q(K^TV)$ is written the first time as the solution explicitly to a saddle point optimization problem in Hilbert spaces, thus to deliver the first provable approximation capacity independent of the sequence length with mathematical rigor.
>
> However, simply removing the softmax does give us a nice mathematical interpretation (same form with a learnable Petrov-Galerkin projection), the proposed Galerkin Transformer variant does not work numerically. Table 9 in the Appendix: if we remove the softmax only and does not change other structures, the model's evaluation accuracy is 200 times worse than our final model.
>
> So the story continues to the second difference.
>
> ----
>
> ### Difference 2: a Galerkin projection-inspired layer normalization scheme
> The new way of adding the layer normalization allows scaling factors to be learned and thus be propagated through the layers (line 105), opposed to killing the scaling overall (line 112). This shows great potential in operator learning benchmarks. The way of adding this type of layer normalization is inspired by the Galerkin projection (line 169 in the full paper), if $\boldsymbol{v}$ is seen as a column of $V$, let $\dim \mathbb{Q}\_h = r$, and $\\{\boldsymbol{q}_i\\}$ be a set of orthogonal basis.
>
> $$
> {\mathrm{proj}(\boldsymbol{v} )=\sum_{i=1}^{r} {\frac {\langle \boldsymbol{v} ,\boldsymbol{q}\_i \rangle }{\langle \boldsymbol{q}\_i ,\boldsymbol{q}\_i \rangle }}{\boldsymbol {q}\_i },}
> $$
>
> This is a special case of formula (16) where the value space and the key space are the same. The new layer normalization is inspired by this formula (the denominator). The Fourier series partial expansion or the Gram-Schmidt process also bears this form, since they are essentially Galerkin projections with an orthogonal set of basis functions.
>
> However, in reality, the key and value spaces are not the same, may not have the same dimension, different bases are not orthogonal unless special procedures are performed (e.g., FAVOR+ in [1]), and they may not even lie in the same underlying Hilbert space (think of different languages). To conquer all these difficulties (mentioned in line 169-171), we introduce a completely new tool to this field: the Petrov-Galerkin projection that allows the response testing space and value representation space to be different (Appendix Assumption D1 to D4, and D9).
>
> Despite that we have verbally mentioned the difference of the new way of adding layer normalization vs the conventional one, inspired by the reviewer's insightful comments, as well as Reviewer xknS's suggestion, we plan to move Figure 1 to the Appendix B Network Structure and use the following two: [new scale-preserving Galerkin-type attention operator](https://i.imgur.com/CRB0Xsp.png) vs [the classic softmax attention](https://i.imgur.com/CqOjGmZ.png).
>
> ----
>
> ### Difference 3: Galerkin Transformer has a linear complexity attention operator
> > Galerkin Transformer (GT) has a different ordering of matrix multiplication.
>
> That is absolutely correct. Using this order, our main model GT reduces the usual $\mathcal{O}(n^2 d)$ complexity of the scaled dot-product attention to $\mathcal{O}(nd^2)$ (similar to [2] Section 3.2.1). We plan to add the following table of the attention operator complexity to the Appendix B Network Structures:
>
> - if we only consider the matrix product of $Q,K^T,V$, and assumes an exponential operation has complexity $l\gg 1$, but is uniform with respect to $n$ and $d$, where the $1$ being the complexity of a floating point operation, we have
>
>     |        	     | FT 	| ST | GT	| LT |
>     |-------	     |---	|---	  |---          |---	                 |
>     | Computational complexity  |  $O(n^2 d)$ 	| $O(ln^2d)$  | $O(nd^2)$  | $O(n(d^2+ld))$
>
> Additionally, when building the computational graph for chain rule (`autograd`), the denominator in softmax makes the chain rule's dependence global with respect each entry in the input vector because the quotient rule is applied.
>
> For example, consider the quadratic complexity attention in the classic Transformer [3], consider the first row in the attention matrix $\text{Softmax}(QK^T)$. $z = \text{Softmax}(q_1 K^T)$ and $a:=q_1 K^T$ where $q_1\in \mathbb{R}^{1\times d}$ is the first row of $Q$ and $a, z\in \mathbb{R}^{1\times n}$. Apply Exercise 8.4 in a classical college text book by K.P. Murphy [4], we have for the $j$-th entry of $z$, its derivative with respect to the $i$-th column of $K^T$ (i.e., $i$-row of $K$) is
>
> $$
> \frac{\partial z_j}{ \partial k_i} = z_k (1[j=k] - z_j) q_1
> $$
>
> while
>
> $$
> \frac{\partial a_j}{ \partial k_i} = 1[j=k] q_1
> $$
>
> From above, it is easy to see, if softmax is involved, the auto-differentiation needs to store both the attention matrix itself and the latent representation $Q$; while without it, only the latent representation $Q$ needs to be stored in this specific operation. This storage requirement will be further passed down when we try to take derivative with respect to the weight $W^Q$ from $Q$.
>
> Thanks to the reviewer's suggestion, we plan to include this as a remark near the table above to help more attention practitioners understand the calculus why the saving is huge, thus to reach a broader audience.
>
> ----
>
> ### Difference 4: position encoding determines the topology of the approximation space
>
> Inspired by the Hilbert space operator approximation theory we have introduced, we had the intuition that the approximation is better off if we concatenate the positional encoding (Euclidean coordinates) to every latent representation. The aim is to get a more direct update of the approximation dynamically (Remark 2 under Theorem 3.3 line 195-202 in the full paper, line 756-769 in the full paper). The deeper reason is to get a better stability bound in the Ladyzhenskaya–Babuška–Brezzi inf-sup condition (see Remark D.5 on how the positional encodings affect this constant).
>
> Our claim of the importance of the positional encodings (topology-characterizing) has recently been corroborated in the supplemental material Section 1.8 of the AlphaFold 2 paper [5], where the affine linear Euclidean coordinates (together with a transform to a reference frame) added in multiple places to achieve certain trainable invariance in this coordinate frame.
>
> (full response continues in Part 2)
>
>
> -----
> [1]: Choromanski, Krzysztof, Valerii Likhosherstov, David Dohan, Xingyou Song, Andreea Gane, Tamas Sarlos, Peter Hawkins et al. "Rethinking attention with performers." *International Conference on Learning Representations* (2021). https://openreview.net/forum?id=Ua6zuk0WRH
>
> [2]: Katharopoulos, Angelos, Apoorv Vyas, Nikolaos Pappas, and François Fleuret. "Transformers are rnns: Fast autoregressive transformers with linear attention." In *International Conference on Machine Learning*, pp. 5156-5165. PMLR, 2020.
>
> [3]: Vaswani, Ashish, Noam Shazeer, Niki Parmar, Jakob Uszkoreit, Llion Jones, Aidan N. Gomez, Łukasz Kaiser, and Illia Polosukhin. "Attention is all you need." In *Advances in neural information processing systems*, pp. 5998-6008. 2017.
>
> [4]: Murphy, Kevin P. *Machine learning: a probabilistic perspective*. MIT press, 2012.
>
> [5]: Jumper, John, Richard Evans, Alexander Pritzel, Tim Green, Michael Figurnov, Olaf Ronneberger, Kathryn Tunyasuvunakool et al. "Highly accurate protein structure prediction with AlphaFold." *Nature* (2021): 1-11.

---

> ### Author Response · Authors · 2021-08-12
> **Short response to Reviewer wufR**
>
> We greatly appreciate the reviewer's comments, especially it offers us insightful perspectives on how to present new technical results to a broader audience in the community.
>
> > Although the variants seem to be inspired by the mathematical representations of some integrations, the proposed variants have only a few differences from the standard Transformer in the source code. I suggest that the author can list all these differences explicitly to make them clear.
>
> - For practitioners of attentions, all the differences with the standard attention are listed in the current draft.
>
>     - No softmax: abstract, equation (6), (7), (10), (16); reason to do so: Section 3.1.1 and Assumption 3.2
>     - New layer normalization: abstract, equation (6), (7), line 105, 107-108, 169-171, Remark 3.1, Table 9.
>     - Galerkin Transformer's scaled dot-product attention is of linear complexity: abstract, line 192-194; a computational complexity Table will be added in Appendix B.
>     - Positional encodings: line 90-95, 195-202, 756-769.
>     - Tweak the Xavier initialization: equation (18), line 215-218, Table 9.
>     - Diagonally dominant initialization: equation (18), line 215-218, Table 9.
>
>     For the reason why we have made these changes, the answers will be self-evident after reading our proof of the approximation capacity as a saddle problem in a Hilbertian setting in Appendix D.
>
>     We also provide some heuristics (without the mathematical rigor we wanted to achieve in the paper) below to help the reviewer to understand our motivation.
>
> ----
>
>
> > the proposed variants have only a few differences from the standard Transformer in the source code.
>
> - This is one of our method's biggest strength, as practitioners of attention can adapt to our code handily with little changes.
>
>    All these small changes are motivated by the Hilbert space theory. They eventually amount to Galerkin Transformer's evaluation accuracy 30% to 1000% better than the current best operator learner, Fourier Neural Operator [1], and it also shows how much further potential the attention mechanism has.
>
> [1]: Li, Zongyi, Nikola Kovachki, Kamyar Azizzadenesheli, Burigede Liu, Kaushik Bhattacharya, Andrew Stuart, and Anima Anandkumar. "Fourier neural operator for parametric partial differential equations." *International Conference on Learning Representations* (2021). https://openreview.net/forum?id=c8P9NQVtmnO
>
>
> ----
>
> > It would be better if there are more detailed explanations (maybe manually calculated FLOPs/memory)
>
> - FLOPS/memory is not a measure of the efficiency of an algorithm completing the same task.
>
> ----
>
> For a more detailed response to each criticism and suggestions, please see below.

---

### Official Review · Reviewer_3j15 · 2021-07-15

**Rating:** 6
**Confidence:** 3

**Summary:**

This paper claims to apply two Transformer variants to a data-driven operator learning problem, where the mappings between infinite-dimensional spaces of functions are learned. However, this paper only discusses the uniform grid problem setting, which can be regarded as 0D/1D/2D time series forecasting problem. After applying an interpolation-based CNN network (CiNN) to get $n_c \times n_c \times d$ features, the authors use their proposed Fourier or Galerkin Transformer variants to encode the features and use another CiNN as a decoding regressor. Compared to the previous work FNO, the novelty of the paper comes from using Transformers to encode features instead of applying convolutions by FFT/iFFT. However, the use of Transformers makes the proposed method hard to conduct a zero-shot super-resolution prediction, which can be regarded as the merit of the neural PDE solvers.


**Limitations And Societal Impact:**

Yes

**Main Review:**


Advantages:
1. The interpretations of non-softmax attention as Fourier expansion and Petrov–Galerkin-type projection seem to be interesting and insightful, while the proposed methods are not novel compared to other linear attention variants.
2. The proposed Fourier or Galerkin Transformer variants show some improvements over the standard softmax normalized scaled dot-product attention and the linear variant on operator learning problems.

Weaknesses:
1. While the application of random Fourier features on Transformers is discussed in the paper, neither Performer nor RFA is compared in the experiments.
2. Unlike the previous work FNO, the use of Transformers makes the proposed method hard to conduct zero-shot super-resolution prediction. In this case, whether the proposed method can be regarded as a neural PDE solver is questionable. And the paper doesn't present any experiments w.r.t this point (i.e., zero-shot super-resolution prediction).

Possible Typo:

line 105: $y \rightarrow y + \sigma(y + \mathrm{Attn}(y))$ are not consistnet with the original definition of Transformer. It could be $y \rightarrow y + \mathrm{Attn}(y) + \sigma(y + \mathrm{Attn}(y))$ if possible.


**Time Spent Reviewing:**

2

---

> ### Author Response · Authors · 2021-08-06
> **Answer to Reviewer 3j15**
>
> We greatly appreciate the reviewer's comments on our work, especially bringing RFA and Performer up, not mentioning fixing our negligent typo back when we desperately tried to shrink the paper to match the page limit.
>
> ----
> > However, the use of Transformers makes the proposed method hard to conduct a zero-shot super-resolution prediction, which can be regarded as the merit of the neural PDE solvers.
>
> - This is not true.
>
>     For a simple benchmark for viscous Burgers' equation, please download a minimal example here: [link to `ex1_burgers_super_res.py`](https://www.dropbox.com/s/kvoszfwf3shsetm/ex1_burgers_super_res.py?dl=0). The training is done on 2048-length sequences, while the evaluation is done on 8192-length sequences. Please refer to the table of the relative evaluation error below, batch size is 4, GT = Galerkin Transformer. FNO = Fourier Neural Operator.
>
>     |        	     | GT 	| FNO (new code) 	| FNO (old code in [1])  |
>     |-------	     |---	|---	            |---	                 |
>     | Rel. err. (eval) |  $1.113\times 10^{-3}$ 	| $4.178\times 10^{-3}$  | $1.262\times 10^{-2}$  |
>
> ----
>
> > The paper doesn't present any experiments w.r.t this point (i.e., zero-shot super-resolution prediction)
>
> - We did not include the zero-shot super-resolution in our work, because we think the math is not there yet.
>
>     Why Transformer-based architecture can do zero-shot super-resolution in inference? We are still fathoming the math of it, therefore we opt for a safer narrative and chose not to include it in the paper.
>
>     Nevertheless, we have shown ***explicitly*** that the Galerkin-type attention's approximation capacity can be written as a Petrov-Galerkin projection (Appendix D equation (35) in the full version of our paper in supplementary material). We have proved the Galerkin Transformer's approximation capacity is sequence-length invariant (Lemma D.4 in the Appendix). However, to get this approximation capacity, we have to bridge using a saddle-point optimization problem, which naturally translates to the training procedure.
>
> ----
>
> > However, this paper only discusses the uniform grid problem setting.
>
> - The best operator learner to date, Fourier Neural Operator [1], can only apply on uniform grid (because FFT/iFFT is used).
>
>     We want to have a fair comparison with the baselines. From the proof in Theorem 3.3 (Assumption D2, D4, line 897 to 907 in the current paper), we can see that if we use degrees of freedom instead of pointwise value, the current theory has no problem adapted to nonuniform grid. For example, we can integral over a non-uniform region like the Nédélec-Whitney form in Differential Geometry/Computational Electromagnetism, please refer to Section D.2 last paragraph on possible generalization of the theory. Nevertheless, even on the uniform grid, this is the first time that $Q(K^TV)$ in a linear attention has been written explicitly as a solution to a min-max problem (thus to get its approximation capacity approved as a Galerkin-type projection).
>
> ----
>
> > The proposed methods are not novel compared to other linear attention variants.
>
> - The approach to explain the attention is different from all other linear varients.
>
>      All other linear variants of attention use a "row-wise" interpretation, as a result, every single of them tried to use the various techniques to approximate the softmax kernel, e.g., see [2] abstract, the approximation in (4); see [3] page 2 introduction. Since the intuition in NLP is:   the new latent representation should be a convex combination for the previous latent representation.
>
>   While in our work, we opt for a new "column-wise" interpretation the first time, and removed softmax ***completely*** the first time by introducing a simple change: apply the layer normalization pre-dot-product like the Gram–Schmidt process (or say, similar to projections in any other orthogonal basis). We introduce the tool of the operator approximation theory in Hilbert spaces the first time to an audience interested in the mathematical theory of attention.
>
>     While our method may look esoteric and completely new to the community, the operator approximation theory in Hilbert spaces have mature tools, for example, like the one we have played with in equation (29) to equation (35) to write the dot-product attention explicitly as solution to an optimization problem. Introducing these tools to the community opens new research possibilities to better understand the attention mechanism.
>
>   Most importantly, these modifications worked! At the same time, we stood on the shoulder of Giants (the best state-of-the-art Fourier Neural Operator [1]), and better it 30% to 1000% in certain PDE operator learning tasks (with a few simple changes to the attention mechanism to mesh better with the Hilbertian interpretation, of course).
>
> ----
> > While the application of random Fourier features on Transformers is discussed in the paper, neither Performer nor RFA is compared in the experiments.
>
> - We appreciate the reviewer bringing the Performer and RFA up. We have compared both RFA and Performer with the Galerkin Transformer.
>
>     A minimal runnable example with other setup matching our operator learner setup is here: [link to `ex1_burgers_random_fourier_features.py`](https://www.dropbox.com/s/asbmaeo2qew1jcj/ex1_burgers_random_fourier_features.py?dl=0). Changing `favor` to `rfa` in the setup dictionary will run RFA as the feature map. For the results please refer to the table below, both train and evaluation are done at a grid of $n=2048$ and batch size is 4, GT = Galerkin Transformer.
>
>     |        	     | GT 	| Performer | Performer+our init	| Random Feature Attention  |
>     |-------	     |---	|---	  |---          |---	                 |
>     | Rel. err. (eval) |  $1.090\times 10^{-3}$ 	| $1.676\times 10^{-3}$  | $1.582\times 10^{-3}$  | $1.715\times 10^{-2}$
>
>     Unsurprisingly, both RFA and Performer use Fourier features to approximate the softmax kernel, thus perform worse than the Galerkin Transformer.
> Like we have demonstrated in the paper, softmax kernel does more harm than good in PDE-related operator learning. In Petrov-Galerkin projections, a negative weight may still contribute (think the `[-1 2 -1]` tridiagonal Laplacian projection matrix) in softmax-less attention. Yet, training a softmax kernel will drive negative similarity score's contribution negligible. Negative interaction between nodes is one of the key in a successful message passing on a graph (or say a mesh).
>
>   One of the punchlines of our work is to convince the researchers working on data-driven approach for PDE-related problems: softmax kernel (or other similar exponential-based normalizations) is like a shackle, we will have more freedom without it.
>
> ----
>
> > Possible typo
>
> - We greatly appreciate that the reviewer spotted this typo. In our code it is indeed what the reviewer has written (cf. line 125-132 in `/code/libs/model.py` in the supplementary material). Now that we have shrunk the Fourier-type attention part as its major purpose is only to introduce the integral interpretation of the $1/n$ normalization, not the approximation capacity, we have returned to the two liner style to present the scale-preserving simple attention:
> $$
> \mathbf{z} =  \mathbf{y} + \operatorname{Attn}_{\dagger} (\mathbf{y}),
> \quad
> \mathbf{y} \mapsto \mathbf{z} +  \sigma(\mathbf{z}).
> $$
>
>
>
> ----
>
> [1]: Li, Zongyi, Nikola Kovachki, Kamyar Azizzadenesheli, Burigede Liu, Kaushik Bhattacharya, Andrew Stuart, and Anima Anandkumar. "Fourier neural operator for parametric partial differential equations." *International Conference on Learning Representations* (2021).  https://openreview.net/forum?id=c8P9NQVtmnO
>
> [2]: Peng, Hao, Nikolaos Pappas, Dani Yogatama, Roy Schwartz, Noah A. Smith, and Lingpeng Kong. "Random feature attention."  *International Conference on Learning Representations* (2021).  https://openreview.net/forum?id=QtTKTdVrFBB
>
> [3]: Choromanski, Krzysztof, Valerii Likhosherstov, David Dohan, Xingyou Song, Andreea Gane, Tamas Sarlos, Peter Hawkins et al. "Rethinking attention with performers." *International Conference on Learning Representations* (2021). https://openreview.net/forum?id=Ua6zuk0WRH
>
> [4]: Katharopoulos, Angelos, Apoorv Vyas, Nikolaos Pappas, and François Fleuret. "Transformers are rnns: Fast autoregressive transformers with linear attention." *In International Conference on Machine Learning*, pp. 5156-5165. PMLR, 2020.
>
> -----
>
> ### Acknowledgement
> We want to express our appreciation to the open-source community. The RFA and Performer codes are modified and simplified from the official repository of [4] to match our setting under an MIT license, main change includes a diagonal-dominant initialization and the positional encoding being concatenated to the latent representation.

---

> > ### Comment · Reviewer_3j15 · 2021-08-19
> > **Official Comment**
> >
> > I have read other reviews and authors' rebuttals. I appreciate the detailed explanations in the rebuttal and the additional experimental results. My concern about the effectiveness of the proposed model for zero-shot super-resolution prediction is resolved. Therefore, I will increase my score from 5 to 6.
> >
> > But I have one more question w.r.t to model evaluation. The previous work (i.e. FNO) evaluates its model on three datasets, including Burgers' Equation, Darcy Flow, and Navier-Stokes Equation, but this paper only report results on the first two datasets. Assuming this paper uses the same data generation approach as FNO, I'm wondering why the results on Navier-Stokes Equation are not reported.

---

> > > ### Author Response · Authors · 2021-08-23
> > > **Response to Reviewer 3j15 on the reason why Navier-Stokes equation is not reported**
> > >
> > > We greatly appreciated that the reviewer has given us another chance.
> > >
> > > > Assuming this paper uses the same data generation approach as FNO, I'm wondering why the results on Navier-Stokes Equation are not reported.
> > >
> > > - Short answer: in fact we do have the results on Navier-Stokes Equation, please check [the attached minimal pipeline example](https://www.dropbox.com/sh/3cip8ozjjjjb9rl/AAAdxa58GomMxMnYZp-h5qxVa?dl=0): `ns_lite.py` and `ex4_navier_stokes_2+1d.py` trained using the same `1cycle` scheduler on ADAM optimizer for 100 epochs.
> > >
> > >     In the comparison table below, FNO networks are using our re-implemented codes instead of the one used in [1] for fairness. Our re-implementation is similar to the ones used in the current [public repository of FNO](https://github.com/zongyi-li/fourier_neural_operator/), but we replaced the `Conv1D` filters by simpler `nn.Linear`, and changed the ReLU activation to SiLU, which is about 30%-100% better than the ones presented in the original FNO paper [1]. FNO 3D is only using 3 modes in the temporal direction, opposed the 8 used in [1] which results over 6m parameters, since we set the time-marching to 10 steps so that 3D and (2+1)D examples are set up the same. Note that we count a single complex parameter as two real parameters.
> > >
> > >     |        	| GT (2+1D)	| FNO (2+1D) 	| FNO 3D   |
> > >     |-------	|---	|---	            |---	   |
> > >     | Rel. err. |  $3.078\times 10^{-3}$ 	| $5.142\times 10^{-3}$  | $6.163\times 10^{-3}$  |
> > >     | Num. params | 861617 	| 926517  | 2462377  |
> > >
> > >     Like the reason why we chose not to report the zero-shot super-resolution, the result of approximating Navier-Stokes Equation (NSE) is not reported because we feel that the math is not there yet.
> > >
> > >     The current narrative of our paper is to bring the operator approximation theory of a saddle point problem to the table, where the theory and the examples mesh well together. In contrast, explaining the approximation to the NSE requires a different branch of mathematics, as stated in our response to Reviewer 48dv, the mathematics of stability theory in ODE approximation to the dynamical systems.
> > >
> > > For the full answer to this question, please see the long version below.
> > >
> > > [1]: Li, Zongyi, Nikola Kovachki, Kamyar Azizzadenesheli, Burigede Liu, Kaushik Bhattacharya, Andrew Stuart, and Anima Anandkumar. "Fourier neural operator for parametric partial differential equations." *International Conference on Learning Representations* (2021).
> > >
> > >
> > > ----
> > >
> > > ## Full response on why results on Navier-Stokes are not reported
> > >
> > > The reason we have not reported the result on the Navier-Stokes equation is actually quite a long story.
> > >
> > > ### Reason 1: math is not there yet
> > > Currently, constrained by essentially a sequential structure of the attention mechanism, the proposed Galerkin Transformer deals with the NSE as a (2+1)D problem. The 2 stands for the spacial dimension, and the "+1" stands for the time marching in the temporal dimension. This is the same with the FNO 2D+time model (Table 1 FNO 2D row, $\nu = 10^{-3}$ column).
> > >
> > > The "time-marching" means that the solution at a single time step is inferred from the solutions at several previous time steps, then the inferred solution is concatenated to the previous solutions, and this concatenated representation becomes the input the model again (we also would like to remark the autoregressiveness here is in the temporal dimension, different with the autoregressive language model using the causal masking in decoders).
> > >
> > > Analyzing the approximation error between $u(\cdot, t)$ and the operator learner's inference $(T_h u)(\cdot, t)$ needs stability theory in ODE time-marching approximation. To be more specific, when using the "column" interpretation of the attention mechanism, a forward propagation scheme with skip-connection structure can be viewed certain time integrating scheme to get an approximation to a certain subset of functions.
> > >
> > > While the Burgers' equation and Darcy flow are more stable (even in the continuum of the Hilbert spaces), or say less "stiff", in the sense that the solution operator has certain spectral smoothing property due to the presence of diffusion, the Navier-Stokes equation has a stiff part and non-stiff part. For the stiff part (unstable with respect to perturbation), we would like to use an implicit scheme (e.g. like implicit Euler $\mathbf{y}\_{t+1} = \text{Attn} (\mathbf{y}\_{t+1})$).
> > >
> > > However, in reality, we could only afford explicit schemes (Leapfrog, explicit Runge-Kutta, etc) in the forward propagation, analyzing the $A$-stability of the attention operator becomes somewhat a necessary piece to the puzzle in our story, if we want to include the Navier-Stokes equation, both theory and numerics.
> > >
> > > After reading some early work on the Lipschitz constant of the attention operator [2] which shows the attention map is NOT Lipschitz (Lipschitz under certain stringent condition), we took a safer path, and chose not to pursue this direction as the math part of the story.
> > >
> > > [2]: Kim, Hyunjik, George Papamakarios, and Andriy Mnih. "The Lipschitz Constant of Self-Attention." (2020). https://openreview.net/forum?id=DHSNrGhAY7W
> > >
> > > ----
> > >
> > > ### Reason 2: 3D FFT/iFFT+conv in frequency domain is extremely fast
> > >
> > > Despite having much better evaluation accuracy, in terms of speed, the (2+1)D time marching Galerkin Transformer is no match to the Fourier Neural Operator 3D model [1] (see also https://github.com/zongyi-li/fourier_neural_operator/blob/master/fourier_3d.py).
> > >
> > > In FNO 3D, the 3D FFT/iFFT is used to get the latent representations. If we treat FFT/iFFT as an efficient algorithm to do a non-trainable change of ("column") basis, then among the three latent representations $Q, K, V$, two of them do not involve trainable parameters, opposed to Transformer's all trainable latent representations. Thus, FNO 3D is much faster than the (2+1)D Galerkin Transformer.
> > > We note that, for the NSE problem, the linear complexity Galerkin Transformer without softmax are already much faster than the conventional quadratic complexity softmax attention-based Transformer.
> > >
> > > |        	| GT (2+1D)	| FNO (2+1D) 	| FNO 3D   | ST (2+1D) |
> > > |-------	|---	|---	            |---	   |--- |
> > > | Average time per training epoch (s) |  $\approx 32$ 	| $\approx 18$  | $\approx 5$  | $\approx 160$ |
> > >
> > > ----
> > >
> > > ### Reason 3: the data are generated using the stream-vorticity formulation
> > >
> > > This reason is more of a preference. The Navier-Stokes data in the FNO paper [1] are generated using the stream function+vorticity formulation of the NSE, please refer to the $\psi$ (stream function) and $\boldsymbol{\omega}$ (vorticity) here: https://github.com/zongyi-li/fourier_neural_operator/blob/master/data_generation/navier_stokes/ns_2d.py. The introduction of a vorticity+stream function makes
> > >  - imposing the incompressibility (divergence free) of the flow automatic because now the velocity is the curl of the stream function;
> > >  - the curl-curl part becomes a Laplacian so that the equation becomes easier to solve.
> > >
> > > However, if there complicated nonhomogeneous boundary conditions, the vorticity formulation has difficulties to enforce the boundary condition.
> > >
> > > Moreover, stream function + vorticity is very good model to play with in two-dimensional spacial domains, but it is not advised to use this formulation in three-dimensional spacial domain.
> > >
> > > The reason for this difficulty is that the stream function is not a scalar function anymore. We prefer formulations with the velocity. However, as of right now, to our best knowledge, no data-driven method has been able to impose the incompressibility condition (divergence free) for the velocity variable exactly as what numerical analysts have been doing for decades: "impose the divergence free condition through a saddle point problem" (Ref. [26] in the draft paper, line 934-936 in Appendix D). Our effort to interpret the network structure of the attention mechanism as solutions to saddle point problems is toward a more interpretable deep learning framework for the data-driven PDE operator approximation problems.

---

### Official Review · Reviewer_48dv · 2021-07-16

**Rating:** 7
**Confidence:** 2

**Summary:**

In this paper, they propose a general operator learner based on a simple attention mechanism to remove the softmax operation in standard scaled dot product attention. which is regarded as a PDE-related operator. They also introduce a new scale-preserving layer normalization module to better support their operator learning. Combing with  Fourier Neural Operator, the proposed model obtains great performance on some benchmarks with superior memory and time efficiency.

**Limitations And Societal Impact:**

Yes, they adequately address the limitations.

**Main Review:**

This paper is written clearly and they regard the standard attention mechanism in Transformer as an operator learning problem with removing the softmax operations. Extensive experiments are conducted to confirm the effectiveness and efficiency of their proposed method.   It would be interesting if this proposed method could be integrated into pre-train language models for speeding up.  Just have a few questions:
1. Could the proposed method be applied to tasks like machine translation or GLUE?
2. Is the proposed method able to model the autoregressive attention such as in NMT or language modelling?

**Time Spent Reviewing:**

2

---

> ### Author Response · Authors · 2021-08-05
> **Response to Reviewer 48dv**
>
> We greatly appreciate the review, especially the questions on relating the current work with a broader audience working on NLP.
>
> 1. We could only answer the first question from the software side.
>     - As of now, the encoder in our code is directly usable for NLP tasks (BERT-like pre-train, sequence query, sentiment analysis). As one might have already seen, the code in the supplementary material has very similar structure to the official PyTorch implementations of `TransformerEncoderLayer` (a single layer of attention) and `MultiheadAttention`. It is almost like a "plug-and-play" after upgrading to PyTorch 1.9, which has added `batch_first`, `norm_eps` arguments to `nn.transformer` to conform with our implementation. Users in the community can call `Transformer` from PyTorch as a wrapper to use our encoder layer as a `custom_encoder`, or use `TransformerEncoder` as a wrapper to use our `FourierTransformerEncoderLayer` directly. We built our code not just for operator learning, but with the NLP community in mind.
>     - As for the decoder part, it needs some extra attention. Since in our work the major interest lies in non-causal operator learning tasks in spacial domain, we did not include the decoder code in the supplementary material. Implementation-wise, the decoder can be implemented with some tricks applied to the one discussed in [1] Section 3.3. Here are [the updated `layers.py` and `model.py`](https://www.dropbox.com/sh/3cip8ozjjjjb9rl/AAAdxa58GomMxMnYZp-h5qxVa?dl=0). Replace this updated `model.py` and `layers.py` with the one in the `libs` folder in the supplementary material we have a decoder at hand that can be called by the PyTorch `TransformerDecoder` wrapper as well. We believe the attention practitioners in the NLP community will be pleasantly surprised to see how good if a learnable scaling can be propagated through encoder and decoder layers without exploding gradients. We plan to add a minimal BLEU benchmark on `IWSLT14` German-to-English in using `fairseq` using `nn.transformer` API can be easily ported with a function name change using the encoders and decoder based on the newly proposed method. We will upload a test later (but not for the paper) for the community to try out.
>     - In the context of PDE operator learning problem, we have offered our explanation in Appendix C.3.3 for the Burgers' equation using the point of view of the energy decay of the wave. For language translation problem, the preservation of "energy", or to a lesser extent, "entropy" (the "orderness") might heuristically help to capture some language-specific traits, opposed to a scale-less encoder/decoder. As for GLUE the reviewer has suggested, we have not tried, and we feel that the ample experiences in PDE-related theories does not directly translate to NLP expertise. So, we would leave this as possible future research for the community, as we believe the introduction of this new method and the never-seen-before new tool (Petrov-Galerkin projection) would add another brick to help the community thrive.
>
> 2. As for the autoregressiveness, algorithmic-wise the newly proposed method might have some potential. From a mathematical point of view, with the initialization we have introduced in the current work, the new attention has more of less of small step size in front, thus equation (5) in the paper can be understood as
> $$
> \mathbf{y}\_{k+1/2}\gets \mathbf{y}\_k + h \operatorname{Attn}(\mathbf{y}\_k )
> \quad \text{and}\quad
> \mathbf{y}\_{k+1} \gets \mathbf{y}\_{k+1/2} + h\, \sigma(\mathbf{y}\_{k+1/2}; \mathbf{x}).
> $$
> This is like a (nonlinear) LeapFrog integrator scheme to approximate differential equations while marching in time, similar to the RNN interpretation in [1,2].
> At each iteration, our proposed attention can be viewed as a nonlinear perturbation to the identity operator. Opting for these integrator-like scheme results our new model being more stable than the dual-LayerNorm approach during training. Please refer to Table 9 in the full version of our paper, that our newly proposed method is 200 times more accurate in evaluation than the default Transformer setup in the Burgers' benchmark.
>
>      From this "marching in time" perspective, adapting our work to be an autoregressive model and constructing a causal masking similar to that of [1] Section 3.3 would be an interesting future research. If the explicit scheme above is stable, it would invoke more research in this direction to design new attentions. For example, symplectic integrator scheme (for systems in Hamiltonian mechanics) to update two latent representations at the same time; or the Runge-Kutta scheme to achieve a better stability. However, back to our current work, introducing the theory of approximating the temporal dimension in this aspect is a totally different beast, it would ask for a totally different branch of mathematics from the one used our current work: the stability theory in the dynamical system vs operator approximation theory in functional analysis in the current work. Therefore, we shall leave this as a possible exciting future work for the community to explore.
>
> ----
> [1]: Katharopoulos, Angelos, Apoorv Vyas, Nikolaos Pappas, and François Fleuret. "Transformers are RNNs: Fast autoregressive transformers with linear attention." In *International Conference on Machine Learning*, pp. 5156-5165. PMLR, 2020.
>
> [2]: Schlag, Imanol, Kazuki Irie, and Jürgen Schmidhuber. "Linear Transformers are secretly fast weight programmers." In *International Conference on Machine Learning*, pp. 9355-9366. PMLR, 2021.

---

> > ### Comment · Reviewer_48dv · 2021-08-19
> > **score unchanged.**
> >
> > I have read other reviews and authors' rebuttals. I appreciate the detailed explanations in the rebuttal. I will keep my score unchanged.

---

> > > ### Author Response · Authors · 2021-09-01
> > > **Additional test for the NLP community to try**
> > >
> > > We appreciate the reviewer's comments again for bringing up possible applications in NLP, and we would like to invite the NLP community to try some simple tests ([download link](https://www.dropbox.com/sh/v78jhe43jsjumar/AABGjGYHmJll8kt673fhSAjNa?dl=0)). Our test is prepared using `fairseq`'s `TransformerModelBase` interface on the IWSLT'14 De-En dataset. Using a dev install of `fairseq`, replacing the `__init__.py` file in the `/fairseq/models/transformer`, we can train the model following the cli command in `README` using `fairseq`'s tokenization.
> > >
> > > The IWSLT'14 De-En dataset in `fairseq` has the standard 160,239 train - 7,283 valid - 6,750 test split. Here we demonstrate how the proposed simple changes in our draft can speed up the training of Transformers significantly.
> > >
> > > For example, in the following table, CT = Classic Transformer in [1], GLN = Galerkin projection-type layernorm in the encoder's attention (+ the trick in equation (18) to initialize the projection matrices)
> > >
> > > |           | CT 	| CT+GLN|
> > > |:-----:	|:-:	|:-:	|
> > > | Valid BLEU after epoch 1     	| 1.68 	| 3.93 	|
> > > | Valid BLEU after epoch 2     	| 3.54  | 11.36	|
> > > | Valid BLEU after epoch 5     	| 20.29 | 26.84 |
> > > | Valid BLEU after epoch 10    	| 30.67 | 32.33 |
> > > | Best valid score model's test BLEU epoch 20 	|  33.56 	| 33.81  |
> > > | Best valid score model's test BLEU epoch 50 	|  34.06 	| 34.17  |
> > >
> > > We also would to acknowledge that since submitting the draft, we found some newer work has systematically studied some numerically similar trick, e.g., scaling down the embedding in [2], to greatly improve the evaluation in certain tasks, and we plan to add several papers we have found in our revision.
> > >
> > > ----
> > >
> > > [1]: Vaswani, Ashish, Noam Shazeer, Niki Parmar, Jakob Uszkoreit, Llion Jones, Aidan N. Gomez, Łukasz Kaiser, and Illia Polosukhin. "Attention is all you need." In *Advances in neural information processing systems*, pp. 5998-6008. 2017.
> > >
> > > [2]: Csordás, Róbert, Kazuki Irie, and Jürgen Schmidhuber. "The Devil is in the Detail: Simple Tricks Improve Systematic Generalization of Transformers." arXiv preprint arXiv:2108.12284 (2021).

---

### Official Review · Reviewer_xknS · 2021-07-17

**Rating:** 7
**Confidence:** 3

**Summary:**

This paper aims to improve the well-established attention mechanism, and challenges the necessity of its softmax normalization through an integral perspective.

**Limitations And Societal Impact:**

Yes

**Main Review:**

This paper aims to improve the well-established attention mechanism, and challenges the necessity of its softmax normalization through an integral perspective. The outcome are two “softmax-free” attention variants, connected respectively to Fourier feature transform (and thereby to several recent work sharing this insight, as noted by the paper) and Petroc-Galerkin projection. For the latter, non-trivial theoretical results on approximation error are presented. The proposed attention variants have practical benefits too: in several operator learning experiments, they outperform baselines in terms of accuracy and efficiency.

Overall I appreciate the paper’s technical significance, and really like it that the paper connects to previous works every now and then. Although the presentation is reasonably clear, I do have some confusion/suggestions:
- Replacing the softmax with a linear-ish interaction hints significant efficiency gain. The proposed method achieves speedup and memory saving over baselines, and the paper would appeal to a broader audience if it promises so upfront.
- I would appreciate it if the narrative does a better job connecting to what the implantation looks like. E.g., walking the readers through a quick algorithm would do.
- Thm. 3.3 proceeds with a discrete space with length n, but isn’t bilinear form is over the Hilbert spaces?  I am not an expert in this direction and did not have the capacity to checkout out the proof, so I might be missing something pretty obvious. It would be great if the authors can clarify.
- Adding onto the above, Thm. 3.3 definitely stills has its value, since in practice the spaces are discretized anyway. But it is intriguing to see that the bound actually does not depend on n. I’m curious whether or not there is a more general version of 3.3, that could hold for the infinite-dimensional Hilbert-space case?

Overall the paper is solid and timely. I vote for acceptance.

Strengths:
- Interesting and timely contribution to “linearizing” attention in transformers.
- Well-designed experiments and solid results.
- The theoretical treatment of attention could inspire future research.

Weaknesses:
- The paper could do a better job in connecting to implementation and benefits in practice
- The key theoretical results (Thm. 3.3) needs further clarification.

Missing reference:
https://arxiv.org/abs/2103.02143

**Time Spent Reviewing:**

2.5

---

> ### Author Response · Authors · 2021-08-04
> **Response to Reviewer xknS**
>
> We greatly appreciate the review, as well as the suggestion on making this paper more accessible to the whole community (we hope we are lucky enough to have the chance to address them in the paper).
>
> - Indeed, we wholeheartedly agree that it would be appealing to a much wider audience if the speedup and memory saving features are brought upfront. We naively thought that "removing the softmax normalization" in the abstract is shocking enough, as softmax in the old architecture is indispensable in taming the explosive gradient produced by the matrix product. Meanwhile, in the conventional "row-based" interpretation of the attention mechanism, a new latent representation position-wise must be a convex combination of the old. Some writings have been moved from later sections upfront in Contributions.
>
> - In the new integral-inspired architecture, heuristically, the layer normalization is applied in a way very similar to the Gram–Schmidt process (or say the Galerkin projection with the same test/trial spaces): the denominator has each test/trial basis's norm square as normalization.
>
>    In the case of attention, the query comes, we want to find keys to match it (test the response), and then use value to construct the latent representation (trial bases in the context of Galerkin method). In this case, the biggest difficulty to establish a sound Hilbertian framework is that the bases to test the response and the trial bases to construct the projector are different, therefore we have to use a Petrov-Galerkin projection interpretation. (Updated on Aug 8, 2021) Following Reviewer xknS's suggestion as well as Reviewer wufR's we plan to use [this Galerkin-type attention diagram](https://i.imgur.com/CRB0Xsp.png) comparing with [the classic softmax attention operator diagram](https://i.imgur.com/CqOjGmZ.png) as the new Figure 1, and move the current Figure 1 to the Appendix B Networks structures.
>
> - We appreciate this question a lot, as we have realized that the answers are buried in the proofs in the Appendix (which is less accessible to a general audience). In conventional methods of approximating an operator (PDE, integral equation, etc.), the underlying space is always infinite dimensional, yet the discretization space is finite dimensional like the reviewer has mentioned.
>
>      For example, on $\Omega\subset \mathbb{R}^m$, let us consider the space of the continuous piecewise linear functions sampled at $n$ grid points of a discretization of $\Omega$. No matter how big that $n$ is, this space is a subspace of $L^2(\Omega)$ (or other more advanced Sobolev spaces). To establish
> the approximation capacity, the reason why we first introduce "the bilinear form in this underlying infinite-dimensional Hilbert space", yet prove "the Riesz map associated to it being bounded below in the discretization space" is three-fold:
>
>     1. The finite dimensional approximation space is dynamically changing during the training stage for a specified layer. Therefore, setting up the lower bound (LBB inf-sup condition), and then proving the approximation capacity in changing spaces is not a good idea.
>     2. From the aspect of the proof of Theorem 3.3, in a static view of the current approximation spaces for a single attention operator, the surjectivity (stability) of the map from the key space to the value space is essential to establish whether the attention operator has enough approximation power (line 829-830, the application of Riesz representation theorem). This condition has only to be established in the discretization space (which is also a Hilbert space).
>     3. This "continuous in the continuum, stable in the discrete" nature is THE key to establish a sequence length-invariant approximation power for Transformer.
>
>         To make an analogy with finite element analysis to approximate (Navier-)Stokes flow, this discrete lower bound has to be proved to be independent of $n$ (or mesh size). To set up the problem, first the Ladyzhenskaya–Babuška–Brezzi (LBB) inf-sup condition in the continuum will be shown ((12.2.10) in [1]); then, for the discrete problem to produce a good approximation, the discrete version of LBB condition ((12.5.1) in [1]) needs to be established independent of (or say uniform with respect to) the mesh size (sequence length, number of grid points). Otherwise, as the mesh gets finer (the sequence length gets longer), the approximation converges toward the worse direction (e.g., locking phenomena, 11.4 in [1], the remedy is to opt for a different way to construct the approximation space). In the context of PDE operator approximation, the LBB in the continuum is to guarantee that the problem in the continuum is well-posed, while the discrete LBB condition (the one we have verified for the attention operator in Lemma D.4) guarantees the discrete saddle point problem that delivers this approximation is well-posed.
>
>     We would like to remark that this is exactly what has been observed in [2] (e.g., Table 3 in the Appendix of [2]: despite having much more parameters, ResNet's approximation gets worse when the mesh is finer).
>
>    Owing to this comment, we plan to
>     - expand the remarks below Theorem 3.3 in Section 3.1.3 to not only emphasize the sequence-length invariant approximation capacity (line 186-191 in the current version), also move some writings in the Appendix D.2 here;
>     - briefly mention the mathematical reason in our answer above for this technicality near line 191.
>
> - The result can be established in any infinite-dimensional Hilbert space whose inner product is integral-like ($L^2$, $\ell^2$, or $H^1(\Omega)$). One direction would be considering a function's countable expansion in an orthonormal basis (e.g., Haar wavelet, Fourier, or Chebyshev), it shall always have a spectral decay property in order to be square integrable. Then the bound in Theorem 3.3 will simply have a cut-off error. The other more interesting direction is to establish the result in a compact yet infinite dimensional subspace of $L^2(\Omega)$ (e.g., on $H^1$ using the methodology in (12.2.12) in [1]). Even though in practice, the discrete inf-sup bound is still needed, inspecting this guides us to design new types of attention operator (see Remark D.2 in our manuscript). Both of these two directions can be interesting future research topics.
>
> [1]: Brenner, S. C., Scott, L. R., & Scott, L. R. (2008). *The mathematical theory of finite element methods* (Vol. 3). New York: Springer.
>
> [2]: Li, Zongyi, Nikola Kovachki, Kamyar Azizzadenesheli, Burigede Liu, Kaushik Bhattacharya, Andrew Stuart, and Anima Anandkumar. "Fourier neural operator for parametric partial differential equations." *International Conference on Learning Representations* (2021).

---

> > ### Comment · Reviewer_xknS · 2021-08-19
> > **Re: Response to Reviewer xknS**
> >
> > Thanks the authors for the extensive efforts to improve the paper! I keep my initial score of 7 unchanged.

---

> > > ### Author Response · Authors · 2021-09-01
> > > **Thanks**
> > >
> > > We would like to thank the reviewer again for replying our response. We would also like to express our thanks to the Area Chair for handling our paper and initiate the discussions.

---

### Author Response · Authors · 2021-08-08
**Summary and roadmap of revision**

We greatly appreciate reviewers for the effort spent on reviewing our work. Learning from the insightful and valuable suggestions and criticisms, we lay out the following roadmap of some revisions to the paper, to better present this new method and a set of new tools to the community.

----

### Changes done to the draft

- **Algorithmic advantage of removing softmax**:
    1. moved the significance of removing the softmax from the attention operator from Broader Impact upfront to Main Contributions, including the summary of the speed-up and memory saving from Table 1 in the original paper, thanks to Reviewer xknS's and Reviewer wufR's suggestion.
     2. replaced the "time spent per backprop iteration" by a more accurate measure of the algorithm's efficiency: "GFLOP per backprop iteration" measured by the CUDA profiler in Table 1, thanks to Reviewer wufR's suggestion. A preliminary version and code are in [the response to Reviewer wufR](https://openreview.net/forum?id=ssohLcmn4-r&noteId=_b4_Paadadn).
     3. added a Table in Appendix B comparing the computational complexity of the matrix multiplications ($(QK^T)V$ and $Q(K^TV)$, with or without softmax. We would like to emphasize that the entry-wise exponential (or cosine/sine like the ones in RFA [1] or Performer [2]) operation having an explicit FLOP of $\ell \gg 1$, which most current literature chose to skip; added a remark on the chain rule complexity when taking the derivative of this matrix multiplication (a preliminary write-up in [the response to Reviewer wufR](https://openreview.net/forum?id=ssohLcmn4-r&noteId=LXE_I-wH32N)).


- **Summary of technical results**: thanks to Reviewer xknS's suggestion, we added a summary upfront in the Contributions of the significance of introducing for the first time a Petrov-Galerkin projection from operator approximation theory in Hilbert spaces to prove the approximation capacity of attention, including shortened paragraphs covering:
    - Thm 3.3 in the current version: different from the "row-wise" or position-wise interpretations of the attention mechanism in all literature, by treating columns of the latent representation as basis functions' degrees of freedom, we introduce the Hilbert space operator approximation theory for a saddle point problem the first time to the community.
    - Proof of Thm 3.3, equation (35) in the current version: bridging the (nonlinear) attention operator without softmax explicitly to have the capacity to express the solution to a saddle point problem (a linear Petrov-Galerkin projection).
    - Remark 3.1.3, Section D.2 in the current version: verifying the Ladyzhenskaya–Babuška–Brezzi (LBB) inf-sup condition for a bilinear form gives guidance to the development of new types of dot-product in attention operators.
    - **Sequence-length invariance**: added a remark before the proof of Thm 3.3 in the Appendix of the full paper: aside from the Remark 3.1.3 (the one below Theorem 3.3 in the 10 page version), explain more how the nature of the "continuous in continuum, stable in discretization" nature of the LBB inf-sup condition leads to a sequence-length invariant accuracy during optimizations for Transformers.
    - Lemma D.4 in the current version: verifying the established scaled dot-product attention's approximation capacity in training is independent of the sequence length.


- **Difference with other attentions**: highlighted the difference of the proposed linear variant (Galerkin-type attention operator) with existing variants of linear attentions thanks to the comments from Reviewer xknS, Reviewer 3j15, and Reviewer wufR.
    1. **No softmax**: remove the softmax completely (original full paper line 47) versus the "approximating softmax" approaches adopted by other notable linear variants [1,2], mention the huge computational gain (add near line 47).
    2. **New layer normalization**: changed the way of applying the LN (original paper line 107-108) to conform more to the nature of Galerkin-type projections (add near line 47).
    3. **New initialization trick**: mentioned the numerical trick (original paper line 213) inspired by the proof of Theorem 3.3 (line 851 to 859 in the original paper) in the Experimental Results (add near line 56).
     4. **A new illustrative diagram**: replaced Figure 1 with a diagram comparing the new scale-preserving Galerkin-type attention operator ([link to the new diagram](https://i.imgur.com/CRB0Xsp.png)) to the classic softmax attention ([link to the new diagram](https://i.imgur.com/CqOjGmZ.png)) thanks to xknS's and Reviewer wufR's suggestions. Move the original Figure 1 to the Appendix B Network Structures.


- Shrank Section 3.1.2 of the Fourier-type attention to allow some paragraphs to be added upfront. This section's major purpose is to introduce the integral interpretation, shrinking it enables us to:
    - re-allocated some word quota on presenting the approximation capacity of the Galerkin-type attention, emphasizing differences with existing linearizing efforts e.g., [1,2,3]; as [1,2,3] are leveraging an exponential-based feature map to approximate the effect of softmax (convex combination).
    - fixed the typo Reviewer 3j15 has found, change the formula to a clearer two-liner formula together with the diagram ([link to the new diagram](https://i.imgur.com/CRB0Xsp.png)).


- **Adding references**: added the Random Feature Attention (RFA) reference [1] Reviewer xknS and Reviewer 3j15 have mentioned. Add RFA's linearization contribution alongside with Performer [2] in Related Works.


----

### Changes done to the codes
For the request by Reviewer 3j15 asking us to compare our model with the encoder replaced by Random Feature Attention or FAVOR+ in Performer, we have added them in the code repository for the community to test, and we have included [a sample code in our response to Reviewer 3j15](https://openreview.net/forum?id=ssohLcmn4-r&noteId=6psTFHAMAti). We have also added zero-shot super-resolution code (sample code in [our response to Reviewer 3j15](https://openreview.net/forum?id=ssohLcmn4-r&noteId=6psTFHAMAti)).

We choose not to include these tests in the revision as we feel that the current story of
- Removing softmax completely from the attention operator to enable a new columnistic interpretation (Assumption 3.2 in the current paper) for the latent representation. Then, we can express the attention operator's approximation capacity as a Petrov-Galerkin projection (equation (35), Appendix D in the current paper), which allows us to leverage the operator approximation theory in Hilbert spaces (Theorem 3.3 in the current paper);
- A Petrov-Galerkin projection is the solution to a saddle point problem in the finite dimensional approximation spaces, which naturally leads to the proof of sequence-length invariant capacity in training once the the discrete Ladyzhenskaya–Babuška–Brezzi inf-sup condition is verified (Lemma D.4 in the current paper);
- Theoretically nice but numerically unstable (Table 9 in the current paper), we tweak the attention furthermore by applying a more Galerkin-projection conforming layer normalization to eventually get a 200 times accuracy gain.
- Standing on the shoulder of the giants (the best operator learners Fourier Neural Operator (FNO)), Galerkin Transformer has surpassed it in certain benchmarks from the FNO paper [4] (30% to 1000% better in terms of evaluation accuracy). Meanwhile, the softmax-less ones also is more accurate than the ones with softmax under the same lab condition (Section 4 and Table 9 in the current paper) with a seed-invariant performance (Figure 7);

is self-contained in terms of the train of thoughts. This shows that the attention mechanism not only has great potential in the operator learning tasks, also making small changes according to the Hilbert space theory leading to such huge gain (Table 9 in the current paper's Appendix). Using the ideas of orthogonality in RFA or Performer but without aiming to approximate softmax can be an exciting future study.

----

### To be added in the code (Done)
Owing to the comments from Reviewer 48dv, we are currently working on contributing our proposed scale-preserving attention to the NLP community. Even though this is not directly relevant to the operator learning story in the current paper, we hope that the community would try our proposals to speed up the training thus reducing the computational environmental impact.

- (Done) Instructions on how to use PyTorch's `TransformerEncoder` wrappers to wrap the encoder layer code in our repository (a preliminary sample code in [the response to Reviewer 48dv](https://openreview.net/forum?id=ssohLcmn4-r&noteId=wjG1TbJ2zpV), the usage of causal masks together adapted from [3] in a testing code, more comprehensive comparisons can be future study).
- (Done) Prepare a BLEU benchmark in `fairseq` for a relatively small dataset (IWSLT'14 De-En). The encoder is using the Galerkin projection-type layernorm scheme, and we invite the NLP community to try it out.

----

[1]: Peng, Hao, Nikolaos Pappas, Dani Yogatama, Roy Schwartz, Noah A. Smith, and Lingpeng Kong. "Random feature attention." *International Conference on Learning Representations* (2021).

[2]: Choromanski, Krzysztof, Valerii Likhosherstov, David Dohan, Xingyou Song, Andreea Gane, Tamas Sarlos, Peter Hawkins et al. "Rethinking attention with performers." *International Conference on Learning Representations* (2021).

[3]: Katharopoulos, Angelos, Apoorv Vyas, Nikolaos Pappas, and François Fleuret. "Transformers are rnns: Fast autoregressive transformers with linear attention." In *International Conference on Machine Learning*, pp. 5156-5165. PMLR, 2020.

[4]: Li, Zongyi, Nikola Kovachki, Kamyar Azizzadenesheli, Burigede Liu, Kaushik Bhattacharya, Andrew Stuart, and Anima Anandkumar. "Fourier neural operator for parametric partial differential equations." *International Conference on Learning Representations* (2021).

---

### Decision · Program_Chairs · 2021-09-27

**Decision:**

Accept (Poster)

**Comment:**

This paper studies Transformer variants for data-driven operator learning problems of parametric partial differential equations: the Fourier-type and the Galerkin-type. This paper includes both theoretical and empirical evidence to show that these two Transformer variants are accurate and efficient for solving various parametric PDEs.

The authors have nice discussions with all the reviewers during the rebuttal period and put great efforts into improving the paper quality. All the reviewers agree that the mapping between Transformer and PDE solvers is interesting and important. Therefore, I recommend acceptance. I hope the authors can revise the paper in the camera-ready version.